# Hidden Progress in Deep Learning:
# SGD Learns Parities Near the Computational Limit

**Boaz Barak**
Harvard University

**Benjamin L. Edelman**
Harvard University

**Surbhi Goel**
Microsoft Research &
University of Pennsylvania

**Sham Kakade**
Harvard University

**Eran Malach**
Hebrew University of Jerusalem

**Cyril Zhang**
Microsoft Research

b@boazbarak.org,        bedelman@g.harvard.edu,        surbhig@cis.upenn.edu
sham@seas.harvard.edu,        eran.malach@mail.huji.ac.il,        cyrilzhang@microsoft.com

## Abstract

There is mounting evidence of *emergent phenomena* in the capabilities of deep learning methods as we scale up datasets, model sizes, and training times. While there are some accounts of how these resources modulate statistical capacity, far less is known about their effect on the *computational* problem of model training. This work conducts such an exploration through the lens of learning a $k$-sparse parity of $n$ bits, a canonical discrete search problem which is statistically easy but computationally hard. Empirically, we find that a variety of neural networks successfully learn sparse parities, with discontinuous phase transitions in the training curves. On small instances, learning abruptly occurs at approximately $n^{O(k)}$ iterations; this nearly matches SQ lower bounds, despite the apparent lack of a sparse prior. Our theoretical analysis shows that these observations are *not* explained by a Langevin-like mechanism, whereby SGD "stumbles in the dark" until it finds the hidden set of features (a natural algorithm which also runs in $n^{O(k)}$ time). Instead, we show that SGD gradually amplifies the sparse solution via a *Fourier gap* in the population gradient, making continual progress that is invisible to loss and error metrics.

## 1 Introduction

In deep learning, performance improvements are frequently observed upon simply scaling up resources (such as data, model size, and training time). While these improvements are often continuous in terms of these resources, some of the most surprising recent advances in the field have been *emergent capabilities*: at a certain threshold, behavior changes qualitatively and *discontinuously*. Through a statistical lens, it is well-understood that larger models, trained with more data, can fit more complex and expressive functions. However, far less is known about the analogous *computational* question: *how does the scaling of these resources influence the success of gradient-based optimization?*

These *phase transitions* cannot be explained via statistical capacity alone: they can appear even when the amount of data remains fixed, with only model size or training time increasing. A timely example is the emergence of reasoning and few-shot learning capabilities when scaling up language models (Radford et al., 2019; Brown et al., 2020; Chowdhery et al., 2022; Hoffmann et al., 2022); Srivastava et al. (2022) identify various tasks which language models are only able to solve if they are larger than a critical scale. Power et al. (2022) give examples of discontinuous improvements in population accuracy ("grokking") when running time increases, while dataset and model sizes remain fixed.

36th Conference on Neural Information Processing Systems (NeurIPS 2022).

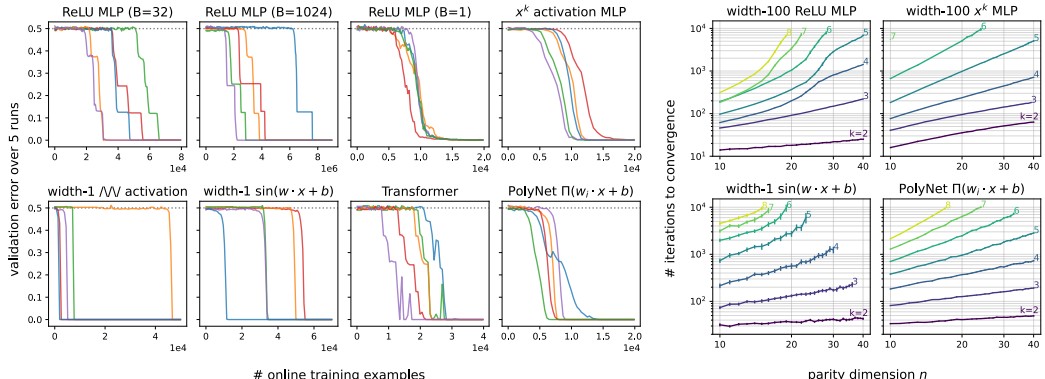

Figure 1: Main empirical findings at a glance. A variety of neural networks, with standard training and initialization, can solve the $(n, k)$-parity learning problem, with a number of iterations scaling as $n^{O(k)}$. *Left:* Training curves under various algorithmic choices (architecture, batch size, learning rate) on the $(n = 50, k = 3)$-parity problem. *Right:* Median convergence times for small $(n, k)$.

In this work, we analyze the computational aspects of scaling in deep learning, in an elementary synthetic setting which already exhibits discontinuous improvements. Specifically, we consider the supervised learning problem of *learning a sparse parity*: the label is the parity (XOR) of $k \ll n$ bits in a random length-$n$ binary string. This problem is computationally difficult for a range of algorithms, including gradient-based (Kearns, 1998) and streaming (Kol et al., 2017) algorithms. We focus on analyzing the resource measure of *training time*, and demonstrate that the loss curves for sparse parities display a phase transition across a variety of architectures and hyperparameters (see Figure 1, left). Strikingly, we observe that SGD finds the sparse subset (and hence, reaches 0 error) with a variety of activation functions and initialization schemes, even with *no* over-parameterization.

A natural hypothesis to explain SGD's success in learning parities, with no visible progress in error and loss for most of training, would be that it simply "stumbles in the dark", performing random search for the unknown target (e.g. via stochastic gradient Langevin dynamics). If that were the case, we might expect to observe a convergence time of $2^{\Omega(n)}$, like a naive search over parameters or subsets of indices. However, Figure 1 *(right)*, already provides some evidence against this "random search" hypothesis: the convergence time adapts to the sparsity parameter $k$, with a scaling of $n^{O(k)}$ on small instances. Notably, such a convergence rate implies that SGD is closer to achieving the *optimal* computation time among a natural class of algorithms (namely, statistical query algorithms).

Through an extensive empirical analysis of the scaling behavior of a variety of models, as well as theoretical analysis, we give strong evidence against the "stumbling in the dark" viewpoint. Instead, there is a *hidden progress measure* under which SGD is steadily improving. Furthermore, and perhaps surprisingly, we show that SGD achieves a computational runtime much closer to the optimal SQ lower bound than simply doing (non-sparse) parameter search. More generally, our investigations reveal a number of notable phenomena regarding the dependence of SGD's performance on resources: we identify phase transitions when varying data, model size, and training time.

## 1.1 Our contributions

**SGD learns sparse parities.** It is known from SQ lower bounds that with a constant noise level, gradient descent on *any* architecture requires at least $n^{\Omega(k)}$ computational steps to learn $k$-sparse $n$-dimensional parities (for background, see Appendix A). We first show a wide variety of positive empirical results, in which neural networks successfully solve the parity problem in a number of iterations which scales near this computational limit:

**Empirical Finding 1.** *For all small instances ($n \leq 30, k \leq 4$) of the sparse parity problem, architectures $\mathcal{A} \in \{$2-layer MLPs, Transformers[1], sinusoidal/oscillating neurons, PolyNets[2]$\}$, initializations in $\{$uniform, Gaussian, Bernoulli$\}$, and batch sizes $1 \leq B \leq 1024$, SGD on $\mathcal{A}$ solves the $(n, k)$-sparse parity problem (w.p. $\geq 0.2$) within at most $c \cdot n^{\alpha k}$ steps, for small constants $c, \alpha$.*

---

[1]With a smaller range of hyperparameters.

[2]A non-standard architecture introduced in this work; see Section 3 for the definition.

**Theoretical analyses of sparse feature emergence.** Our empirical results suggest that, in a number of computational steps matching the SQ limit, SGD is able to solve the parity problem and identify the influential coordinates, without an explicit sparse prior. We give a theoretical analysis which validates this claim.

**Informal Theorem 2.** *On 2-layer MLPs of width $2^{\Theta(k)}$, and with batch size $n^{O(k)}$, SGD converges with high probability to a solution with at most $\epsilon$ error on the $(n, k)$-parity problem in at most $2^{O(k)} \cdot \mathrm{poly}(1/\epsilon)$ iterations.*

We also present a stronger analysis for an idealized architecture (which we call the *disjoint-PolyNet*), which allows for any batch size, and captures the phase transitions observed in the error curves.

**Informal Theorem 3.** *On disjoint-PolyNets, SGD (with any batch size $B \geq 1$) converges with high probability to a solution with at most $\epsilon$ error on the $(n, k)$-parity problem in at most $n^{O(k)} \cdot \log(1/\epsilon)$ iterations. Continuous-time gradient flow exhibits a phase transition: it spends a $1 - o(1)$ fraction of its time before convergence with error $\geq 49\%$.*

Our theoretical and empirical results hold in non-overparameterized regimes (including with a width-1 sinusoidal neuron), in which no fixed kernel, including the neural tangent kernel (NTK) (Jacot et al., 2018), is sufficiently expressive to fit all sparse parities with a large margin. Thus, our findings comprise an elementary example of *combinatorial feature learning*: SGD can only successfully converge by learning a low-width sparse representation.

**Further empirical explorations.** Building upon our core positive results, we provide a wide variety of preliminary experiments, showing sparse parity learning to be a versatile testbed for understanding the challenges and surprises in solving combinatorial problems with neural networks. These include quantities which reveal the continual *hidden progress* behind uninformative training curves (as predicted by the theory), experiments at small sample sizes which exhibit *grokking* (Power et al., 2022), as well as an example where greedy layer-wise learning is impossible but end-to-end SGD can learn the layers jointly.

## 1.2 Related work

We present the most directly related work on feature learning, and learning parities with neural nets. A broader discussion can be found in Appendix A.3.

**SGD and feature learning.** Theoretical analysis of gradient descent on neural networks is notoriously hard, due to the non-convex nature of the optimization problem. That said, it has been established that in some settings, the dynamics of GD keep the weights close to their initialization, thus behaving like convex optimization over the Neural Tangent Kernel (see, for example, (Jacot et al., 2018; Allen-Zhu et al., 2019; Du et al., 2018)). In contrast, it has been shown that in various tasks, moving away from the fixed features of the NTK is essential for the success of neural networks trained with GD (for example (Yehudai and Shamir, 2019; Allen-Zhu and Li, 2019; Wei et al., 2019) and the review in (Malach et al., 2021)). These results demonstrate that feature learning is an important part of the GD optimization process. Our work also focuses on a setting where feature learning is essential for the target task. In our theoretical analysis, we show that the initial population gradient encodes the relevant features for the problem. The importance of the first gradient step for feature learning has been recently studied in (Ba et al., 2022).

**Learning parities with neural networks.** The problem of learning parities using neural networks has been investigated in prior works from various perspectives. It has been demonstrated that parities are hard for gradient-based algorithms, using similar arguments as in the SQ analysis (Shalev-Shwartz et al., 2017; Abbe and Sandon, 2020). One possible approach for overcoming the computational hardness is to make favorable assumptions on the input distribution. Indeed, recent works show that under various assumptions on the input distribution, neural networks can be efficiently trained to learn parities (XORs) (Daniely and Malach, 2020; Shi et al., 2021; Frei et al., 2022; Malach et al., 2021). In contrast to these results, this work takes the approach of intentionally focusing on a hard benchmark task, without assuming that the distribution has some favorable (namely, non-uniform) structure. This setting allows us to probe the performance of deep learning at a known computational limit. Notably, the work of Andoni et al. (2014) provides analysis for learning polynomials (and in

particular, parities) under the uniform distribution. However, their main results require a network of size $n^{O(k)}$ (i.e., extremely overparameterized network), and provides only partial theoretical and empirical evidence for the success of smaller networks. Studying a related subject, some works have shown that neural networks display a spectral bias, learning to fit low-frequency coefficients before high-frequency ones (Rahaman et al., 2019; Cao et al., 2019).

## 2  Preliminaries

We provide an expanded discussion of background and related work in Appendix A.

**Sparse parities.**  For integer $n \geq 1$ and non-empty set $S \subseteq [n]$, the $(n, S)$-*parity function* $\chi_S : \{\pm 1\}^n \to \{\pm 1\}$ is defined as $\chi_S(x) = \prod_{i \in S} x_i$. We define the $(n, S)$-*parity distribution* $\mathcal{D}_S$ as the joint distribution over $(x, y)$[3] where $x$ is drawn from $\mathrm{Unif}(\{\pm 1\}^n)$, the uniform distribution over random length-$n$ sign vectors, and $y := \chi_S(x)$ is the product of the inputs at the indices given by the "relevant features" $S$ (thus, $\pm 1$, depending on whether the number of relevant $-1$ inputs is even or odd). We define the $(n, k)$-*parity learning problem* as the task of recovering the $S$ using samples from $\mathcal{D}_S$, where $S$ is chosen at random from $\binom{[n]}{k}$.

A key fact about parities is that they are orthogonal under the correlation inner product: for $S' \subseteq [n]$,

$$\mathbb{E}_{x \sim \mathrm{Unif}(\{\pm 1\}^n)} [\chi_S(x)\chi_{S'}(x)] = \mathbb{E}_{(x,y) \sim \mathcal{D}_S} [\chi_{S'}(x)\, y] = \begin{cases} 1 & S' = S \\ 0 & \text{otherwise} \end{cases}.$$

That is, a learner who guesses indices $S'$ cannot use correlations (equivalently, the accuracy of the hypothesis $\chi_{S'}$) as feedback to reveal which indices in $S'$ are correct, unless $S'$ is *exactly* the correct subset. This notion of indistinguishability leads to a *computational* lower bound in the statistical query (SQ) model (Kearns, 1998): $\Omega(n^k)$ constant-noise queries are necessary, which is far greater than the statistical limit of $\Theta(\log \binom{n}{k}) \approx k \log n$ samples. The hardness of parity has been used to derive computational hardness results for other settings, like agnostically learning halfspaces (Klivans and Kothari, 2014) and MLPs (Goel et al., 2019). Beyond the restricted computational model of statistical queries, noiseless parities can be learned in $\mathrm{poly}(n)$ time via Gaussian elimination. However, learning sparse *noisy* parities, even at a very small noise level (i.e., $o(1)$ or $n^{-\delta}$), is believed to inherently require $n^{\Omega(k)}$ computational steps.[4] In all, learning sparse parities is a well-studied combinatorial problem which exemplifies the computational difficulty of learning a joint dependence on multiple relevant features.

**Notation for neural networks and training.**  Our main results are presented in the *online learning* setting, with a stream of i.i.d. batches of examples. At each iteration $t = 1, \ldots, T$, a learning algorithm $\mathcal{A}$ receives a batch of $B$ examples $\{(x_{t,i}, y_{t,i})\}_{i=1}^{B}$ drawn i.i.d. from $\mathcal{D}_S$, then outputs a classifier $\widehat{y}_t : \{\pm 1\}^n \to \{\pm 1\}$. We say that $\mathcal{A}$ solves the parity task in $t$ steps (with error $\epsilon$) if

$$\Pr_{(x,y) \sim \mathcal{D}_S} [\widehat{y}_t(x) = y] \geq 1 - \epsilon.$$

We will focus on the case that $\widehat{y}_t = \mathrm{sign}(f(x; \theta_t))$ for some parameters $\theta_t$ in a continuous domain $\Theta$ and for a continuous function $f : \{\pm 1\}^n \times \Theta \to \mathbb{R}$[5], updated with the ubiquitous online learning algorithm of gradient descent (GD), whose update rule is given by

$$\theta_{t+1} \leftarrow (1 - \lambda_t)\theta_t - \eta_t \cdot \nabla_\theta \left( \frac{1}{B} \sum_{i=1}^{B} \ell(y_{t,i}, f(x_{t,i}; \theta_t)) \right), \tag{1}$$

for a loss function $\ell : \{\pm 1\} \times \mathbb{R} \to \mathbb{R}$, learning rate schedule $\{\eta_t\}_{t=1}^{T}$, and weight decay schedule $\{\lambda_t\}_{t=1}^{T}$[6]. The initialization $\theta_0$ is drawn randomly from a chosen distribution.

---

[3]Our theoretical analyses and experiments can tolerate noisy parities, that is, random flipping of the label; see Appendix C.6. For ease of presentation, we state the noiseless setting in the main paper.

[4]This was first explicitly conjectured by Alekhnovich (2003), and has been the basis for several cryptographic schemes (e.g., (Ishai et al., 2008; Applebaum et al., 2009; 2010; Bogdanov et al., 2019)).

[5]When $f(x; \theta) = 0$ in practice (e.g. with sign initialization), we break the tie arbitrarily. We ensure in the theoretical analysis that this does not happen.

[6]We allow different layers to have different learning rate and weight decay schedules.

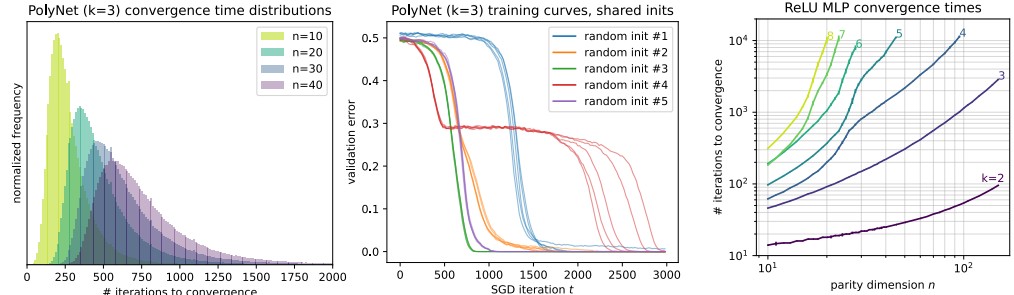

Figure 2: Black-box observations on the training dynamics. *Left:* Histograms of convergence times over $10^6$ random trials, with heavy upper tails but no observed successes near $t = 0$ (unlike random search). *Center:* Loss curves (and thus, convergence time) depend heavily on initialization, not the randomness of SGD; $B = 128, \eta = 0.01$ are shown here. *Right:* The power-law exponent ($\alpha$ such that $t_c \propto n^\alpha$) eventually worsens on larger problem instances.

## 3 Empirical findings

### 3.1 SGD on neural networks learns sparse parities

The central phenomenon of study in this work is the empirical observation that neural networks, with standard initialization and training, can solve the $(n, k)$-parity problem in a number of iterations scaling as $n^{O(k)}$ on small instances. We observed robust positive results for randomly-initialized SGD on the following architectures, indexed by Roman numerals:

- **2-layer MLPs:** ReLU ($\sigma(z) = (z)_+$) or polynomial ($\sigma(z) = z^k$) activation, in a wide variety of width regimes $r \geq k$. Settings (i), (ii), (iii) (resp. (iv), (v), (vi)) use $r = \{10, 100, 1000\}$ ReLU (resp. polynomial) activations. We also consider $r = k$ (exceptional settings (*i), (*ii) ), the minimum width for representing a $k$-wise parity for both activations.

- **1-neuron networks:** Next, we consider non-standard activation functions $\sigma$ which allow a one-neuron architecture $f(x; w) = \sigma(w^\top x)$ to realize $k$-wise parities. The constructions stem from letting $w^* = \sum_{i \in S} e_i$, and constructing $\sigma(\cdot)$ to interpolate (the appropriate scaling of) $\frac{k - w^{*\top} x}{2}$ mod 2 with a piecewise linear $k$-*zigzag* activation (vii), or a degree-$k$ polynomial (viii). Going a step further, a single $\infty$-*zigzag* (ix) or *sinusoidal* (x) neuron can represent *all* $k$-wise parities. In settings (xi), (xii), (xiii), (xiv), we remove the second trainable layer (setting $u = 1$). We find that wider architectures with these activations also train successfully.

- **Transformers:** There is growing interest in using parity as a benchmark for combinatorial function learning, long-range dependency learning, and length generalization in Transformers (Lu et al., 2021; Edelman et al., 2021; Hahn, 2020; Anil et al., 2022; Liu et al., 2022). Motivated by these recent theoretical and empirical works, we consider a simplified specialization of the Transformer architecture to this sequence classification problem. This is the less-robust setting (*iii); the architecture and optimizer are described in Appendix D.1.3.

- **PolyNets:** Our final setting (xv) is the PolyNet, a slightly modified version of the parity machine architecture. Parity machines have been studied extensively in the statistical mechanics of ML literature (see the related work section) as well as in a line of work on 'neural cryptography' (Rosen-Zvi et al., 2002). A parity machine outputs the sign of the product of $k$ linear functions of the input. A PolyNet simply outputs the product itself. Both architectures can clearly realize $k$-sparse parities. The PolyNet architecture was originally motivated by the search for an idealized setting where an end-to-end optimization trajectory analysis is tractable (see Section 4.1); we found in these experiments that this architecture trains very stably and sample-efficiently.

**Robust space of positive results.** All of the networks listed above were observed to successfully learn sparse parities in a variety of settings. We summarize our findings as follows: for all combinations of $n \in \{10, 20, 30\}$, $k \in \{2, 3, 4\}$, batch sizes $B \in \{1, 2, 4, \ldots, 1024\}$, initializations {uniform, Gaussian, Bernoulli}, loss functions {hinge, square, cross entropy}, and architecture configurations $\{(i), (ii), \ldots, (xv)\}$, SGD solved the parity problem (with $100\%$ accuracy, validated

on a batch of $2^{13}$ samples) in at least $20\%$ of 25 random trials, for at least one choice of learning rate $\eta \in \{0.001, 0.01, 0.1, 1\}$. The models converged in $t_c \leq c \cdot n^{\alpha k} \leq 10^5$ steps, for small architecture-dependent constants $c, \alpha$ (see Appendix C). Figure 1 *(left)* shows some representative training curves.

**Less robust configurations.** Settings (*i) and (*ii), where the MLP just barely represents a $k$-sparse parity, and the Transformer setting (*iii), are less robust to small batch sizes. In these settings, the same positive results as above only held for sufficiently large batch sizes: $B \geq 16$. Also, setting (*iii) used the Adam optimizer (which is standard for Transformers); see Appendix D.1.3 for details.

**Phase transitions in training curves.** For almost all of the architectures, we find that that the training curves exhibit phase transitions in terms of running time (and thus, in the online learning setting, dataset size as well): long durations of seemingly no progress, followed by periods of rapid decrease in the validation error. Strikingly, for architectures (v) and (vi), this plateau is absent: the error in the initial phase appears to decrease with a linear slope. See Appendix C.8 for more plots.

## 3.2 Random search or hidden progress?

The remainder of this paper seeks to answer the question: *"By what mechanism does deep learning solve these emblematic computationally-hard optimization problems?"*

A natural hypothesis would be that SGD somehow implicitly performs Monte Carlo random search, "bouncing around" the loss landscape in the absence of a useful gradient signal. Upon closer inspection, several empirical observations clash with this hypothesis:

- **Scaling of convergence times:** Without an explicit sparsity prior in the architecture or initialization, it is unclear how to account for the runtimes observed in experiments, which adapt to the sparsity $k$. The initializations, which certainly do not prefer sparse functions[7], are close to the correct solutions with probability $2^{-\Omega(n)} \ll n^{-k}$.

- **No early convergence:** Over a large number of random trials, no copies of this randomized algorithm get "lucky" (i.e. solve the problem in significantly fewer than the median number of iterations); see Figure 2 *(left)*. The success times of random exhaustive search would be distributed as $\mathrm{Geom}(1/\binom{n}{k})$, whose probability mass is highest at $t = 0$ and decreases monotonically with $t$.

- **Sensitivity to initialization, not SGD samples:** Running these training setups over multiple stochastic batches from a common initialization, we find that loss curves and convergence times are highly correlated with the architecture's random initialization, and are quite concentrated conditioned on initialization; see Figure 2 *(center)*.

- **Elbows in the scaling curves:** For larger $n$, the power-law scaling ceases to hold: the exponent worsens (see Figure 2 *(right)*, as well as the discussion in Appendix C.2). This would not be true for random exhaustive search.

Even these observations, which do not probe the internal state of the algorithm, suggest that exhaustive search is an insufficient picture of the training dynamics, and a different mechanism is at play.

# 4 Theoretical analyses

## 4.1 Provable emergence of the parity indices in high-precision gradients

We now provide a theoretical account for the success of SGD in solving the $(n, k)$-parity problem. Our main theoretical observation is that, in many cases, the *population* gradient of the weights at initialization contains enough "information" for solving the parity problem. That is, given an accurate enough estimate of the initial gradient (by e.g. computing the gradient over a large enough batch size), the relevant subset $S$ can be found.

---

[7]Indeed, under all standard architectures and initialization, the probability that a random network is $\Omega(1)$-correlated with a sparse parity would be $2^{-\Omega(n)}$, since with that probability $1 - o(1)$ of the total influence would be accounted by the $n - k$ irrelevant features.

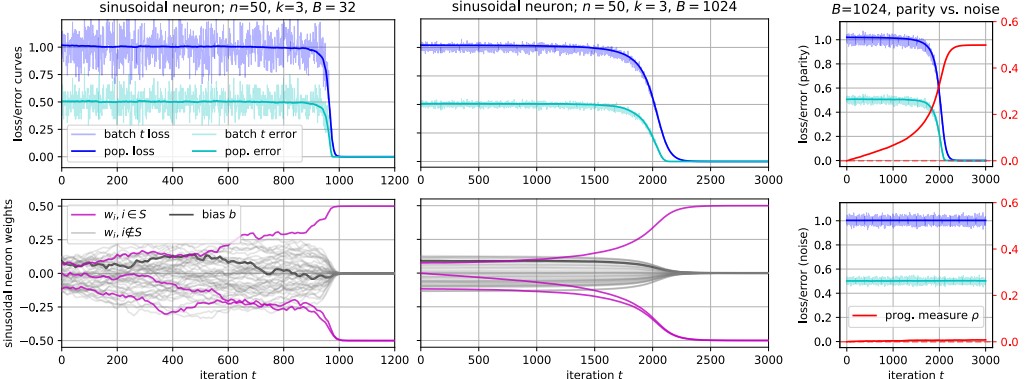

Figure 3: Hidden progress when learning parities with neural networks. *Left, center:* Black-box losses and accuracies exhibit a long plateau and sharp phase transition (top), hiding gradual progress in the SGD iterates (bottom). *Right:* A hidden progress measure which distinguishes gradual feature amplification (top) from training on noise (bottom).

As a warm-up example, consider training a single ReLU neuron $\widehat{y}(x; w) = (w^\top x)_+$ with the correlation loss $\ell(y, \widehat{y}) = -y\widehat{y}$ over $\mathcal{D}_S$, from an all-ones initialization $w = [1 \ \dots \ 1] \in \mathbb{R}^n$. While a single neuron cannot *express* the parity, we observe that the correct subset can be extracted from the population gradient at initialization:

$$\mathop{\mathbb{E}}_{(x,y)\sim\mathcal{D}_S} [\nabla_{w_i}\ell(y, \widehat{y}(x; w))] = \mathop{\mathbb{E}}_{(x,y)\sim\mathcal{D}_S} \left[-y\,\nabla_{w_i}(w^\top x)_+\right] = \mathop{\mathbb{E}}_{(x,y)\sim\mathcal{D}_S} \left[-\chi_S x_i \mathbb{1}\left[\sum_i x_i \geq 0\right]\right].$$

The key insight is that each coordinate in the above expression is a correlation between a parity and the function $x \mapsto -\mathbb{1}[\sum_i x_i \geq 0]$, and thus a Fourier coefficient of this Boolean function. At each relevant coordinate ($i \in S$), the population gradient is the order-$(k-1)$ Fourier coefficient $S \setminus \{i\}$; for the irrelevant features ($i \notin S$), it is instead the order-$(k+1)$ coefficient $S \cup \{i\}$. All we require is a detectable *gap* between these quantities. Formally, letting $f(x; w) = \sigma(w^\top x)$, letting $\widehat{f}(S) := \mathbb{E}[f(x)\chi_S(x)]$ denote the Fourier coefficient of $f$ at $S$, we isolate the desired property:

**Definition 1** (Fourier gap). *For a function $f : \{\pm 1\}^n \to \mathbb{R}$ and $S \subseteq [n]$ of size $k$, we say that $f$ has a $\gamma$-Fourier gap at $S$ if, for every $(k-1)$-element subset $S^- \subset S$ and $(k+1)$-element superset $S^+ \supset S$, it holds that $|\widehat{f}(S^-)| \geq |\widehat{f}(S^+)| + \gamma$.*

For the all-ones initialization, observe $\mathbb{1}[\sum_i x_i \geq 0] = \frac{1+\text{sign}(\sum_i x_i)}{2}$ is just an affine transformation of the majority function of $x$, for which a Fourier gap can be established, with $\gamma = \Theta(n^{-(k-1)/2})$. This arises from closed-form formulas for the Fourier spectrum of majority (see Lemma 2 in Appendix B.1), a landmark result from the harmonic analysis of Boolean functions (Titsworth, 1962; O'Donnell, 2014). Thus, the coordinates in $S$ can be recovered from $\widetilde{O}(1/\gamma^2) = \widetilde{O}(n^{k-1})$ samples; see Proposition 9 in Appendix B.2 for a formal argument.

Carefully extending this insight, we obtain an end-to-end convergence result for ReLU-activation MLP networks with a particular symmetric choice of $\pm 1$ initialization, trained with the hinge loss:

**Theorem 4** (SGD on MLPs learns sparse parities). *Let $\epsilon \in (0, 1)$. Let $k \geq 2$ an even integer, and let $n = \Omega(k^4 \log(nk/\epsilon))$ be an odd integer. Then, there exist a random initialization scheme, $\eta_t$, and $\lambda_t$ such that for every $S \subseteq [n]$ of size $k$, SGD on a ReLU MLP of width $r = \Omega(2^k k \log(k/\epsilon))$, with batch size $B = \Omega(n^k \log(n/\epsilon))$ on $\mathcal{D}_S$ with the hinge loss, outputs a network $f(x; \theta_t)$ with expected[8] loss $\mathbb{E}[\ell(f(x; \theta_t), y)] \leq \epsilon$ in at most $O(k^3 r^2 n/\epsilon^2)$ iterations.*

This does not capture the full range of settings in which we empirically observe successful convergence. First, it requires a sign vector initialization, while we observe convergence with other random initialization schemes (namely, uniform and Gaussian). Second, it requires the batch size to scale with

---

[8]The expectation is over the randomness of initialization, training and sampling $(x, y) \sim \mathcal{D}_S$.

$n^{\Omega(k)}$[9], while we also obtain positive results when $B$ is small (even $B = 1$). Analogous statements for these cases (as well as other activations and losses) would require Fourier gaps for population gradient functions other than majority; lower bounds on the degree-$(k-1)$ coefficients ("*Fourier anti-concentration*") are particularly elusive in the literature, and we leave it as an open challenge to establish them in more general settings. We provide preliminary empirics in Appendix C.1, suggesting that the Fourier gaps in our empirical settings are sufficiently large.[10]

**Low width necessitates feature learning.** We note that in the low-width (non-overparameterized) regimes considered in this work, no fixed kernel (including the neural tangent kernel (Jacot et al., 2018), whose dimensionality is the network's parameter count) can solve the sparse parity problem. The following is a consequence of results in (Kamath et al., 2020; Malach and Shalev-Shwartz, 2022):

**Theorem 5** (Low-width NTK cannot fit all parities). *Let $\Psi : \{\pm 1\}^n \to \mathbb{R}^D$ be any $D$-dimensional embedding with $\sup_x \|\Psi(x)\|_2 \le 1$. Let $R, \epsilon > 0$, and let $\ell$ denote the 0-1 loss or hinge loss. If $DR^2 < \epsilon^2 \cdot \binom{n}{k}$, then there exists some $S \subseteq [n]$ of size $k$ such that*

$$\inf_{\|w\| \le R} \mathbb{E}_{(x,y) \sim \mathcal{D}_S} \left[ \ell(\Psi(x)^\top w, y) \right] > 1 - \epsilon.$$

Thus, our low-width results lie outside the NTK regime, which requires far larger models (size $n^{\Omega(k)}$) to express parities. However, we note that better sample complexity bounds are possible in the NTK regime, with an algorithm more similar to standard SGD (see (Telgarsky, 2022) and Appendix A.3).

### 4.2 Disjoint-PolyNet: exact trajectory analysis for an idealized architecture

In this section, we present an architecture (a version of PolyNets (xv)) which empirically exhibits similar behavior to MLPs and bypasses the difficulty of analyzing Fourier gaps. The *disjoint-PolyNet* takes a product over $k$ linear functions of an equal-sized[11] partition $P_1, \ldots, P_k$ of the input coordinates: $f(x; w_{1:k}) := \prod_{i=1}^k \langle w_i, x_{P_i} \rangle$. As noted in the Section 1.2, this is equivalent to a tree parity machine, with real-valued (instead of $\pm 1$) outputs.

This architecture also requires us to assume that the set $S$ of size $k$ in the $(n, k)$-parity problem is selected such that exactly one index belongs to each disjoint partition, that is, for all $i \in [k]$, $S \cap P_i = 1$. We refer to this problem as the $(n, k)$-disjoint parity problem. Note that there are still $(n')^k = (n/k)^k$ different possibilities for set $S$ under this restriction. For fixed $k$, these represent a constant fraction of the $\binom{n}{k} \approx (ne/k)^k$ (by Stirling's approximation) possibilities for $S$ in the general non-disjoint case.

Consider training a disjoint-PolyNet w.r.t. the correlation loss. Without loss of generality, assume that each relevant coordinate in $S$ is the first element $P_i$. Then, the population gradient is non-zero only at indices $i \in S$:

$$g_i(w_{1:k}) = \mathbb{E} \left[ \nabla_{w_i} \ell(f(x; w_{1:k}), y) \right] = -\mathbb{E} \left[ y \left( \prod_{j \neq i} \langle w_j, x_{P_j} \rangle \right) x_{P_i} \right] = -\left( \prod_{j \neq i} w_{j,1} \right) e_1.$$

This allows us to analyze the gradient flow dynamics of the disjoint-PolyNet, without needing to establish Fourier gaps. For each $i \in [k]$, in this section we treat $w_i$ as a function from $\mathbb{R}_{\ge 0} \to \mathbb{R}^{n'}$ which satisfies the following differential equation: $\dot{w}_i = -g_i(w_{1:k}(t))$. For clarity of exposition, assume all-ones initialization.[12] Then, all of the relevant weights $\{w_{i,1} : i \in [k]\}$ follow the same trajectory. By analyzing the resulting differential equations, we can formally exhibit "phase transition"-like behavior in the fully deterministic gradient flow setting.

**Theorem 6** (Loss plateau for gradient flow on disjoint-PolyNets). *Suppose $k \ge 3$. Let $T(\epsilon)$ denote the smallest time at which the error is at most $\epsilon$. Then,*

$$\frac{T(0.49)}{T(0)} \ge 1 - O\left( (n')^{1-k/2} \right).$$

---

[9]In fact, at this batch size, the correct parity indices emerge in a single SGD step.

[10]Interestingly, we observe that the Fourier gap tends to *increase* over the course of training. This is not captured by our current theoretical analysis.

[11]We assume for simplicity that $n$ is divisible by $k$.

[12]Results for Bernoulli and Gaussian initializations are similar, and can be found in the appendix.

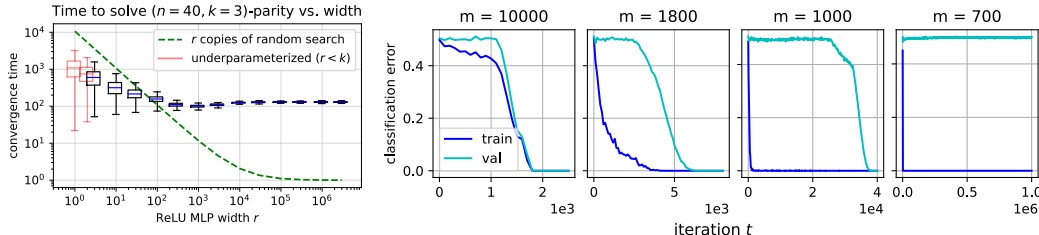

Figure 4: Parity as a sandbox for understanding the effects of model size and dataset size. *Left:* Success times vs. network width $r$ on a fixed $(40, 3)$-parity task: in accordance with the theory, parallelization experiences diminishing returns (unlike expected success times for random search, shown in green). Underparameterized models ($r = 1, 2$) were considered successful upon reaching $55\%$ accuracy. *Right:* Training curves where only the sample size $m$ is varied. The two center panels display *"grokking"*: a large gap between the time to zero train error vs. zero test error.

Informally, the network takes much longer to reach slightly-better-than-trivial accuracy than it takes to go from slightly better than trivial to perfect accuracy. Returning to discrete time, we also analyze the trajectory of disjoint-PolyNets trained with online SGD at any batch size, confirming that a neural network can learn $k$-sparse disjoint parities within $n^{O(k)}$ iterations.

**Theorem 7** (SGD on disjoint-PolyNets learns disjoint parities). *Suppose we train a disjoint-PolyNet, initialized as above, with online SGD. Then there exists an adaptive learning rate schedule such that for any $\epsilon > 0$, with probability 0.99, the error falls below $\epsilon$ within $\tilde{O}\left((n')^{(2k-1)} \log(1/\epsilon)\right)$ steps.*

Extended versions of these theorems, along with proofs, can be found in Appendix B.3.

## 5 Hidden progress: discussion and additional experiments

So far, we have shown that sparse parity learning provides an idealized setting in which neural networks successfully learn sparse combinatorial features, with a mechanism of continual progress hiding behind discontinuous training curves. In this section, we outline preliminary explorations on a broader range of interesting phenomena which arise in this setting. Details are provided in Appendix C, while more systematic investigations are deferred to future work.

**Hidden progress measures for learning parities.** The theoretical and (black-box) empirical results suggest that SGD does *not* learn parities via the memoryless process of random exhaustive search. This suggests the existence of *progress measures*: scalar quantities which are functions of the training algorithm's state (i.e. the model weights $w_t$) and are predictive of the time to successful convergence. We provide some *white-box* investigations which further support the hypothesis of hidden progress, by examining the gradual improvement in quantities other than the training loss. In Appendix C.1, we directly plot the Fourier gaps of the population gradient, as a function of $t$, finding that they are large (within a small constant factor of those of majority) in practice. In Figure 3 and Appendix C.3, we examine the weight movement norm $\rho(w_{0:t}) := \|w_t - w_0\|_\infty$ to reveal hidden progress, motivated by the fact that $w_t - w_0$ is a linearized estimate for the initial population gradient.

**Roles of overparameterization vs. *oversampling*.** An interesting consequence of our analysis is that it illuminates scaling behaviors with respect to a third fundamental resource parameter: *model size*, which we study in terms of network width $r$. If SGD operated by a "random search" mechanism, one would expect width to provide a parallel speedup. Instead, we find that SGD sequentially amplifies progress. The sharp lower tails in Figure 2 *(left)* imply that running $r$ identical copies of SGD does *not* give $(1/r)\times$ speedups; more directly, in Appendix C.4 (previewed in Figure 4 *(left)*), we find that convergence times for sparse parities empirically plateau at large model sizes.

**Emergence of grokking in the finite-sample (multi-pass) setting.** Our main results are presented in the *online learning* setting (fresh minibatches from $\mathcal{D}_S$ at each iteration). While this mitigates the confounding factor of overfitting, it couples the resources of training time and independent samples in a suboptimal way, due to the computational-statistical gap for parity learning. In Appendix C.5,

we find empirically that minibatch SGD (with weight decay) can learn sparse parities, even with smaller sample sizes $m \ll n^k$. We reliably observe the *grokking* phenomenon (Power et al., 2022): an initial overfitting phase, then a *delayed* phase transition in the generalization error; see the two center panels of Figure 4 *(right)*. These results complement and corroborate the findings of Nanda and Lieberum (2022), who analyze the hidden progress of Transformers trained on arithmetic tasks (a setting which also exhibits grokking).

**Deeper networks.** It is a significant challenge (and generally outside the scope of this work) to understand the interactions between network *depth* and computational/statistical efficiency. In Appendix C.7, we show that learning parities with deeper polynomial-activation MLPs comprises a simple counterexample to the *"deep only works if shallow is good"* principle of Malach and Shalev-Shwartz (2019): a deep network can get near-perfect accuracy, even when greedy layer-wise training (e.g. (Belilovsky et al., 2019)) cannot beat trivial performance. By providing positive theory and empirics which elude these simplified explanations of SGD, we hope to point the way to a more complete understanding of learning dynamics in the challenging cases where no apparent progress is made for extended periods of time.

## 6   Conclusion

This work puts forward sparse parity learning as an elementary test case to explore the puzzling features of the role of computational (as opposed to statistical) resources in deep learning. In particular, we have shown that a variety of neural architectures solve this combinatorial search problem, with a number of computational steps nearly matching the sparsity-dependent SQ lower bound. Furthermore, we have shown that despite abrupt phase transitions in the loss and accuracy curves, SGD works by gradually amplifying the sparse features "under the hood".

Even in this simple setting, there are several open experimental and theoretical questions. This work largely focuses on the online learning case, which couples training iterations with fresh i.i.d. samples. We believe it would be instructive to investigate parity learning when the three resources of samples, time, and model size are scaled separately. Some very preliminary findings along these lines are presented in Section 3. It is an open problem to extend our theoretical results to the small-batch setting, as well as to the full range of architectures and losses in our experiments. Resolving these questions would require a better understanding of the anti-concentration behavior of Boolean Fourier coefficients, which is much less studied than the analogous concentration phenomena.

Another important follow-up direction is understanding the extent to which these insights extend from parity learning to more complex (including real-world) combinatorial problem settings, as well as the extent to which non-synthetic tasks (in, e.g., natural language processing and program synthesis) embed within them parity-like subtasks of exhaustive combinatorial search. We hope that our results will lead to further progress towards understanding and improving the optimization dynamics behind the recent slew of dramatic empirical successes of deep learning in these types of domains.

**Broader impact.** This work seeks to contribute to the foundational understanding of computational scaling behaviors in deep learning. Our theoretical and empirical analyses are in a heavily-idealized synthetic problem setting. Hence, we see no direct societal impacts of the results in this study.

**Acknowledgements.** We would like to thank Lenka Zdeborová for providing us with references to the statistical physics literature on phase transitions in the learning curves of neural networks, and Matus Telgarsky for bringing to our attention the better sample complexity guarantees of 2-sparse parity learning in the NTK regime. Sham Kakade acknowledges funding from the Office of Naval Research under award N00014-22-1-2377.

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
