# Appendix

## Table of Contents

# A Additional background, preliminaries, and related work

## A.1 Parities: orthogonality and computational hardness

For each integer $n \geq 1$ and nonempty subset of indices $S \subseteq [n]$, define the $(n, S)$-*parity function* $\chi_S(x) = \prod_{i \in S} x_i$, i.e. the parity of the bits at the indices in $S$. We define the $(n, S)$-*parity distribution* $\mathcal{D}_S$ over examples $(x, y)$ as follows: the features $x \sim \text{Unif}(\{\pm 1\}^n)$ are uniform random bits, with labels $y \in \{\pm 1\}$ given by the parity function $y = \chi_S(x)$. For $0 \leq k \leq n$, the corresponding $(n, k)$-*parity learning problem* is the task of identifying an unknown size-$k$ set $S$ (chosen at random), using samples from $\mathcal{D}_S$. With knowledge of $k$ but not $S$, a learner must use the labels to distinguish between $\binom{n}{k}$ possible "relevant feature sets"; thus, the statistical limit for this problem is $\log \binom{n}{k} = \Theta(k \log n)$ samples.

This work leverages the parity problem as a "computationally hard case" for identifying the features $S$ which are relevant to the label. Observe that for any $S' \subseteq [n]$, it holds that

$$\mathop{\mathbb{E}}_{x \sim \text{Unif}(\{\pm 1\}^n)} [\chi_S(x) \chi_{S'}(x)] = \mathop{\mathbb{E}}_{(x,y) \sim \mathcal{D}_S} [\chi_{S'}(x) \, y] = \begin{cases} 1 & S' = S \\ 0 & \text{otherwise} \end{cases}. \tag{2}$$

That is, a learner who guesses indices $S'$ cannot use correlations as feedback to reveal which (or how many) indices in $S'$ are correct, unless $S'$ is *exactly* the correct subset. In this sense, for the $(n, k)$-parity problem, the $\binom{n}{k} - 1$ wrong answers are indistinguishable from each other. Thus, the structure of this problem forces this form of learner (but not necessarily all learning algorithms) to perform exhaustive search over subsets.

Property 2 (a.k.a. the orthogonality of parities under the correlation inner product) implies that any function $f : \{\pm 1\}^n \to \mathbb{R}$ has a unique Fourier expansion (see, e.g. (O'Donnell, 2014)):

$$f(x) = \sum_{S \subseteq [n]} \widehat{f}(S) \chi_S(x), \qquad \widehat{f}_S = \mathop{\mathbb{E}}_{x \sim \text{Unif}(\{\pm 1\}^n)} [f(x) \chi_S(x)]. \tag{3}$$

In the statistical query (SQ) model (Kearns, 1998), Property (2) implies computational lower bounds. In this model, a learner, rather than having access to examples drawn from the distribution, can query an oracle, which responds with noisy estimates of the query over the distribution. Namely, each iteration the learner outputs a query $q_i : \{\pm 1\}^n \to [-1, 1]$, and the oracle returns some value $v_i$ satisfying $|v_i - \mathbb{E}_{(x,y) \sim \mathcal{D}} [q_i(x, y)]| \leq \tau$, for some tolerance parameter $\tau > 0$. It can be shown that Equation 2 implies that each query will have a non-trivial correlation only with a small fraction (namely, $1/\tau^2$) of the possible parities, which then implies that an SQ algorithm which solves the $(n, k)$-parity problem using $T$ queries of tolerance $\tau$ must satisfy $T/\tau^2 \geq \Omega(n^k)$. This constitutes a lower bound on the number of queries (and/or on the tolerance), which indicates that essentially, SQ algorithms cannot do much better than exhaustive search (going over all the possible choices of size-$k$ subsets).

It should be mentioned that the $(n, k)$-parity problem can be solved efficiently by a learning algorithm that has access to examples (i.e., an algorithm that does not operate in the SQ framework). Specifically, this problem can be solved by the Gaussian elimination algorithm. Moreover, it has been shown that the (Stochastic) Gradient Descent algorithm, discussed in the next section, can also be utilized for solving parities, given accurate enough estimates of the gradient and a very particular choice of neural network architecture Abbe and Sandon (2020). That said, when the accuracy of the gradients is not sufficient, GD suffers from the same SQ lower bound mentioned above (i.e., GD is essentially an SQ algorithm Abbe et al. (2021)).

Learning sparse *noisy* parities, even at a very small noise level (i.e., $o(1)$ or $n^{-\delta}$) is believed to be computationally hard. This was first explicitly conjectured by Alekhnovich (2003), and has been the basis for several cryptographic schemes (e.g., (Ishai et al., 2008; Applebaum et al., 2009, 2010)). For noiseless sparse parities, Kol et al. (2017) show time-space hardness in the setting where $k = \omega(1)$. We present some experiments with noisy parities in Appendix C.6, finding that our empirical results (and theoretical analysis) are robust to $\Theta(1)$ noise.

## A.2 Neural networks and standard training

Next, we establish notation for the standard neural network training pipeline. Our main results are presented in the online learning setting, with a stream of i.i.d. batches of examples. At each iteration

$t = 1, \ldots, T$, a learning algorithm receives a batch of $B$ examples $\{(x_{t,i}, y_{t,i})\}_{i=1}^{B}$ drawn i.i.d. from $\mathcal{D}_S$, then outputs a classifier $\widehat{y}_t : \{\pm 1\}^n \to \{\pm 1\}$. If $\mathbb{E}_{(x,y)\sim\mathcal{D}_S}[\mathbb{1}[\widehat{y}_t(x) \neq y]] \leq \epsilon$ (i.e. $\widehat{y}_t$ agrees with the correct parity on at least a $(1 - \epsilon)$ fraction of inputs), the learner is said to have solved the parity problem with error $\epsilon$ in $t$ iterations ($tB$ samples); the smallest $t$ for which this is true is the *convergence time* $t_c$. A learner may also output an initial classifier $\widehat{y}_0$ before observing any data.

This formulation permits *improper* function classes (i.e. other than parities over subsets $S'$) for the parity learning problem. In particular, we will focus on hypothesis classes of continuous functions $f : \{\pm 1\}^n \times \Theta \to \mathbb{R}$, which map to classifiers $\widehat{y}(x) = \text{sign}(f(x;\theta))$ [13]. When $\Theta$ is a vector space over $\mathbb{R}$, a ubiquitous online learning algorithm is gradient descent (GD). For a choice of loss function $\ell : \{\pm 1\} \times \mathbb{R} \to \mathbb{R}$, initialization $\theta_0$, learning rate schedule $\{\eta_t\}_{t=1}^{T} \subseteq \mathbb{R}$ and weight-decay schedule $\{\lambda_t\}_{t=1}^{T} \subseteq \mathbb{R}$, GD defines iterative updates

$$\theta_{t+1} \leftarrow (1 - \lambda_t)\theta_t - \eta_t \nabla_\theta \left( \frac{1}{B} \sum_{i=1}^{B} \ell(y_{t,i}, f(x_{t,i}; \theta_t)) \right), \tag{4}$$

where $f$ (the *architecture*) and $\ell$ are assumed to be such that this gradient (more generally, subgradient) is well-defined. In this context, *online* and *stochastic* gradient descent (OGD/SGD) are equivalent names for the update rule (4).

A fundamental object of study in deep learning is the multi-layer perceptron (MLP). In this setting, a 2-layer MLP with *width* $h$ and activation function $\sigma : \mathbb{R} \to \mathbb{R}$, parameterized by $W \in \mathbb{R}^{r \times n}, b \in \mathbb{R}^r, u \in \mathbb{R}^r$, specifies the function

$$f(x; W, b, u) = u^\top \sigma(Wx + b),$$

where $\sigma(\cdot)$ is applied entrywise. It is standard to use GD to jointly update the network's parameters. Our results include positive results about "single neurons": MLPs with width $r = 1$. We note that for our theoretical analysis, when training MLPs with GD, we allow for different learning rate and weight decay schedule for the different layers.

Finally, we will analyze randomized learning algorithms, such as GD with random initialization $\theta_0$, whose iterates $\theta_t$ (and thus classifiers $\widehat{y}$) are random variables even when the samples are not. A learning algorithm has *permutation symmetry* if, for all sequences of data $\{(x_{t,i}, y_{t,i})\}$, the classifiers $\widehat{y}_t \circ \pi$ resulting from feeding $\{(\pi(x_{t,i}), y_{t,i})\}$ to the learner have identical distributions as $\pi$ ranges over all permutations of indices. The neural architectures and initializations (and thus, SGD) considered in this work are permutation-symmetric; for this reason, it is convenient for notation to choose $S = [k]$ as the canonical $(n, k)$-parity learning problem, without loss of generality.

### A.3 Additional related work

**Feature learning using GD on neural networks.** A line of recent work has focused on understanding the feature learning ability of gradient descent dynamics on neural networks. These analyses go beyond the Neural Tangent Kernel (NTK) regime, where they show a separation between learning with fixed features versus GD on neural networks, for these problems. Several of these works assume structure (often "sparse") in the input data which is useful for the prediction task, and helps avoid computational hardness. In contrast, our work focuses on studying hard problems at their computational limit. Here we discuss the most relevant works in detail:

A line of work (Diakonikolas et al. (2020); Yehudai and Ohad (2020); Frei et al. (2020)) focuses on learning a single non-linearity $y = \sigma(w^\top x)$ (typically $\sigma(\cdot)$ is the ReLU or sigmoid) using gradient-based methods. These works obtain polynomial-time convergence guarantees when the distribution satisfies a *spread* condition. These results do not extend to the Boolean hypercube.

Daniely and Malach (2020) also study the problem of learning sparse parities using neural networks. One key difference from our work is that they assume a modified version of the problem, where the input distribution is not uniform over the hypercube, but instead leaks information about the label. In particular, the distribution ensures that the relevant parity bits always have the same value. Shi et al. (2021) generalize this setting by considering a setting where labels are generated based on certain class specific patterns and the data itself is generated using these patterns with some extra

---

[13]When $f(x; \theta) = 0$ in practice (e.g. with sign initialization), we break the tie arbitrarily. We ensure in the theoretical analysis that this does not happen.

background patterns. This also embeds information in the data itself regarding the label, unlike our setting, where the labels are uncorrelated with the input features. Under these structural assumption, the papers study how GD on a two-layer network can learn useful features in polynomial time. Both these analysis also exploits the first gradient step to find useful features. Shi et al. (2021) additionally require a second step to refine the features.

Ba et al. (2022) show how the first gradient step is important for feature leaning. In particular, they show that first update is essentially rank-1 and aligns with the linear component of the underlying function. The functions we consider (parity) do not have a linear component.

Abbe et al. (2022) define a notion of initial alignment between the network at initialization and the target function and show that it is essential to get polynomial time learnability with noisy gradients on a fully connected network. Our MLP results also exploit the correlation between the gradient and the label to ensure that the gradient update gives us signal.

Frei et al. (2022) also study learnability of a parity-like function with $k = 2$ under noisy labels. The paper analyzes early stopping GD for learning the underlying labeling function. Our setup is quite different from theirs and can handle $k > 2$.

In concurrent work, Damian et al. (2022) consider the problem of learning polynomials which depend on few relevant directions using gradient descent on a two-layer network. They assume that the distributional Hessian of the ground truth function spans exactly the subspace of the relevant direction. Using this, they show that gradient descent can learn the relevant subspace with sample complexity scaling as $d^2$ and not $d^p$ where $p$ is the degree of the underlying polynomial as long as the number of relevant directions is much less than $d$. Their proof technique is similar to our two-layer MLP result which also exploits correlation in the first gradient step. However, for our setting, the distributional Hessian has rank 0 and does not satisfy their assumptions.

**Statistical mechanics of machine learning.**  An extensive body of work originating in the statistical physics community has studied phase transitions in the learning curves of neural networks (Gardner and Derrida, 1989; Watkin et al., 1993; Engel and Van den Broeck, 2001). These works typically focus on student-teacher learning in the "thermodynamic limit", in which the number of training examples is $\alpha$ times larger than the input dimension and both are taken to infinity. One of the classic toy architectures analyzed in this literature is the *parity machine* (Mitchison and Durbin, 1989; Hansel et al., 1992; Opper, 1994; Kabashima, 1994; Simonetti and Caticha, 1996). In our work, we introduce PolyNets, a variant of parity machines in which the output is real-valued rather than $\pm 1$; and we theoretically analyze disjoint-Polynets, which are the real-output analogue of the oft-considered parity machines with tree architecture. While much of the statistical mechanics of ML literature focuses on an idealized training limit in which the weights reach a Gibbs distribution equilibrium, there is a strand of the literature that aims to characterize the trajectory of SGD training in the high-dimensional limit with constant-sized sets of ordinary differential equations (Saad and Solla, 1995b,a; Goldt et al., 2019). These papers discuss cases, including problems that share aspects with 2-sparse parities Refinetti et al. (2021), where the network gets stuck in (and then escapes from) a plateau of suboptimal generalization error. Recently, Arous et al. (2021) studied (for rank-one parameter estimation problems) the relative amount of time spent by SGD in an initial high-error "search" phase versus a final "descent" phase, which is reminiscent of the framing of Theorem 6. However, to our knowledge prior work has not shown $k$-sparse parities can be learned with a number of iterations that nearly matches known lower bounds, nor has it specifically studied phase transitions in $k$-sparse parity learning during gradient descent.

**Learning parities with the NTK.**  Another relevant line of work studies learning the parity problem using the neural tangent kernel (NTK) (Jacot et al., 2018). Namely, in some settings, when the network's weights stay close to their initialization throughout the training, SGD converges to a solution that is given by a linear function over the initial features of the NTK. As shown in Theorem 5, learning parities over a fixed set of features requires the size of the model to be $\Omega(n^k)$. In contrast, the model size (number of hidden neurons) considered in this paper does not depend at all on the input dimension $n$. Nevertheless, the NTK analysis does give better sample complexity guarantees than the ones presented in this work, with a somewhat more natural version of SGD. For example, the work of Ji and Telgarsky (2019) demonstrates learning 2-sparse parities using NTK analysis with a sample complexity of $O(n^2)$, which matches the sample complexity lower bound for learning this problem with NTK (see Wei et al. (2019)). Concurrent work by Telgarsky (2022) shows that

this sample complexity can be improved to $O(n)$ once the optimization leaves the NTK regime. However, this analysis is given for networks of size $O(n^n)$, much larger than the networks considered in this paper. We refer the reader to Table 1 in Telgarsky (2022) for a complete comparison of the sample-complexity, run-time and model-size bounds achieved by different works studying 2-sparse parities.

# B Proofs

## B.1 Global convergence for SGD on MLPs

For some even number $r$, consider a ReLU MLP of size $r$:

$$f(x; \theta) = \sum_{i=1}^{r} u_i \sigma(w_i^\top x + b_i)$$

Where $\sigma$ is the ReLU activation $\sigma(x) = \max\{x, 0\}$, $w_1, \ldots, w_r \in \mathbb{R}^{r \times n}, b \in \mathbb{R}^r, u \in \mathbb{R}^r$ and we denote the set of parameters by $\theta = \{w_1, \ldots, w_r, b, u\}$. We denote by $u_i^{(t)}, w_i^{(t)}, b^{(t)}$ and $\theta_t$ the value of the relevant parameters at iteration $t$ of gradient-descent. For brevity, we sometimes denote $w_i = w_i^{(0)}, b_i = b_i^{(0)}, u_i = u_i^{(0)}$. W.l.o.g., we assume that $S = [k]$, and so $y = \chi_{[k]}(x) = \prod_{i=1}^{k} x_i$. Indeed, since the weights' initialization we consider is permutation symmetric, this does not limit the generality of the results. We take $\ell$ to be the **hinge loss**: $\ell(y, \widehat{y}) = \max\{1 - y\widehat{y}, 0\}$. We use the following unbiased initialization:

- For all $1 \leq i \leq r/2$, randomly initialize

$$w_i^{(0)} \sim \text{Unif}\left(\{\pm 1\}^n\right), u_i^{(0)} \sim \text{Unif}\left(\{\pm 1\}\right), b_i^{(0)} \sim \text{Unif}(\{-1 + 1/k, \ldots, 1 - 1/k\})$$

- For all $r/2 < i \leq r$, initialize

$$w_i^{(0)} = w_{i-r/2}^{(0)}, b_i^{(0)} = b_{i-r/2}^{(0)}, u_i^{(0)} = -u_{i-r/2}^{(0)}$$

We start by computing the population gradient at initialization. Using the fact that $\ell'(0, y) = -y$ we get the following:

$$\mathbb{E}\left[\nabla_{w_{i,j}} \ell(f(x; \theta_0), y)\right] = \mathbb{E}\left[-y \nabla_{w_{i,j}} f(x; \theta_0)\right] \tag{5}$$
$$= \mathbb{E}\left[-y u_i \mathbb{1}\left\{w_i \cdot x + b_i > 0\right\} x_j\right].$$

For $j \in [k]$ we have:

$$\mathbb{E}\left[\nabla_{w_{i,j}} \ell(f(x; \theta_0), y)\right] = -u_i \mathbb{E}\left[\left(\prod_{j' \in [k] \setminus \{j\}} x_{j'}\right) \mathbb{1}\left\{w_i \cdot x + b_i > 0\right\}\right]$$

For $j \notin [k]$ we have:

$$\mathbb{E}\left[\nabla_{w_{i,j}} \ell(f(x; \theta_0), y)\right] = -u_i \mathbb{E}\left[\left(\prod_{j' \in [k] \cup \{j\}} x_{j'}\right) \mathbb{1}\left\{w_i \cdot x + b_i > 0\right\}\right]$$

Finally, we have:

$$\mathbb{E}\left[\nabla_{b_i} \ell(f(x; \theta_0), y)\right] = -u_i \mathbb{E}\left[\left(\prod_{j' \in [k]} x_{j'}\right) \mathbb{1}\left\{w_i \cdot x + b_i > 0\right\}\right]$$

Denote

$$g_{i,j} = \mathbb{E}\left[\nabla_{w_{i,j}} \ell(f(x; \theta_0), y)\right], \gamma_i = \mathbb{E}\left[\nabla_{b_i} \ell(f(x; \theta_0), y)\right]$$

For some function $f$ and some subset $S \subseteq [n]$, denote $\widehat{f}(S) = \mathbb{E}[f(x)\chi_S(x)]$.

Denote $\text{LTF}_{w,b}(x) = 1\{w \cdot x + b > 0\}$ and let $\text{Maj}(x) = \text{sign}\left(\sum_{i=1}^{n} x_i\right)$ and observe that, if $|b| < 1$,

$$\text{LTF}_{w,b}(x) = \frac{1}{2} + \frac{1}{2}\text{Maj}(w \odot x)$$

where $w \odot x = (w_1 x_1, \ldots, w_n x_n) \in \{\pm 1\}$. Since $\{\chi_S\}_{S \subseteq [n]}$ is a Fourier Basis, we can write $\text{Maj} = \sum_{S \subseteq [n]} \widehat{\text{Maj}}(S)\chi_S$ and therefore:

$$\text{LTF}_{w,b}(x) = \frac{1}{2} + \frac{1}{2}\text{Maj}(w \odot x) = \frac{1}{2} + \frac{1}{2}\sum_{S \subseteq [n]} \widehat{\text{Maj}}(S)\chi_S(w \odot x)$$

$$= \frac{1}{2} + \frac{1}{2}\sum_{S \subseteq [n]} \widehat{\text{Maj}}(S)\chi_S(w)\chi_S(x)$$

So, for every $S \subseteq [n]$ with $|S| \geq 1$ we have $\widehat{\text{LTF}}_{w,b}(S) = \frac{1}{2}\widehat{\text{Maj}}(S)\chi_S(w)$ and so $|\widehat{\text{LTF}}_{w,b}(S)| = \frac{1}{2}|\widehat{\text{Maj}}(S)|$.

**Lemma 1** (O'Donnell (2014), Section 5.3). *Fix some $k$, and assume that $n \geq 2k^2$. Then, for every $S \subseteq [n]$ s.t. $|S| = k$ it holds that:*

- *If $k$ is even: $\widehat{\text{Maj}}(S) = 0$.*

- *If $k$ is odd:*

$$c_1^2 k^{-2/3} \leq \binom{n}{k}\widehat{\text{Maj}}(S)^2 \leq c_2^2 k^{-2/3}$$

*for some universal constants $c_1, c_2$. More precisely,*

$$\widehat{\text{Maj}}(S) = (-1)^{\frac{k-1}{2}} \frac{\binom{\frac{n-1}{2}}{\frac{k-1}{2}}}{\binom{n-1}{k-1}} \cdot 2^{-(n-1)}\binom{n-1}{\frac{n-1}{2}}$$

Observe that for all $S \subseteq [n]$ s.t. $|S| = k$ it holds that $\widehat{\text{Maj}}(S) = \widehat{\text{Maj}}([k])$ and denote $\xi_k := \widehat{\text{Maj}}([k])$.

Therefore, by the previous lemma we get that for every even $k$ the following holds:

$$\frac{\xi_{k+1}^2}{\xi_{k-1}^2} \leq \frac{c_2^2(k-1)^{2/3}\binom{n}{k-1}}{c_1^2(k+1)^{2/3}\binom{n}{k+1}} \leq \frac{c_2}{c_1}\frac{(k+1)k}{(n-k)(n-k+1)} \leq \frac{8c_2}{c_1}\frac{k^2}{n^2} \tag{6}$$

Also, observe that

**Lemma 2** (Fourier gap for majority). *Fix some $k$ and assume that $n \geq 4k$. Then, Majority has a $\gamma_k$-Fourier gap at $S$ of size $k$ with $\gamma_k = 0.03(n-1)^{-\frac{k-1}{2}}$.*

*Proof.* First we establish a simple relationship between $|\xi_{k-1}|$ and $|\xi_{k+1}|$.

$$|\xi_{k-1}| = \frac{\binom{\frac{n-1}{2}}{\frac{k}{2}-1}}{\binom{n-1}{k-2}}2^{-(n-1)}\binom{n-1}{\frac{n-1}{2}}$$

$$= \frac{n-k}{k-1} \cdot \frac{\binom{\frac{n-1}{2}}{\frac{k}{2}}}{\binom{n-1}{k}} \cdot 2^{-(n-1)}\binom{n-1}{\frac{n-1}{2}}$$

$$= \frac{n-k}{k-1} \cdot |\xi_{k+1}|.$$

Here, the first equation follows from Lemma 1, and the second equation follows by simple algebra using the following equality: $\binom{m}{r} = \frac{m-r+1}{r}\binom{m}{r-1}$.

Now, we can bound the difference,

$$|\xi_{k-1}| - |\xi_{k+1}| = \frac{n-2k+1}{k-1} \cdot |\xi_{k+1}|$$

$$= \frac{n-2k+1}{k-1} \cdot \frac{\binom{\frac{n-1}{2}}{\frac{k}{2}}}{\binom{n-1}{k}} \cdot 2^{-(n-1)} \binom{n-1}{\frac{n-1}{2}}$$

$$\geq \frac{n-2k+1}{k-1} \cdot \left(\frac{n-1}{k}\right)^{-k/2} \cdot e^{-k} \cdot \frac{2\sqrt{2\pi}}{e^2\sqrt{n-1}}$$

$$\geq 0.03(n-1)^{-\frac{k-1}{2}}.$$

Here, the first equality holds from above, the second by Lemma 1, the third inequality holds from standard approximations of the binomial coefficients, and the last inequality follows from the following inequalities: $n - 2k + 1 \geq (n-1)/2$ (by assumption on $n$) and $\frac{\sqrt{2\pi}k^{k/2}}{(k-1)e^{k+2}} > 0.03$ (by standard calculus). This gives us the desired result.

$\square$

**Lemma 3.** *Assume that $k$ is even and that $n \geq 2(k+1)^2$. Then, the following hold:*

1. *If $j \in \{1, \ldots, k\}$ then:*

$$g_{i,j} = -\frac{1}{2} u_i \xi_{k-1} \cdot \chi_{[k]\setminus\{j\}}(w_i)$$

2. *If $j \in \{k+1, \ldots, n\}$ then:*

$$g_{i,j} = -\frac{1}{2} u_i \xi_{k+1} \cdot \chi_{[k]\cup\{j\}}(w_i)$$

3. *$\gamma_i = 0$.*

*Proof.* If $j \in [k]$ then:

$$g_{i,j} = -u_i \mathbb{E}\left[\chi_{[k]\setminus\{j\}}(x)\mathrm{LTF}_{w_i,b_i}(x)\right] = -u_i \widehat{\mathrm{LTF}}_{w_i,b_i}([k]\setminus\{j\})$$

$$= -\frac{1}{2} u_i \widehat{\mathrm{Maj}}([k]\setminus\{j\})\chi_{[k]\setminus\{j\}}(w_i) = -\frac{1}{2} u_i \xi_{k-1} \cdot \chi_{[k]\setminus\{j\}}(w_i)$$

Similarly, if $j \notin [k]$ we have:

$$g_{i,j} = -u_i \mathbb{E}\left[\chi_{[k]\cup\{j\}}(x)\mathrm{LTF}_{w_i,b_i}(x)\right] = -u_i \widehat{\mathrm{LTF}}_{w_i,b_i}([k]\cup\{j\})$$

$$= -\frac{1}{2} u_i \widehat{\mathrm{Maj}}([k]\cup\{j\})\chi_{[k]\cup\{j\}}(w_i) = -\frac{1}{2} u_i \xi_{k+1} \cdot \chi_{[k]\cup\{j\}}(w_i)$$

Finally, we have:

$$\gamma_i = -u_i \widehat{\mathrm{LTF}}_{w_i,b_i}([k]) = -u_i \widehat{\mathrm{Maj}}([k]) = 0$$

$\square$

**Lemma 4.** *Let $\tau > 0$ be some tolerance parameter, fix $\epsilon \in (0,1)$ and let $\eta = \frac{1}{k|\xi_{k-1}|}$. Assume that $k$ is an even number. Fix some $w_1, \ldots, w_k \in \{\pm 1\}^n$, $b_1, \ldots, b_r \in (-1, 1)$ and $u_1, \ldots, u_k \in \{\pm 1\}$. Let $\widehat{w}_i = -\eta\widehat{g}_i$ and $\widehat{b}_i = b_i - \eta\widehat{\gamma}_i$ s.t. $\|\widehat{g}_i - g_i\|_\infty \leq \tau$ and $\|\widehat{\gamma}_i - \gamma_i\| \leq \tau$. Assume the following holds:*

- *For all $i, j \in [k]$ it holds that $w_{i,j} = u_i \cdot \mathrm{sign}\,\xi_{k-1}$.*

- *$b_i = -\frac{1}{2} + \frac{i+1}{k}$*

*Then, if $\tau \leq \frac{|\xi_{k-1}|}{16k\sqrt{2n\log(2k/\epsilon)}}$ and $n \geq \frac{2^{11}k^4 c_2^2 \log(2k/\epsilon)}{c_1^2}$ there exists some $\widehat{u} \in \mathbb{R}^k$ with $\|\widehat{u}\|_\infty \leq 8k$ s.t. $f(x) = \sum_{i=1}^k \widehat{u}_i \sigma(\widehat{w}_i \cdot x + \widehat{b}_i)$ satisfies*

$$\mathbb{E}_x[\ell(f(x), \chi_{[k]}(x))] \leq 16\epsilon k^2 n$$

*Additionally, for all $i$ and all $x$ it holds that $|\sigma(\widehat{w}_i \cdot x + \widehat{b}_i)| \leq n + 1$.*

*Proof.* We start with the following claim.

**Claim 1**: For all $i$ and for all $j \in [k]$ it holds that $\left| \widehat{w}_{i,j} - \frac{1}{2k} \right| \leq \frac{\tau}{|\xi_{k-1}|}$.

**Proof**: First, observe that by the assumption it holds that for all $i, j \in [k]$,

$$u_i \cdot \chi_{[k]\setminus\{j\}}(w_i) = u_i \cdot \prod_{j' \in [k]\setminus\{j\}}^{k} w_{i,j} = u_i \cdot (u_i \cdot \text{sign}\,\xi_{k-1})^{k-1} = \text{sign}\,\xi_{k-1}$$

Now, from Lemma 3 we have:

$$g_{i,j} = -\frac{1}{2} u_i \xi_{k-1} \cdot \chi_{[k]\setminus\{j\}}(w_i) = -\frac{1}{2}|\xi_{k-1}|$$

and so

$$\left| w_{i,j} - \frac{1}{2k} \right| = \left| -\eta \widehat{g}_{i,j} - \frac{1}{2k} \right| = \frac{1}{k}\left| -k\eta \widehat{g}_{i,j} + \frac{g_{i,j}}{|\xi_{k-1}|} \right| = \frac{1}{k|\xi_{k-1}|}|g_{i,j} - \widehat{g}_{i,j}| \leq \frac{\tau}{|\xi_{k-1}|}$$

**Claim 2**: For all $i$ and for all $j > k$ it holds that $|\widehat{w}_{i,j}| \leq \frac{|\xi_{k+1}|+2\tau}{2k|\xi_{k-1}|}$

**Proof**: Using Lemma 3 we have:

$$|\widehat{w}_{i,j}| = \eta |\widehat{g}_{i,j}| \leq \eta(|g_{i,j}| + |g_{i,j} - \widehat{g}_{i,j}|) \leq \eta\left( \frac{|\xi_{k+1}|}{2} + \tau \right) = \frac{|\xi_{k+1}| + 2\tau}{2k|\xi_{k-1}|}$$

**Claim 3**: For all $i$ it holds that $|\widehat{b}_i - b_i| \leq \frac{\tau}{k|\xi_{k-1}|}$

**Proof**: Using Lemma 3 we have:

$$|\widehat{b}_i - b_i| = \eta |\widehat{g}_{i,0}| = \eta |g_{i,j} - \widehat{g}_{i,j}| \leq \eta \tau = \frac{\tau}{k|\xi_{k-1}|}$$

**Claim 4**: Fix $\delta > 0$. Let $h_i$ be a function s.t. $h_i(x) = \sigma(\frac{1}{2k}\sum_{j=1}^{k} x_j + b_i)$ and $\widehat{h}_i$ a function s.t. $\widehat{h}_i(x) = \sigma(\widehat{w}_{i,j} \cdot x + \widehat{b}_i)$. Then, if $\tau \leq \frac{\delta}{2} \frac{k|\xi_{k-1}|}{\sqrt{2n \log(2k/\epsilon)}}$ and $n \geq \frac{32 c_2^2 \log(2k/\epsilon)}{c_1^2 \delta^2}$, the following holds:

1. $\mathbb{P}_{x \sim \{\pm 1\}^n}\left[ |h_i(x) - \widehat{h}_i(x)| \geq \delta \right] \leq \frac{\epsilon}{k}$

2. $|\widehat{h}_i(x)| \leq n + 1$

**Proof**: Let $x \sim \{\pm 1\}^n$, and denote $\tilde{x}_j = \widehat{w}_{i,j} x_j$. Denote $\Delta = \frac{|\xi_{k+1}|+2\tau}{2k|\xi_{k-1}|}$. So, for $j > k$, $\tilde{x}_j$ is a random variable satisfying $|\tilde{x}_j| \leq \Delta$. Furthermore, it holds that $\mathbb{E}\left[ \sum_{j>k} \tilde{x}_j \right] = 0$. Therefore, from Hoeffding's inequality:

$$\mathbb{P}\left[ \left| \sum_{j>k} \tilde{x}_j \right| \geq \Delta \sqrt{\frac{n \log(2k/\epsilon)}{2}} \right] \leq 2\exp\left( -\frac{2\left( \Delta \sqrt{\frac{n \log(2k/\epsilon)}{2}} \right)^2}{n\Delta^2} \right) \leq \frac{\epsilon}{k}$$

Now, fix some $x$ s.t. $\left|\sum_{j>k} \tilde{x}_j\right| \le \Delta \sqrt{\frac{n \log(2k/\epsilon)}{2}}$. In this case we have:

$$|h_i(x) - \widehat{h}_i(x)| \le \left| \frac{1}{2k} \sum_{j=1}^{k} x_j + b_i - \widehat{w}_i \cdot x + \widehat{b}_i \right|$$

$$\le \sum_{j=1}^{k} \left| \frac{1}{2k} x_j - \widehat{w}_{i,j} x_j \right| + \left| \sum_{j>k} \widehat{w}_{i,j} x_j \right| + \left| b_i - \widehat{b}_i \right|$$

$$\le \frac{k\tau}{|\xi_{k-1}|} + \Delta \sqrt{\frac{n \log(2k/\epsilon)}{2}} + \frac{\tau}{k|\xi_{k-1}|}$$

$$= \frac{\tau(k^2+1)}{k|\xi_{k-1}|} + \frac{|\xi_{k+1}| + 2\tau}{2k|\xi_{k-1}|} \cdot \sqrt{\frac{n \log(2k/\epsilon)}{2}}$$

$$= \frac{\tau \left( \sqrt{2}(k^2+1) + \sqrt{n \log(2k/\epsilon)} \right)}{\sqrt{2} k |\xi_{k-1}|} + \frac{|\xi_{k+1}| \sqrt{n \log(2k/\epsilon)}}{2\sqrt{2} k |\xi_{k-1}|}$$

$$\le \frac{\tau \sqrt{2n \log(2k/\epsilon)}}{k|\xi_{k-1}|} + \frac{2\sqrt{2} c_2 \sqrt{\log(2k/\epsilon)}}{c_1 \sqrt{n}}$$

where in the last inequality we use Eq. (6). So, choosing $\tau \le \frac{\delta}{2} \frac{k|\xi_{k-1}|}{\sqrt{2n \log(2k/\epsilon)}}$ and $n \ge \frac{32 c_2^2 \log(2k/\epsilon)}{c_1^2 \delta^2}$ gives the required.

**Claim 5:** Let $h_1, \ldots, h_k$ be the functions defined in the previous claim. Then, there exists weights $u^*$ with $\|u^*\|_\infty \le 8k$ s.t. for $f^*(x) = \sum_{i=1}^{k} u_i^* h_i(x)$ it holds that $f^*(x) = 2\chi_{[k]}(x)$ for all $x \in \{\pm 1\}^n$.

**Proof:** For $i \le k-2$ define $u_i^* = 8k(-1)^{i+1}$ and $u_{k-1}^* = 6k$, $u_k^* = -2k$.

**Proof of Lemma 4:** Choose $\widehat{u} = u^*$. Using Claim 4 and the union bound, w.p. $1-\epsilon$ over $x \sim \{\pm 1\}^n$ it holds that for all $i \in [k]$, $|h_i(x) - \widehat{h}_i(x)| \le \delta$. Therefore, w.p. $\ge 1 - \epsilon$

$$|f(x) - f^*(x)| = \left| \sum_{i=1}^{k} u_i^* (h_i(x) - \widehat{h}_i(x)) \right| \le \sum_{i=1}^{k} |u_i^*| \left| h_i(x) - \widehat{h}_i(x) \right| \le 8k^2 \delta$$

so, choosing $\delta = \frac{1}{8k^2}$ we get that, w.p. at least $1-\epsilon$ over the choice of $x$ it holds that $f(x)\chi_{[k]}(x) \ge 1$. Additionally, for every $x$ it holds that

$$|f(x)| \le \sum_{i=1}^{k} |u_i^*| \left| \widehat{h}_i(x) \right| \le 8k^2(n+1) \le 16k^2 n$$

Therefore, we get:

$$\mathbb{E}_x \left[ \ell(f(x), \chi_{[k]}(x)) \right] \le \epsilon \mathbb{E}_x \left[ \ell(f(x), \chi_{[k]}(x)) | f(x)\chi_{[k]}(x) < 1 \right]$$

$$\le \epsilon \mathbb{E}_x \left[ |f(x)| | f(x)\chi_{[k]}(x) < 1 \right] \le 16\epsilon k^2 n$$

$\square$

**Lemma 5.** *Assume we randomly initialize an MLP using the unbiased initialization defined previously. Consider the following update:*

$$w_i^{(1)} = (1 - \lambda_0) w_i^{(1)} - \eta_0 \widehat{g}_i, \ b_i^{(1)} = b_i^{(1)} - \eta_0 \widehat{\gamma}_i$$

*where*

$$\widehat{g}_i = \frac{1}{B} \sum_{l=1}^{B} \nabla_{w_i} \ell(f(x_{0,l}; \theta_0), y_{0,l}), \ \widehat{\gamma}_i = \frac{1}{B} \sum_{l=1}^{B} \nabla_{b_i} \ell(f(x_{0,l}; \theta_0), y_{0,l})$$

*Let $k$ be even number. Then, for every $\epsilon, \delta \in (0, 1/2)$, denoting $\tau = \frac{|\xi_{k-1}|}{16k\sqrt{2n \log(2k/\epsilon)}}$, if $\eta = \frac{1}{k|\xi_{k-1}|}$, $\lambda_0 = 1$, $r \ge k \cdot 2^k \log(k/\delta)$, $B \ge \frac{2}{\tau^2} \log(4nr/\delta)$ and $n \ge \frac{2^{11} k^4 c_2^2 \log(2k/\epsilon)}{c_1^2}$, w.p. at least $1 - 2\delta$*

*over the initialization and the sample, there exists $\widehat{u} \in \mathbb{R}^r$ with $\|\widehat{u}\|_\infty \leq 8k$ and $\|\widehat{u}\|_2 \leq 8k\sqrt{k}$ s.t.* $f(x) = \sum_{i=1}^r \widehat{u}_i \sigma\left(w_i^{(1)} \cdot x + b_i^{(1)}\right)$ *satisfies*

$$\mathbb{E}_x[\ell(f(x), \chi_{[n]}(x))] \leq 16\epsilon k^2 n$$

*Additionally, it holds that $\|\sigma(W^{(1)} \cdot x + b^{(1)})\|_\infty \leq n + 1$.*

*Proof.* Note that by the choice of initialization it holds that $f(x; W^{(0)}) = 0$, and by the assumption on the loss function $\ell'(f(x; W^{(0)}), y) = -y$. Therefore, we get that $\mathbb{E}\left[\nabla_{w_i}\ell(f(x; W^{(0)}), y)\right] = g_i$ and $\mathbb{E}\left[\nabla_{b_i}\ell(f(x; W^{(0)}), y)\right] = \gamma_i$.

**Claim:** with probability at least $1 - \delta$,

$$\text{for all } i, j: \left\|\widehat{g}_i - \mathbb{E}\left[\nabla_{w_i}\ell(f(x; W^{(0)}), y)\right]\right\|_\infty \leq \tau \text{ and } \left|\widehat{\gamma}_i - \mathbb{E}\left[\nabla_{b_i}\ell(f(x; W^{(0)}), y)\right]\right| \leq \tau \tag{7}$$

**Proof:** Fix some $i, j$ and note that by Hoeffding's inequality,

$$\Pr\left[|\widehat{g}_{i,j} - \mathbb{E}\,\widehat{g}_{i,j}| \geq \tau\right] \leq 2\exp\left(-B\tau^2/2\right) \leq \frac{\delta}{nr + r}$$

and similarly we get $\Pr\left[|\widehat{\gamma}_i - \mathbb{E}\,\widehat{\gamma}_i| \geq \tau\right] \leq \frac{\delta}{nr+r}$. The required follows from the union bound.

Now, assume that Eq. (7) holds. For some random $w_i \sim \{\pm 1\}^n$, the probability that $w_{i,j} = w_{i,j'}$ for all $j, j' \in [k]$ is $2^{-k+1}$. Additionally, for some fixed $i' \in [k]$, the probability that $b_i = -\frac{1}{2} + \frac{i'}{k}$ is $\frac{1}{2k}$. Therefore, for some fixed $i \in [r/2]$ and $i' \in [k]$, with probability $\frac{1}{k \cdot 2^{k-1}}$, $b_i = b_{i+r/2} = -\frac{1}{2} + \frac{i'}{k}$ and either $w_{i,j} = u_i \operatorname{sign} \xi_{k-1}$ or $w_{i+r/2,j} = u_{i+r/2,j} \operatorname{sign} \xi_{k-1}$. Taking $r \geq k \cdot 2^k \log(k/\delta)$, we get that the probability that there is no $i \in [r/2]$ that satisfies the above condition (for fixed $i'$) is:

$$\left(1 - \frac{1}{k2^{k-1}}\right)^{r/2} \leq \exp\left(-\frac{r}{2k2^{k-1}}\right) \leq \frac{\delta}{k}$$

Using the union bound, with probability at least $1 - \delta$, there exists a set of $k$ neurons satisfying the conditions of Lemma 4, and therefore the required follows from the Lemma. $\qquad\square$

We use the following well-known result on convergence of SGD (see for example Shalev-Shwartz and Ben-David (2014)):

**Theorem 8.** *Let $M, \rho > 0$. Fix $T$ and let $\eta = \frac{M}{\rho\sqrt{T}}$ Let $F$ be a convex function and $u^* \in \arg\min_{\|u\|_2 \leq M} f(u)$. Let $u^{(0)} = 0$ and for every $t$, let $v_t$ be some random variable s.t. $\mathbb{E}\left[v_t | u^{(t)}\right] = \nabla_{u^{(t)}} F(u^{(t)})$ and let $u^{(t+1)} = u^{(t)} - \eta v^{(t)}$. Assume that $\|v_t\|_2 \leq \rho$ w.p. 1. Then,*

$$\frac{1}{T}\sum_{t=1}^T F(u^{(t)}) \leq F(u^*) + \frac{M\rho}{\sqrt{T}}$$

We prove the following theorem:

**Theorem 4** (SGD on MLPs learns sparse parities; full statement). *Let $k$ be an even number. Assume we randomly initialize an MLP using the unbiased initialization defined previously. Fix $\epsilon \in (0, 1/2)$ and let $T \geq \frac{2^9 k^3 rn^2}{\epsilon^2}, r \geq k \cdot 2^k \log(8k/\epsilon), B \geq c_1^{-1} 2^8 k^{7/6} n \binom{n}{k-1} \log(128k^3n/\epsilon) \log(32nr/\epsilon), n \geq \frac{2^{11}k^4 c_2^2 \log(128k^3n/\epsilon)}{c_1^2}$. Choose the following learning rate and weight decay schedule:*

- *At the first step, use $\eta_0 = \frac{1}{k|\xi_{k-1}|}, \lambda_0 = 1$ for all weights.*

- *After the first step, use $\eta_t = 0$ for the first layers weights and biases and $\eta_t = \frac{4k^{1.5}}{n\sqrt{r(T-1)}}$ for the second layer weights, with $\lambda_t = 0$ for both layers.*

- *Bias terms are never regularized.*

*Then, the following holds, with expectation over the randomness of the initialization and the sampling of the batches:*

$$\mathbb{E}\left[\min_{t\in[T]}\ell(f(x;\theta_t),y)\right]\leq\epsilon$$

*Proof.* Let $F(u)=\mathbb{E}_x\left[\ell(u^\top\sigma(W^{(1)}x+b^{(1)}),y)\right]$ and notice that $F$ is a convex function. For every $t$, denote

$$v_t=\frac{1}{B}\sum_{l=1}^B\nabla_{u^{(t)}}\ell(f(x_{t,l};\theta_t),y)=\frac{1}{B}\sum_{l=1}^B\nabla_{u^{(t)}}\ell\left(\left(u^{(t)}\right)^\top\sigma(W^{(1)}x_{l,t}+b^{(1)}),y_{l,t}\right)$$

where we use the fact that we don't update the weights of the first layer after the first step. From the above we get $\mathbb{E}[v_t|u^{(t)}]=\nabla_{u^{(t)}}F(u^{(t)})$.

Now, we will show that w.h.p. there exists $u^*$ with good loss. Let $\epsilon'=\frac{\epsilon}{64k^2n},\delta'=\frac{\epsilon}{8}$. Denote $\tau=\frac{|\xi_{k-1}|}{16k\sqrt{2n\log(128k^3n/\epsilon)}}=\frac{|\xi_{k-1}|}{16k\sqrt{2n\log(2k/\epsilon')}}$. Observe that $r\geq k\cdot 2^k\log(k/\delta')$, and using the fact that $|\xi_{k-1}|\geq c_1(k-1)^{-1/3}\binom{n}{k-1}^{-1}$ we get

$$B\geq\frac{2^8k^2\cdot n\log(128k^3n/\epsilon)}{|\xi_{k-1}|}\log(32nr/\delta)=\frac{2}{\tau^2}\log(4nr/\delta')$$

and additionally $n\geq\frac{2^{11}k^4c_2^2\log(2k/\epsilon')}{c_1^2}$.

From the above, applying Lemma 5 with $\epsilon',\delta'$ we get that w.p. $1-\epsilon/4$ there exists $u^*\in\mathbb{R}^r$ with $\|u^*\|_2\leq 8k\sqrt{k}$ s.t. $F(u^*)\leq\epsilon/4$ and for all $i$ and all $x$ it holds that $\|\sigma(W^{(1)}\cdot x+b^{(1)})\|_\infty\leq n+1$. Using this, we get:

$$\|v_t\|_2\leq\frac{1}{B}\sum_{l=1}^B\|\sigma(W^{(1)}x_{l,t}+b^{(1)}),y_{l,t}\|_2\leq\sqrt{r}(n+1)$$

So, we can apply Theorem 8 with $M=8k\sqrt{k}$ and $\rho=2\sqrt{r}n$ and get that, w.p. $1-\epsilon/4$ over the initialization and the first step, it holds that

$$\mathbb{E}_{\text{steps } 2...T}\left[\min_{t\in\{2,...,T\}}\ell(f(x;\theta_t),y)\right]\leq\mathbb{E}\left[\frac{1}{T-1}\sum_{t=2}^T\ell(f(x;\theta_t),y)\right]$$

$$=\mathbb{E}\left[\frac{1}{T-1}\sum_{t=2}^T F(u^{(t)})\right]\leq\epsilon/4+\frac{16k^{1.5}\sqrt{r}n}{\sqrt{T-1}}\leq\epsilon/2.$$

Now, that after the first step we have $u^{(1)}=0$ and therefore $\ell(f(x;\theta_1),y)=1$, and so we always have $\min_{t\in[T]}\ell(f(x;\theta_t),y)\leq 1$. Therefore, taking expecation over all steps we get:

$$\mathbb{E}_{\text{steps } 1...T}\left[\min_{t\in[T]}\ell(f(x;\theta_t),y)\right]\leq\epsilon/2+\epsilon/2=\epsilon.$$

The simplified form $B=\Omega(n^k\log(n/\epsilon))$ in the main paper comes from the fact that $\binom{n}{k-1}\leq n^{k-1}/(k-1)!$. This $1/(k-1)!$ factor dominates the other $\text{poly}(k)$ factors. □

### B.2 Recoverability of the parity indices from Fourier gaps

Given a network architecture where some neuron has a $\gamma$-Fourier gap with respect to the target subset $S$, we quantify how the indices in $S$ can be determined by observing an estimate of the population gradient for a general activation function $\sigma$ and $w_t$:

**Proposition 9** (Fourier gap implies feature recoverability). *For an activation function $\sigma:\mathbb{R}\to\mathbb{R}$, let $f(x;w)=\sigma(w^\top x)$ be the corresponding 1-neuron predictor. Let $\mathcal{D}_S$ be an $(n,k)$-sparse parity distribution. Let $g(w)$ be an estimate[14] for the neuron's population gradient of the correlation loss $\ell$:*

$$\|g(w)-\mathbb{E}_{(x,y)\sim\mathcal{D}_S}[\nabla_w\ell(y,f(x;w))]\|_\infty<\gamma/2.$$

---

[14]For $O(1)$-bounded stochastic gradient estimators, $O\left(\frac{\log n}{\gamma^2}\right)$ samples suffice to obtain such an estimate.

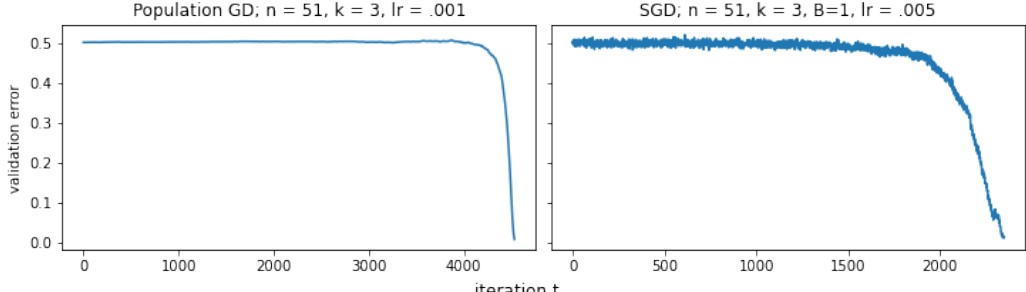

Figure 5: Example training curves for the disjoint-PolyNet trained with correlation loss, which is the setting of Appendix B.3. The left plot shows validation error under population gradient descent with small step size, approximating the setting of Section B.3.1, and the right plot shows a run of SGD with batch size 1, as in Section B.4, though with a constant learning rate schedule. Initialization is i.i.d. standard Gaussian.

*Then, for every $w$ such that $\sigma'(w^\top x)$ has a $\gamma$-Fourier gap at $S$, the $k$ indices at which $g(w)$ has the largest absolute values are exactly the indices in $S$.*

*Proof.* Let $h(x) := \sigma'(w^\top x)$. We compute the population gradient, we we call $\bar{g}(w)$:

$$[\bar{g}(w)]_i = \mathop{\mathbb{E}}_{(x,y)\sim\mathcal{D}_S} [\nabla_{w_i}\ell(y, f(x;w))] = -\mathop{\mathbb{E}}_{(x,y)\sim\mathcal{D}_S} \left[\sigma'(w^\top x)yx_i\right]$$

$$= \begin{cases} \mathbb{E}\left[\sigma'(w^\top x)\prod_{j\in S\setminus\{i\}} x_i\right] & i \in S \\ \mathbb{E}\left[\sigma'(w^\top x)\prod_{j\in S\cup\{i\}} x_i\right] & i \notin S \end{cases}$$

$$= \begin{cases} \widehat{h}(S\setminus\{i\}) & i \in S \\ \widehat{h}(S\cup\{i\}) \le \widehat{h}(S\cup\{i\}) - \gamma & i \notin S \end{cases},$$

where the inequality in the final $i \notin S$ case is due to the Fourier gap property. Then, it holds that for all $i \in S$ we have $|g_i| > \gamma/2$ and for all $i \notin S$ we have $|g_i| < \gamma/2$. Thus, the largest entries of the estimate $g(w)$ occur at the indices in $S$, as claimed. $\square$

## B.3 Global convergence for disjoint-PolyNets

In this section we will develop theory for disjoint-PolyNets trained with correlation loss, as illustrated in Figure 5. Section B.3.1 will consider optimization with gradient flow, and section B.4 will consider optimization with SGD at any batch size $B \ge 1$.

For any $n \ge 1$ and $1 \le k \le n$ such that $n' := n/k$ is an integer, let $P_1, \ldots, P_{n'}$ denote (without loss of generality) the partition $P_i := \{n'(i-1) + 1, \ldots, n' \cdot i\}$. Then, the $(n, k)$-*disjoint-PolyNet* is the neural architecture, with trainable parameters are $\{w_i \in \mathbb{R}^{n'}\}_{i=1}^k$, which outputs

$$f(x; w_{1:k}) := \prod_{i=1}^k \langle w_i, x_{P_i}\rangle.$$

### B.3.1 Gradient flow analysis

For $i$, $\mathbb{E}\left[\nabla_{w_i}\ell(f(x; w_{1:k}), y)\right] = 0$, so the irrelevant weights remain fixed at initialization. For each $i \in [k]$, let $v_i^{(t)}$ be the relevant weight in the $k$th partition.

In gradient flow, the relevant weights evolve according to the following differential equations:

$$\forall i \in [n]: \quad \dot{v}_i = \prod_{j\neq i} v_j$$

**Lemma 6.** *Suppose disjoint-PolyNet for $k > 2$ is initialized such that $\prod_i v_i(0) > 0$, and optimized with gradient flow. Let $\bar{v}_a := \frac{1}{k}\sum_{i=1}^{k}(v_i(0))^2$, and $\bar{v}_g := \left(\prod_{i=1}^{k}(v_i(0))^2\right)^{1/k}$. For any $b \geq 0$ and $i \in [k]$, let $T_i(b) := \arg\sup_{t \geq 0}(|v_i(t)| \leq b)$. Then*

$$T_i(b) \geq \frac{1}{k-2}\left(\bar{v}_a^{1-k/2} - (\bar{v}_a + b^2 - v_i(0)^2)^{1-k/2}\right).$$

*Let $T_i(\infty) := \arg\sup_{t \geq 0}(|v_i(t)| < \infty)$. Then*

$$T_i(\infty) - T_i(b) \leq \frac{1}{k-2}(\bar{v}_g + b^2 - v_i(0)^2)^{1-k/2}.$$

*Proof.* First, observe that the product of the relevant weights is non-decreasing during gradient flow.

$$\frac{d\left(\prod_{i=1}^{k}v_i(t)\right)}{dt} = \sum_{i=1}^{k}\frac{\partial\left(\prod_{i=1}^{k}v_i\right)}{\partial v_i}\cdot\dot{v}_i = \sum_{i=1}^{k}\left(\prod_{j\neq i}v_i\right)^2 > 0$$

Thus, $\prod_i v_i(0) > 0$ implies that $\prod_i v_i(t) > 0$ for all $t$.

Observe that

$$\frac{dv_i^2}{dt} = 2v_i\dot{v}_i = 2\prod_{j=1}^{k}v_j = 2\prod_{j=1}^{k}|v_j|.$$

This implies that for all $i, l \in [t]$,

$$\frac{dv_i^2}{dt} = \frac{dv_l^2}{dt}. \tag{8}$$

In other words, the squares of the relevant weights each follow the same trajectory, shifted according to their initializations. Let $q(t) := (v_i(t))^2 - (v_i(0))^2$, for any $i$. This quantity evolves as follows:

$$\dot{q} = 2\prod_{i=1}^{k}|v_i| = 2\left(\prod_{i=1}^{k}(q(t) + (v_i(0))^2)\right)^{1/2}$$

Since $q(t)$ is strictly increasing, its inverse $q^{-1}$ is well-defined, and we can use the inverse function theorem to characterize $q^{-1}$ for all $t \geq 0$:

$$q^{-1}(c) = \int_0^c \frac{1}{2}\left(\prod_{i=1}^{k}(\gamma + (v_i(0))^2)\right)^{-1/2} d\gamma.$$

We can upper- and lower-bound the integrand by applying Maclaurin's inequality (see page 52 in Hardy et al. (1952)):

$$(\gamma + \bar{v}_g)^k \leq \prod_{i=1}^{k}(\gamma + (v_i(0))^2) \leq (\gamma + \bar{v}_a)^k.$$

The amount of time it takes for $q$ to reach a value of $c$ is thus:

$$\int_0^c \frac{1}{2}\left(\prod_{i=1}^{k}(\gamma + (v_i(0))^2)\right)^{-1/2} d\gamma \geq \int_0^c \frac{1}{2}(\gamma + \bar{v}_a)^{-k/2}d\gamma = \frac{1}{k-2}\left(\bar{v}_a^{1-k/2} - (c + \bar{v}_a)^{1-k/2}\right).$$

Hence, for any $b \geq 0$, for each $i$, $|v_i(t)| \leq b$ as long as

$$t \leq \frac{1}{k-2}\left(\bar{v}_a^{1-k/2} - (\bar{v}_a + b^2 - v_i(0)^2)^{1-k/2}\right).$$

Meanwhile, the amount of time after $q^{-1}(c)$ it takes for $q$ to explode to infinity is

$$\int_c^\infty \frac{1}{2}\left(\prod_{i=1}^{k}(\gamma + (v_i(0))^2)\right)^{-1/2} d\gamma \leq \int_c^\infty \frac{1}{2}(\gamma + \bar{v}_g)^{-k/2}d\gamma = \frac{1}{k-2}(c + \bar{v}_g)^{1-k/2}.$$

Substituting $c = b^2 - v_i(0)^2$, we obtain that the amount of time it takes for $|v_i|$ to grow from $b$ to $\infty$ is

$$\frac{1}{k-2}(\bar{v}_g + b^2 - v_i(0)^2)^{1-k/2}.$$

We can upper- and lower-bound $\dot{q}$ by applying Maclaurin's inequality (see page 52 in Hardy et al. (1952)):

$$2\left(q(t) + \bar{v}_g\right)^{k/2} \leq \dot{q} \leq 2\left(q(t) + \bar{v}_a\right)^{k/2}.$$

When $k = 2$, solving the LHS and RHS differential inequalities yields:

$$\bar{v}_g(e^{2t} - 1) \;\leq\; q(t) \;\leq\; \bar{v}_a(e^{2t} - 1).$$

When $k > 2$, we obtain:

$$\left(\bar{v}_g^{-(k/2-1)} - (k-2)t\right)^{-\frac{1}{k/2-1}} - \bar{v}_g \;\leq\; q(t) \;\leq\; \left(\bar{v}_a^{-(k/2-1)} - (k-2)t\right)^{-\frac{1}{k/2-1}} - \bar{v}_a \quad (9)$$

From the lower bound on $q(t)$, we can infer that the relevant weights all explode to infinity by the following time:

$$t = \frac{1}{(k-2)}\bar{v}_g^{-(k/2-1)} = \frac{1}{(k-2)\left(\prod_i v_i(0)\right)^{1-2/k}}$$

From the upper bound, we can infer that for any $c > 0$, it is the case that $q(t) \leq c$ so long as

$$t \leq \frac{1}{k-2}\left(\bar{v}_a^{-(k/2-1)} - (\bar{v}_a + c)^{-(k/2-1)}\right).$$

Hence, for each $i$, $|v_i(t)| \leq b$ for all

$$t \leq \frac{1}{k-2}\left(\bar{v}_a^{-(k/2-1)} - (\bar{v}_a + b^2 - (v_i(0))^2)^{-(k/2-1)}\right).$$

$\square$

Now we analyze the relationship between the relevant weights and the accuracy of the disjoint-PolyNet.

For $y \in \mathbb{R}$, let

$$\mathrm{sign}(y) = \begin{cases} 1 & \text{if } y > 0 \\ 0 & \text{if } y = 0 \\ -1 & \text{if } y < 0 \end{cases}$$

Then define the error of $f$ with parameters $w_{1:k}$ as

$$\mathrm{err}(w_{1:k}) := \Pr_{x \sim \{\pm 1\}^n}\left[\mathrm{sign}(f(x; w_{1:k})) \neq \chi_S(x)\right]$$

**Lemma 7.** *Let $w$ be any setting of the weights of a disjoint-PolyNet such that $\prod_i v_i > 0$. For ease of notation, let $u_i := w_{i,2:n'}$ be the irrelevant portion of $w_i$. There is a constant $c$ such that*

$$\frac{1}{2} - \frac{1}{2}\prod_{i=1}^k\left(\mathrm{erf}\left(\frac{|v_i|}{\|u_i\|_2\sqrt{2}}\right) + \frac{c\|u_i\|_\infty}{\|u_i\|_2^3}\right) \leq \mathrm{err}(w_{1:k}) \leq 2\sum_{i=1}^k \exp\left(-\frac{|v_i|^2}{\|u_i\|_2^2}\right)$$

*where $\mathrm{erf}$ is the Gauss error function $\mathrm{erf}(y) := \frac{2}{\sqrt{\pi}}\int_0^y e^{-\tau^2}\,d\tau$.*

*Proof.* Let $z_i = x_{(i-1)n'+1}$ be the $i$th relevant coordinate of $x$, and let $z_i^- = x_{(i-1)n'+2:in'}$ be the irrelevant coordinates in $P_i$.

Then we have that the error of $f$ with parameters $w_{1:k}$ is

$$\mathrm{err}(w_{1:k}) = \Pr_x\left[\mathrm{sign}\left(\prod_{i=1}^k w_i^\top x_{P_i}\right) \neq \chi_S(x)\right]$$

$$= \Pr_x\left[\mathrm{sign}\left(\prod_{i=1}^k w_i^\top x_{P_i}\right) = -\chi_S(x)\right] + \Pr_x\left[\prod_{i=1}^k w_i^\top x_{P_i} = 0\right]$$

$$= \Pr_x\left[\mathrm{sign}\left(\prod_{i=1}^k w_i^\top x_{P_i}\right) = -\chi_S(x)\right] + \Pr_x\left[\prod_{i=1}^k w_i^\top x_{P_i} = 0\right]$$

$$= \Pr_x\left[\#\{i \in [k] : \mathrm{sign}(v_i z_i + u_i^\top z_i^-) = -z_i\} \text{ is odd}\right] + \Pr_x\left[\prod_{i=1}^k w_i^\top x_{P_i} = 0\right]$$

$$= \Pr_{z_1^-,\ldots,z_k^-}\left[\#\{i \in [k] : u_i^\top z_i^- > v_i\} \text{ is odd}\right]/2$$

$$+ \Pr_{z_1^-,\ldots,z_k^-}\left[\#\{i \in [k] : u_i^\top z_i^- < -v_i\} \text{ is odd}\right]/2 + \Pr_x\left[\prod_{i=1}^k w_i^\top x_{P_i} = 0\right] \tag{10}$$

$$= \Pr_{z_1^-,\ldots,z_k^-}\left[\#\{i \in [k] : u_i^\top z_i^- > v_i\} \text{ is odd}\right] + \Pr_x\left[\prod_{i=1}^k w_i^\top x_{P_i} = 0\right] \tag{11}$$

$$= \Pr_{z_1^-,\ldots,z_k^-}\left[\#\{i \in [k] : u_i^\top z_i^- > |v_i|\} \text{ is odd}\right] + \Pr_x\left[\prod_{i=1}^k w_i^\top x_{P_i} = 0\right] \tag{12}$$

Line 10 follows because $u_i^\top z_i^-$ and $z_i$ are independent. In line 11 we use that $u_i^\top z_i^-$ is symmetric about 0. Finally, line 12 uses the assumption that $\prod_{i=1}^k v_i > 0$.

We can bound $\Pr_x\left[\prod_{i=1}^k w_i^\top x_{P_i} = 0\right]$ using Hoeffding's inequality:

$$0 \leq \Pr_x\left[\prod_{i=1}^k w_i^\top x_{P_i} = 0\right] \leq \sum_{i=1}^k \Pr_{z_i}[u_i^\top z_i^- = v_i] \leq \sum_{i=1}^k \Pr_{z_i}[u_i^\top z_i^- \geq |v_i|] \leq \sum_{i=1}^k \exp\left(-\frac{|v_i|^2}{\|u_i\|_2^2}\right).$$

The indicator random variables $\mathbb{1}[u_i^\top z_i^- > |v_i|]$ are independent of each other, so the first term in line 12 can be characterized using the distribution of the parity of a sum of independent Bernoulli random variables. Let $X_i \sim \mathrm{Ber}(p_i)$ for $i \in [k]$, and let $X = \sum_{i=1}^k X_i$. The generating function for $X$ is $f(z) = \prod_{i=1}^k((1 - p_i) + p_i z)$. The parity of $X$ then satisfies

$$\Pr[X \text{ is odd}] = \frac{f(1)}{2} - \frac{f(-1)}{2} = \frac{1}{2} - \frac{1}{2}\prod_{i=1}^k(1 - 2p_i).$$

First we will prove the upper bound on $\mathrm{err}$. First, observe that

$$\frac{1}{2} - \frac{1}{2}\prod_{i=1}^k(1 - 2p_i) \leq \sum_{i=1}^k p_i.$$

Thus,

$$\Pr_{z_1^-,\ldots,z_k^-}\left[\#\{i \in [k] : u_i^\top z_i^- > |v_i|\} \text{ is odd}\right] \leq \sum_{i=1}^k \Pr_{z_i}[u_i^\top z_i^- > |v_i|]$$

$$\leq \sum_{i=1}^k \exp\left(-\frac{|v_i|^2}{\|u_i\|_2^2}\right)$$

by Hoeffding's inequality, and we are done.

Now we will prove the lower bound on err. We have

$$\Pr_{z_1^-,\dots,z_k^-}\left[\#\{i \in [k] : u_i^\top z_i^- > |v_i|\} \text{ is odd}\right] = \frac{1}{2} - \frac{1}{2}\prod_{i=1}^{k}(1 - 2\Pr_{z_i}[u_i^\top z_i^- > |v_i|])$$

$$= \frac{1}{2} - \frac{1}{2}\prod_{i=1}^{k}\Pr_{z_i}[|u_i^\top z_i^-| \le |v_i|]. \qquad (13)$$

We can bound this expression using the Berry-Esseen theorem (Berry, 1941; Esseen, 1942). Let $\beta \sim N(0, \|u_i\|_2^2)$. Then, the Berry-Esseen theorem states that there is a constant $c$ (which in practice can be .56) such that for any $i \in [k]$,

$$\left|\Pr_{z_i}[|u_i^\top z_i^-| \le |v_i|] - \Pr[|\beta| \le |v_i|]\right| \le \frac{c\|u_i\|_\infty}{\|u_i\|_2^3}$$

and we can use the characterization

$$\Pr[|\beta| \le |v_i|] = \text{erf}\left(\frac{|v_i|}{\|u_i\|_2\sqrt{2}}\right).$$

Plugging this into equation 13, we obtain the lower bound on err. $\qquad \square$

First, we'll apply Lemma 6 and Lemma 7 to the situation where the $w_i$'s have $\pm 1$ initialization. This generalizes Theorem 6.

**Corollary 10.** *Suppose all the weights in a disjoint-Polynet are initialized randomly in $\pm 1$ and $k \ge 3$.*

*Let $T(\alpha) := \arg\sup_{t\ge 0}(\text{err}(w_{1:k}(t)) \ge \alpha)$. Then, for $\gamma \in (0, 1/2)$, if $\prod_i v_i(0) > 0$ (which happens w.p. 1/2),*

$$\frac{T(1/2 - \gamma)}{T(0)} = 1 - O((n')^{1-k/2} \cdot \gamma^{2/k-1}).$$

*Thus, even for $\gamma$ arbitrarily close to 0, when the input is sufficiently long, the network spends almost all of training with error above $1/2 - \gamma$.*

*Proof.* For $\pm 1$ initialization,

$$\frac{\|u_i\|_\infty}{\|u_i\|_2^3} = (n')^{-3/2}$$

so by Lemma 7,

$$\text{err}(w_{1:k}(t)) \ge \frac{1}{2} - \frac{1}{2}\prod_{i=1}^{k}\left(\text{erf}\left(\frac{|v_i(t)|}{\sqrt{2n'}}\right) + c(n')^{-3/2}\right)$$

$$\ge \frac{1}{2} - \frac{1}{2}\prod_{i=1}^{k}\left(\sqrt{\frac{2}{\pi n'}} \cdot |v_i(t)| + c(n')^{-3/2}\right)$$

for some $c > 0$.

By Lemma 6, $|v_i(t)| \le b$ for all $i$ whenever

$$t \le \frac{1}{k-2}\left(1 - b^{2-k}\right).$$

Setting

$$b = \sqrt{\pi n'/2}\left(\gamma^{1/k} - c(n')^{-3/2}\right),$$

we obtain that $T(1/2 - \gamma) = \frac{1}{k-2}(1 - O((n')^{1-k/2} \cdot \gamma^{2/k-1}))$.

Also by Lemma 6, using the language of that lemma statement, for all $i$

$$T_i(\infty) - T_i(b) \leq \frac{1}{k-2} \cdot b^{2-k} = \frac{1}{k-2} \cdot O((n')^{1-k/2} \cdot \gamma^{2/k-1}).$$

Once all the relevant weights have exploded to infinity, the error of the network will have zero error, so the result follows. $\qquad\square$

Now let us apply Lemma 6 and Lemma 7 to the situation where the $w_i$'s have standard normal initialization. Again we find that with high probability, the phase of learning with near-trivial accuracy is much longer than the subsequent period until perfect accuracy, as illustrated by the left-hand plot in Figure 5. This culminates in the full theorem statement regarding phase transitions in the loss:

**Theorem 6** (Loss plateau for gradient flow on disjoint-PolyNets; full statement). *Suppose all the weights in a disjoint-PolyNet are initialized $\sim N(0,1)$, and $k \geq 3$. Then, conditioned on $\prod_i v_i(0) > 0$, with probability $1 - 1/\text{poly}(n')$ over the randomness of the initialization, for $\gamma \in (0, 1/2)$,*

$$\frac{T(1/2 - \gamma)}{T(0)} = 1 - \tilde{O}((n')^{1-k/2} \cdot \gamma^{2/k-1}).$$

*where $T(\cdot)$ is defined as in Corollary 10.*

*Proof.* By a standard application of the generic Chernoff bound (Wainwright, 2019), for each $i \in [k], j \in [n'-1]$, we have

$$\Pr[|u_{i,j}| > \tau] \leq 2e^{-\tau^2/2} \quad \text{for all } \tau \geq 0.$$

Applying the union bound, we obtain

$$\Pr[\exists i, j \text{ s.t. } |u_{i,j}| > \tau] \leq 2n'ke^{-\tau^2/2} \quad \text{for all } \tau \geq 0.$$

This implies that w.p. $\geq 1 - \epsilon/2$,

$$\forall i : \quad \|u_i\|_\infty \leq \sqrt{2\log(4n'k/\epsilon)}. \tag{14}$$

For each $i$, $\|u_i\|_2^2$ follows a chi-squared distribution with $n'-1$ degrees of freedom. By Laurent and Massart (2000), for any $\tau \geq 0$,

$$\|u_i\|_2^2 \in [(n'-1) - 2\sqrt{(n'-1)\tau}, (n'-1) + 2\sqrt{(n'-1)\tau} + 2\tau] \quad \text{w.p. } \geq 1 - 2e^{-\tau}.$$

Hence, w.p. $\geq 1 - \epsilon/2$,

$$\forall i : \quad \|u_i\|_2 = \sqrt{n'} + O(\sqrt{\log(k/\epsilon)}) \tag{15}$$

With probability $1 - \epsilon$ both $\|u_i\|_\infty$ and $\|u_i\|_2$ are bounded as above, in which case we obtain that for $\epsilon = 1/\text{poly}(n', k)$, and for some $c_1, c_2 > 0$,

$$\frac{\|u_i\|_\infty}{\|u_i\|_2^3} \leq \frac{c_1\sqrt{\log(n'k)}}{\left(\sqrt{n'} + c_2\sqrt{\log(n'k)}\right)^3} \leq \tilde{O}\left((n')^{-3/2}\right).$$

Plugging this into Lemma 7 gives us

$$\text{err}(w_{1:k}(t)) = \frac{1}{2} - \frac{1}{2}\prod_{i=1}^k \left(\text{erf}\left(\frac{|v_i(t)|}{\|u_i\|_2\sqrt{2}}\right) + O\left(\frac{\|u_i\|_\infty}{\|u_i\|_2^3}\right)\right)$$

$$\geq \frac{1}{2} - \frac{1}{2}\prod_{i=1}^k \left(\tilde{O}\left(\frac{|v_i(t)|}{\sqrt{n'}}\right) + \tilde{O}\left((n')^{-3/2}\right)\right).$$

Thus, we can choose $b = \tilde{\Omega}(\gamma^{1/k}\sqrt{n'})$ such that if $v_i(t) \leq b$ for all $i$, then $\text{err}(w_{1:k}(t)) \geq \frac{1}{2} - \gamma$.

By Lemma 6, for all $i$,

$$T_i(b) \geq \frac{1}{k-2}\left(\bar{v}_a^{1-k/2} - (\bar{v}_a + b^2 - v_i(0)^2)^{1-k/2}\right) \geq \frac{1}{k-2}\left(\bar{v}_a^{1-k/2} - (b^2 - O(\log(k)))^{1-k/2}\right).$$

By the Chernoff bound, $\max_i |v_i(0)| \leq \sqrt{2\log(4k/\delta)}$ w.p. $\geq 1 - \delta/2$. And, by the same bound we used for $\|u_i\|_2^2$, we have that $\bar{v}_a = 1 + O(\sqrt{\log(1/\delta)/k})$ w.p. $1 - \delta/2$. Thus, for $\delta = 1/\text{poly}(k)$, we have that with probability $1 - \delta$,

$$T_i(b) \geq \frac{1}{k-2}\left(\tilde{\Omega}(1) - (b^2 - O(\log(k)))^{1-k/2}\right) = \frac{1}{k-2} \cdot \tilde{\Omega}(1). \tag{16}$$

Also by Lemma 6,

$$T_i(\infty) - T_i(b) \leq \frac{1}{k-2}(\bar{v}_g + b^2 - v_i(0)^2)^{1-k/2} \leq \frac{1}{k-2}(b^2 - v_i(0)^2)^{1-k/2}.$$

With probability $1 - 1/\text{poly(k)}$,

$$T_i(\infty) - T_i(b) \leq \frac{1}{k-2}(b^2 - O(\log(k)))^{1-k/2} = \frac{1}{k-2} \cdot \tilde{O}\left((n')^{1-k/2} \cdot \gamma^{2/k-1}\right). \tag{17}$$

Combining Equation 16 and Equation 17, we obtain the desired statement.

$\square$

## B.4 Global convergence and phase transition for gradient flow on disjoint-PolyNets

In this section we will analyze the training of a disjoint-PolyNet using SGD with online (i.i.d.) batches. We will show a convergence result for the 0-1 error of the learned classifier.

**Theorem 7** (SGD on disjoint-PolyNets learns disjoint parities; full statement)**.** *Assume we randomly initialize the disjoint-PolyNet with weights drawn uniformly from $\{\pm 1\}$. Fix $\epsilon \in (0, 1/2)$ and run SGD at any batch size $B \geq 1$ for $T \geq 6\log(2nT/\delta)\log(2k/\epsilon)(3n' - 2)^{2k-1}$ iterations. There exists an adaptive learning rate schedule, such that, with probability 1/2 over the randomness of the initialization and $1 - \delta$ over the sampling of SGD, the following holds:*

$$\text{err}\left(w_{1:k}^{(T+1)}\right) \leq \epsilon.$$

*Proof.* For simplicity of presentation, we will assume $B = 1$. Let the sample at iteration $t$ be $(x^{(t)}, y^{(t)})$ where $x^{(t)} \sim \text{Unif}(\{\pm 1\}^n)$ and $y^{(t)} = \chi_S^{(t)}(x^{(t)})$. Denote the population and stochastic gradient at time $t$ as:

$$\widehat{g}_i^{(t)} = -y^{(t)}\left(\prod_{j\neq i} w_j^{(t)^\top} x_j^{(t)}\right) x_i^{(t)}$$

$$g_i^{(t)} = -\left(\prod_{j\neq i} w_{j,1}^{(t)}\right) e_1$$

Observe that $|\widehat{g}_{i,j}^{(t)}|, |g_{i,j}^{(t)}| \leq \prod_{l\neq i} \|w_l^{(t)}\|_1$; note that this is also true when $B > 1$. Let the learning rate be $\eta_i^{(t)} = \frac{1}{2\sqrt{2T\log(2nT/\delta)}\cdot\prod_{l\neq i}\|w_l^{(t)}\|_1}$. Let $\Delta_{i,j}^{(t)} = \eta_i^{(t)}(g_{i,j}^{(t)} - \widehat{g}_{i,j}^{(t)})$ and $s_{i,j}^{(t)} = \sum_{\tau=1}^{t} \Delta_{i,j}^{(\tau)}$. Observe that $\mathbb{E}\,\Delta_{i,j}^{(t)} = 0$ and therefore $s_{i,j}^{(1)}, \ldots, s_{i,j}^{(t)}$ form a martingale. Furthermore, observe that

$$|s_{i,j}^{(t)} - s_{i,j}^{(t-1)}| = |\Delta_{i,j}^{(t)}| \leq \frac{|g_{i,j}^{(t)}| + |\widehat{g}_{i,j}^{(t)}|}{\sqrt{T\log(2nT/\delta)} \cdot \prod_{l\neq i}\|w_l^{(t)}\|_1} \leq \frac{1}{\sqrt{2T\log(2nT/\delta)}}$$

Then, for every $t < T$ and $i, j$, by the Azuma-Hoeffding inequality, with probability $1 - \frac{\delta}{nT}$, $|s_{i,j}^{(t)}| \leq 1/2$. By the union bound, w.p. at least $1 - \delta$, for every $t < T$ and all $i, j$ it holds that $|s_{i,j}^{(t)}| \leq 1/2$. Let us assume this holds.

**Claim 11.** *For all $t \leq T + 1$, for $j > 1$, $|w_{i,j}^{(t)}| \leq 3/2$.*

*Proof.* Note that $w_{i,j}^{(t+1)} = w_{i,j}^{(1)} + s_{i,j}^{(t)}$, therefore, we have,

$$|w_{i,j}^{(t+1)}| \leq |w_{i,j}^{(1)}| + |s_{i,j}^{(t)}| \leq 3/2.$$

since $|w_{i,j}^{(1)}| = 1$ and $|s_{i,1}^{(t-1)}| \leq 1/2$. $\qquad\square$

Denote $\xi_i = \text{sign}\left(w_{i,1}^{(1)}\right)$ and assume that $\prod_{j=1}^{k} \xi_j > 0$, and note that this happens w.p. $1/2$. We will assume this holds for the next lemma.

**Claim 12.** *For all $t \leq T + 1$ and $i \in [k]$ it holds that*

$$\xi_i w_{i,1}^{(t)} = |w_{i,1}^{(t)}| \geq \frac{1}{2} + \frac{1}{4\sqrt{2}} \left(\frac{1}{3n' - 2}\right)^{k-1} \sqrt{\frac{t-1}{\log(nT/\delta)}}.$$

*Proof.* Observe that the claim holds for $t = 1$ since $|w_{i,\cdot}^{(1)}| = 1$. By induction on $t$, assume the claim holds for all $\tau \leq t$. Now we will prove it holds for $t + 1$.

Observe that for all $\tau \leq t$, by the assumption:

$$g_{i,1}^{(\tau)} = -\prod_{j \neq i} w_{j,1}^{(\tau)} = -\xi_i \prod_{j \neq i} \xi_j w_{j,1}^{(\tau)} = -\xi_i \prod_{j \neq i} |w_{j,1}^{(\tau)}|.$$

Note that $w_{i,1}^{(t+1)} = w_{i,1}^{(1)} + \sum_{\tau=1}^{t} \eta_i^{(\tau)} \left(\prod_{j \neq i} w_{j,1}^{(\tau)}\right) + s_{i,1}^{(t)}$, then we have

$$\xi_i w_{i,1}^{(t+1)} = \xi_i w_{i,1}^{(1)} + \xi_i \sum_{\tau=1}^{t} \eta_i^{(\tau)} \left(\prod_{j \neq i} w_{j,1}^{(\tau)}\right) + \xi_i s_{i,1}^{(t)}$$

$$= 1 + \sum_{\tau=1}^{t} \left(\frac{1}{4\sqrt{2\tau \log(nT/\delta)}} \cdot \prod_{j \neq i} \frac{\xi_j w_{j,1}^{(\tau)}}{\|w_j^{(\tau)}\|_1}\right) + \xi_i s_{i,1}^{(t)} \qquad (18)$$

$$\geq \frac{1}{2} + \frac{1}{2\sqrt{2T \log(2nT/\delta)}} \cdot \sum_{\tau=1}^{t} \left(\prod_{j \neq i} \frac{|w_{j,1}^{(\tau)}|}{\|w_j^{(\tau)}\|_1}\right) - |s_{i,1}^{(t)}| \qquad (19)$$

$$\geq \frac{1}{2} + \frac{1}{2\sqrt{2T \log(2nT/\delta)}} \cdot \sum_{\tau=1}^{t} \left(\prod_{j \neq i} \frac{|w_{j,1}^{(\tau)}|}{|w_{j,1}^{(\tau)}| + \frac{3(n'-1)}{2}}\right) \qquad (20)$$

$$\geq \frac{1}{2} + \frac{1}{2\sqrt{2T \log(2nT/\delta)}} \cdot \sum_{\tau=1}^{t} \left(\prod_{j \neq i} \frac{1}{1 + 3(n'-1)}\right) \qquad (21)$$

$$\geq \frac{1}{2} \left(1 + \left(\frac{1}{3n'-2}\right)^{k-1} \frac{t}{\sqrt{2T \log(2nT/\delta)}}\right).$$

(18) follows from observing that $\xi_i w_{i,1}^{(1)} = |w_{i,1}^{(1)}| = 1$ and $\prod_{j=1}^{k} \xi_j = 1$. (19) follows from the inductive hypothesis $\xi_j w_{j,1}^{(\tau)} = |\xi_j w_{j,1}^{(\tau)}|$. (20) follows from our assumption that $|s_{i,1}^{(t)}| \leq 1/2$ and Claim 11. (21) follows from the inductive hypothesis $|w_{i,1}^{(\tau)}| \geq 1/2$. $\qquad\square$

Setting $T$ such that $T \geq 2 \log(2nT/\delta)\alpha^2(3n' - 2)^{2k-2}$ for $\alpha = 3\sqrt{(n'-1)\log(2k/\epsilon)} - 1$, from the above claims, after iteration $T$, we have

$$w_{i,1}^{(T+1)} \geq \frac{\alpha + 1}{2}$$

$$w_{i,j}^{(T+1)} \leq 3/2 \text{ for } j > 1.$$

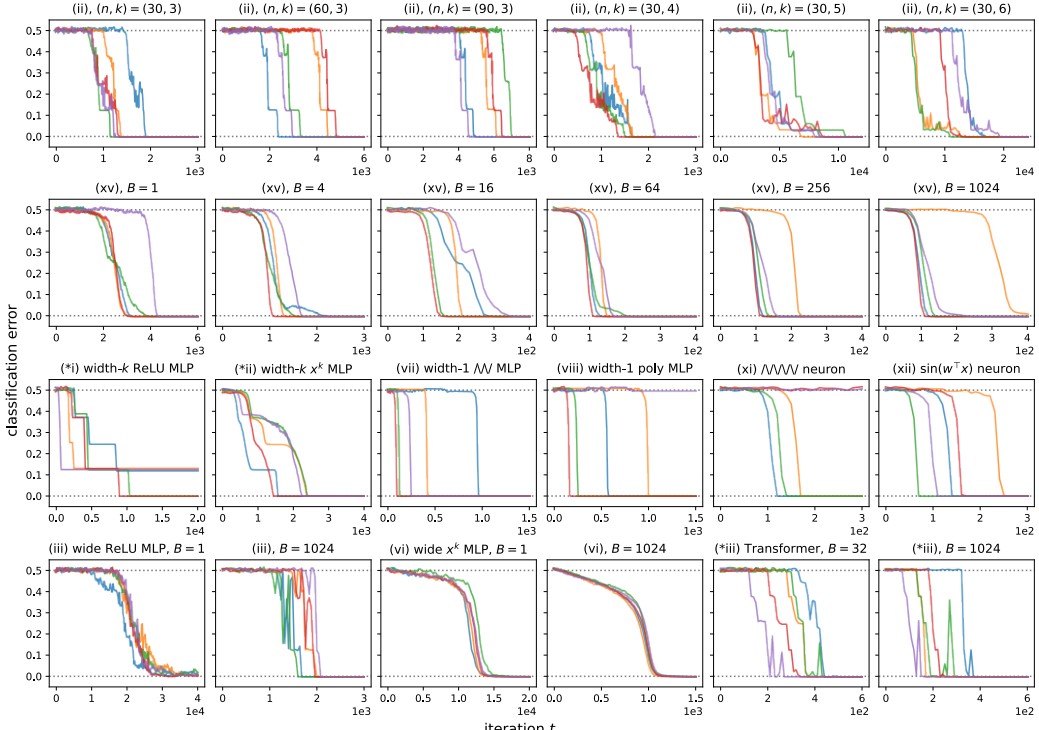

Figure 6: Additional training curves (on the $(50, 3)$-parity problem, except the first row); full details are in Appendix D.1.6. *1st row:* The same architecture, initialization, and training algorithm (a width-100 ReLU MLP in this case), without an explicit sparse prior, adapts to the computational difficulty parameters $n, k$. *2nd row:* Our positive empirical results hold over a wide range of batch sizes $B$, all the way down to $B = 1$. Training is unstable (more outliers) at very large and very small batch sizes. *3rd row:* Even the least overparameterized neural networks, which are barely wide enough to represent parity, converge with reasonable probability (sometimes failing to reach a global minimum). *4th row:* Larger models (width-1000 MLPs and $d_{\mathrm{emb}} = 1024$ Transformers) are robust to a wide range of batch sizes. Note the lack of plateaus in setting (vi), which is revisited in Appendix C.8.

Using Lemma 7, we have with probability $1 - \delta$,

$$\mathrm{err}(w_{1:k}^{(T+1)}) \leq 2k \exp\left( -\frac{(\alpha + 1)^2}{9(n' - 1)} \right) = \epsilon.$$

$\square$

## C   Additional figures, experiments, and discussion

This section contains our unabridged empirical results, visualizations, and accompanying discussion. Additional example training curves (like the assortment in Figure 1 *(left)*) are shown in Figure 6; more examples can be found in the subsections below.

**Convergence times, success probabilities, and scaling laws.** We first present the full empirical results outlined in Section 3 of the main paper. Figure 7 shows convergence times $t_c$ on small parity instances for all of the architecture configurations enumerated in Section 3.1. In some of these settings, $t_c$ exhibits high variance due to unlucky initializations (see Figure 8); thus, we report $10^{\mathrm{th}}$ percentile convergence times. Figure 9 gives coarse-grained estimates for how $t_c$ scales with $(n, k)$, based on small examples. For selected architectures, Figure 10 shows how these convergence times scale with $n$ and $k$ more precisely: for small $n$, power law relationships $t_c \propto n^{\alpha \cdot k}$ (for small constants $\alpha$) are observed for all configurations. Note that for larger $n$, the exponent (i.e. the slope in

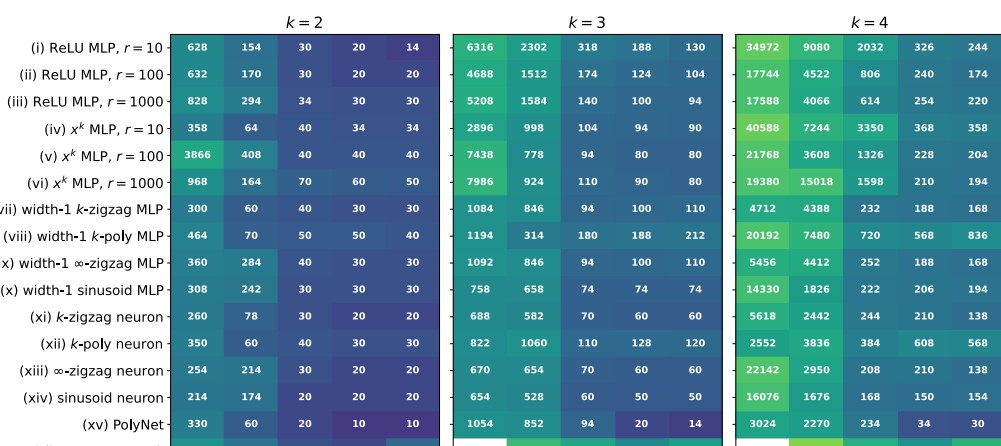

Figure 7: 10th percentile convergence times $t_c$ of SGD (with $n = 30$, hinge loss, uniform initialization, and best learning rate $\eta \in \{1, 10^{-1}, 10^{-2}, 10^{-3}\}$), for various architectures, parity degrees $k$, and batch sizes $B$. See Appendix D for full details.

the log-log plot) increases: with a constant learning rate and standard training, the $n^{\Theta(k)}$ does not continue indefinitely. All additional details are in Appendix D.

**Guide to this section.** The remainder of Appendix C expands on the various discussions and figures from Sections 4 and 5.

- Appendix C.1 gives experimental evidence that Fourier gaps are present at iterates $w_t$ and initializations $w_0$ other than sign vectors, as well as for activation functions other than ReLU. This suggests that the feature amplification mechanism is robust, and illuminates directions for strengthening the theoretical results.

- Appendix C.2 discusses how the building blocks of deep learning (activation functions, biases, initializations, learning rates, and batch sizes) play multiple, sometimes conflicting roles in this setting.

- Appendix C.3 provides additional white-box visualizations of hidden progress from Figure 3.

- Appendix C.4 explores the implications of the feature amplification mechanism for scaling model size– namely, unlike random search, large width does not impart parallel speedups.

- Appendix C.5 shows that our results hold in the finite-sample setting (allowing for multiple passes over a training set of size $m$). In particular, we show that in low-data regimes, the models exhibit the *grokking* phenomenon.

- Appendix C.6 extends our results to *noisy* parities (which comprise the true "emblematic computationally-hard problem").

- Appendix C.7 introduces a counterexample for "layer-by-layer learning", using parity distributions whose degrees are higher than those of the individual layers' polynomial activations. Preliminary experiments show that standard training works in this setting.

- Appendix C.8 presents examples of training curves for wide polynomial-activation MLPs, where, unlike the other settings, there is no initial plateau in the model's error.

## C.1 Fourier gaps at initialization and SGD iterates

Proposition 9 shows that if the function $x \mapsto \sigma'(w_0^\top x)$ has a Fourier gap at $S$, then $S$ can be identified from a batch gradient at initialization $w_0$ with $B = O(1/\gamma^2)$ samples. Our end-to-end

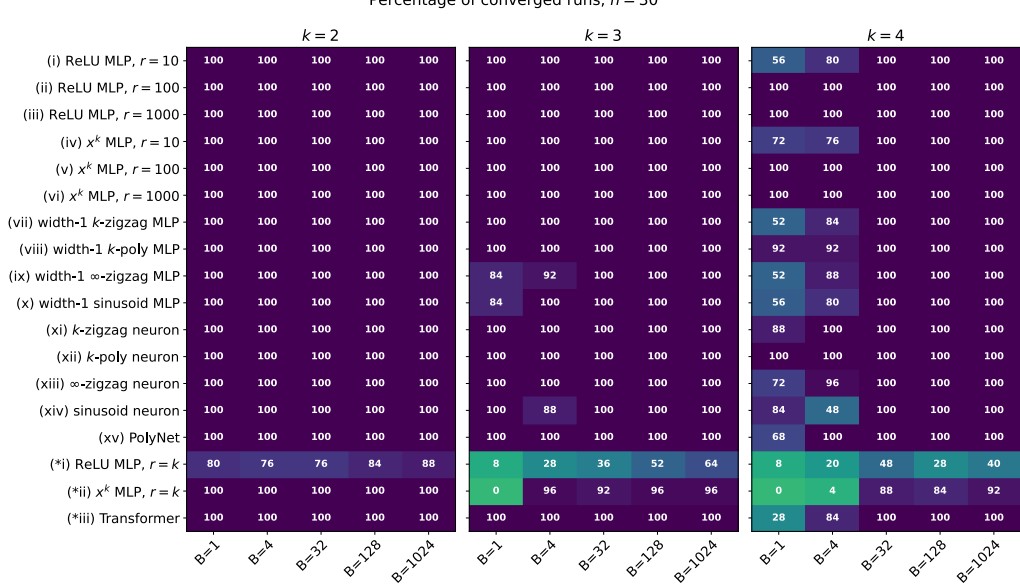

Figure 8: Percentage of converged runs ($t_c < 10^5$) out of 25 random trials, with the same architectures and training parameters as Figure 7. With sufficiently large batch sizes, training is extremely robust in settings (i) through (xv).

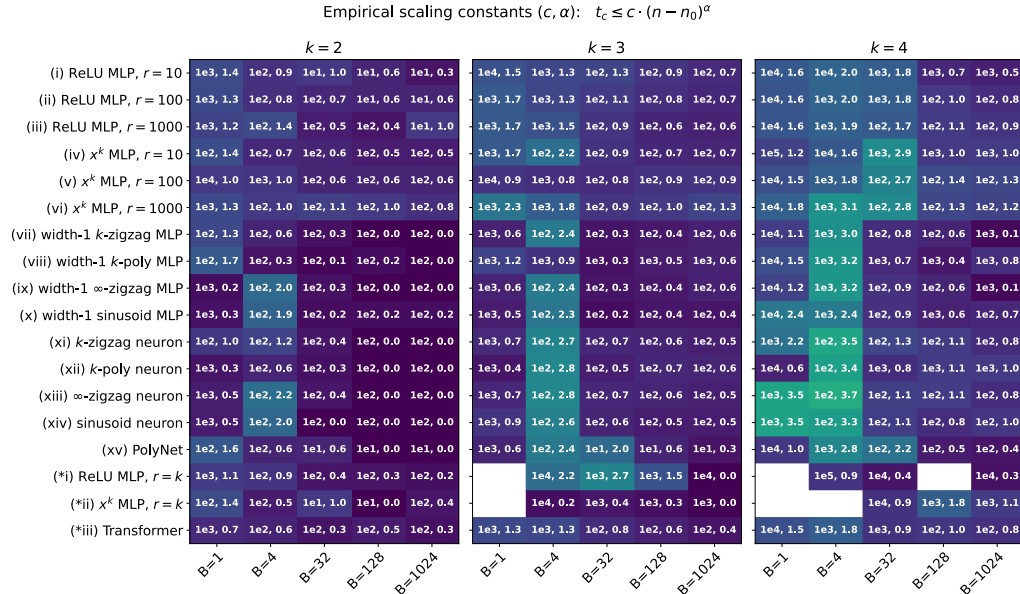

Figure 9: Coarse scaling estimates for the 10th percentile convergence time of SGD (with hinge loss, uniform initialization, and best learning rate) on every architecture: $c, \alpha$ such that $t_c \leq c \cdot (n - n_0)^\alpha$ on small parity instances $n \in \{10, 20, 30\}, k \in \{2, 3, 4\}$. Missing entries denote cases where $< 10\%$ of trials were convergent for any $n$ (see Figure 8). Boxes are colored according to $\alpha$. See Appendix D for full details.

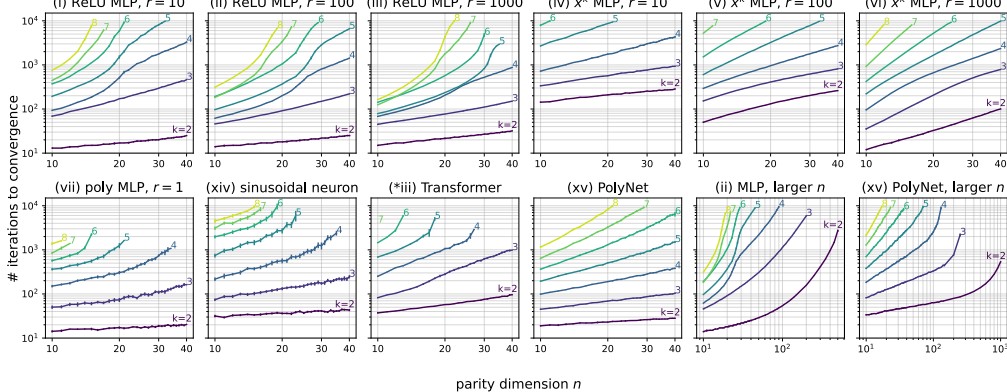

Figure 10: Finer-grained plots of convergence times for selected architecture configurations, for $n \in \{10, 11, \ldots, 39, 40\}$ and $k \in \{2, 3, \ldots, 8\}$. Medians over $1000$ runs are shown, with $95\%$ bootstrap confidence intervals. *Top row:* standard MLP configurations (i) through (vi). *Bottom row:* Miscellaneous settings (1-neuron networks; Transformers; larger $n$). For details on each setting, see Appendix D.1.7.

result (Theorem 4) requires ReLU activations and sign vector initialization, because the Fourier gap condition (Definition 1) arises from exact formulas for the Fourier coefficients of the majority function. Stronger end-to-end theoretical guarantees would follow from analogous Fourier gaps in more general population gradients. This requires $x \mapsto \sigma'(w^\top x)$ to satisfy these conditions simultaneously:

- *Fourier concentration:* upper bounds on the degree-$(k+1)$ coefficients $\widehat{f}(S \cup \{i\})$, for $i \notin S$. The term is borrowed from Klivans et al. (2004), who use upper bounds on Fourier coefficients of LTFs to approximate them (thus, learn halfspaces) with low-degree polynomials.

- *Fourier anti-concentration:* lower bounds on the degree-$(k-1)$ coefficients $\widehat{f}(S \setminus \{i\})$, for $i \in S$.

A natural question is: *which Boolean functions, other than majority, satisfy the $\gamma$-Fourier gap property at $S$, for $\gamma \geq n^{-\Omega(k)}$?*

We present some numerical evidence for large Fourier gaps in functions $x \mapsto \sigma'(w^\top x)$ other than majority, which arise from gradients of architectures other than ReLU MLPs with sign initialization. This shows that the mechanism of feature emergence is empirically robust in settings not fully explained by our current theory. Establishing corresponding theoretical guarantees would enable stronger end-to-end global convergence guarantees for MLPs and other architectures.

In these experiments, population gradients were computed by brute force integration over all $2^n$ Boolean inputs $x \in \{\pm 1\}^n$. In all cases, for various choices of $\sigma, w$, we measure a slightly relaxed notion of Fourier gap $\Gamma$ in the population gradient:

$$\Gamma_{\sigma,k}(w) := \max_{i \in [k]} |g_i| - \max_{i \notin [k]} |g_i|, \qquad g := \mathbb{E}[yx\, \sigma'(w^\top x)].$$

If $\Gamma > 0$, then *one*[15] coordinate from the parity can be identified from $O\left(\frac{\log n}{\Gamma^2}\right)$ samples of the gradient at $w$.

**Random LTFs.** For ReLU activations and symmetric Bernoulli (i.e. random sign) initialization $w_i \sim \mathrm{Unif}(\{\pm c\}^n)$, the Fourier coefficients are the same as those of majority; thus, there is a Fourier gap of $\gamma \geq n^{-\Omega(k)}$ at every set $S$ (and the same is true of $\Gamma$). We probe the Fourier gaps of linear threshold functions (LTFs) under other ubiquitous initializations: i.i.d. uniform and Gaussian. These are shown in Figure 11, which indicates (at least for small $n, k$) that the Fourier gap is comparable to that of majority with non-negligible probability.

---

[15]Replacing the first $\max$ in the definition of $\Gamma$ with $\min$ would give us the same notion of Fourier gap as Definition 1: if *all* the relevant coordinates are larger than *all* of the irrelevant ones, estimating the population gradient allows us to recover the relevant coordinates.

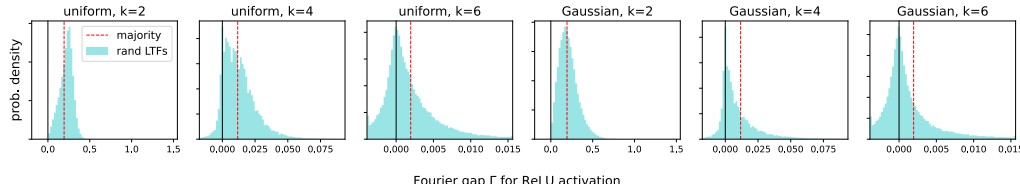

Figure 11: Distributions of exact Fourier gaps $\Gamma_{\sigma,k}(w)$ when $\sigma = \mathrm{ReLU}$, and $w$ is an i.i.d. {uniform, Gaussian} random vector. These are derived from the Fourier coefficients of the corresponding random LTFs, computed here in exponential time for $n = 15$.

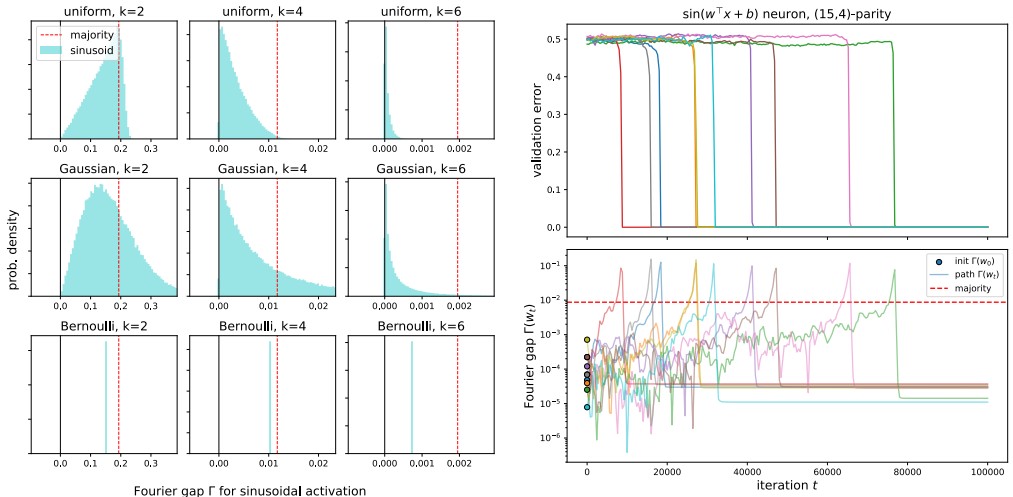

Figure 12: Numerical evidence for Fourier gaps beyond random LTFs. *Left:* Fourier gaps of exact population gradients $\mathbb{E}_{\mathcal{D}_S}[yx_i \sin(w^\top x/\sqrt{n})]$ induced by sinusoidal activations, for random initializations $w$. In these small cases, they are comparable the the Fourier gaps of majority (red dashed line). *Right:* Fourier gaps of a sinusoidal neuron's population gradients along the SGD optimization path (10 trials shown). The Fourier gap is consistently positive at initialization, but somewhat smaller than that of majority (red dashed line). Interestingly, it amplifies through the course of training.

**Random non-LTFs.** The successful convergence of architectures with smoother activations (in the parity setting and beyond) motivates the question of whether large Fourier gaps are present in population gradients corresponding to functions other than LTFs. Figure 12 *(left)* shows that this is the case for sinusoidal activations.

**Boolean functions along the SGD path.** Finally, to further close the gap between Theorem 4 and the empirical results, it is necessary to address the fact that SGD accumulates gradients with respect to *time-varying* iterates, while our analysis approximates this using a large-batch gradient at a static iterate $w_0$. In fact, SGD seems to help in some cases: Figure 12 *(right)* shows that when training a sinusoidal neuron, SGD amplifies the initial Fourier gap.

## C.2 Counterintuitive roles of the building blocks of deep learning

Even in this simple problem setting, the simultaneous computational and statistical considerations lead to counterintuitive consequences for the optimal configurations of architectures and algorithms for this setting. We encountered the following, in the search for architecture configurations for the empirical study:

• *Activation functions.* This mechanism of features emerging via Fourier gaps (see Definition 1) is strongest with non-smooth activations such as the ReLU, whose derivatives are discontinuous threshold functions. This is an orthogonal consideration to representational capacity and mitigation

of local minima (under which one might conclude that degree-$k$ polynomial activations are optimal). In summary, in feature learning settings where the Fourier gaps and low-complexity solutions are simultaneous relevant, there is a *sharpness-smoothness tradeoff* for the activation function.

- *Biases.* The symmetry of the majority function (as well as all unbiased LTFs) causes its even-degree Fourier coefficients to be zero; thus, certain variants of the setups in Section 3 fail for odd $k$. Bias terms (trainable or fixed) are necessary to break this symmetry, in theory and practice. Simultaneously, biases serve the more conventional role of shifting the loss surface; see Section D.1.2 for how this affects the details of how the biases were chosen in the experiments.

- *Initializations.* The role of the initialization distribution is similarly twofold in this setting: $w_0$ should be close to the desired solution $w^*$, but it must also be selected such that SGD will successfully amplify the Fourier gap. A third consideration, which we do not attempt to study in this work, is that multiple randomly-initialized neurons will tend to learn the correct features at different times (see the weight trajectory visualizations in Figure 3 and Figure 13, as well the staircase-like training curves seen for MLPs in Figure 1 *(left)*). We expect this *symmetry breaking* phenomenon to be present in more complex feature learning settings. Finally, as shown in the training curves from setting (vi) in Figure 6, and in more detail in Appendix C.8, the choice of activation function influences the qualitative behavior of the training curves: namely, whether the plateaus disappear at large widths and batch sizes.

- *Batch sizes and learning rates.* The empirical and theoretical results both suggest that SGD uses independent samples to gradually amplify a signal containing the correct features in the initial population gradient of the correlation $\left( \nabla_w \mathbb{E}[-y x_i \sigma'(w^\top x)] \right)|_{w=w_0}$. However, it would be truer to this mechanism to stay at the initialization $w_0$ until the algorithm has accumulated enough data to discern the correct indices (equivalently, scale up the batch size); in contrast, standard training takes gradients with respect to $w_t$ along the SGD path. We hypothesize that the bias incurred by this drifting $w_t$ (and thus drifting population gradient) accounts for the degradations seen in Figure 2 *(right)* and Figure 10. However, Figure 12 *(right)* shows that the movement of SGD can be helpful, amplifying the Fourier gap.

## C.3 Hidden progress measures

In this section, we provide an expanded discussion and plots for the investigations outlined in Figure 3 and the *"hidden progress measures"* section in Section 5.

For a neural network training pipeline which outputs a sequence of iterates $\theta_0, \ldots, \theta_T \in \Theta$, we define a *progress measure* $\rho : \Theta \to \mathbb{R}$ to be any function of the training algorithm's state[16] which is predictive of the time to convergence (i.e. conditioned on $\theta_t$, the random variables $\rho$ and $t_c - t$ are not independent). By this definition, the only algorithms which have no progress measures are those whose convergence times $t_c$ are *memoryless* (independent of $\theta_t$).

Note that many trivial progress measures exist: an example to keep in mind is that for the algorithm which exhaustively enumerates over a deterministic list of hypotheses (say, the possible $k$-element subsets $S$ in lexicographical order) and terminates when it finds the correct one, the current iteration $t$ is a progress measure. Thus, the purpose of demonstrating hidden progress measures $\rho$ is *not* to provide further evidence that SGD finds the features using Fourier gaps. Rather, it is to (1) further refute the hypothesis of SGD performing a memoryless Langevin-like random search, and (2) provide a preliminary exploration of how progress can be quantified even when the natural metrics of loss and accuracy appear to be flat.

**Fourier gaps over time.**    The Fourier gap visualizations in Section C.1 already provides an example of a quantity which varies continuously as the model trains, despite no apparent progress in the loss and accuracy curves. However, none of our theoretical analyses capture the empirical observation that this quantity tends to amplify over time. Below, we consider other quantities which reveal hidden progress in parity learning, which are more straightforward and closer to our analyses.

**Weight movement.**    The most direct observation of hidden progress simply comes from the movement of the neurons' weights at the relevant indices: that is, for a single neuron's weights $w_t \in \mathbb{R}^n$,

---

[16]In addition to the model's parameters, the full state of the training procedure should also include the auxiliary variables defined by the optimization algorithm; two ubiquitous ones in deep learning are the momentum vector and the adaptive preconditioner. Here, we only consider vanilla SGD, which maintains no auxiliary variables.

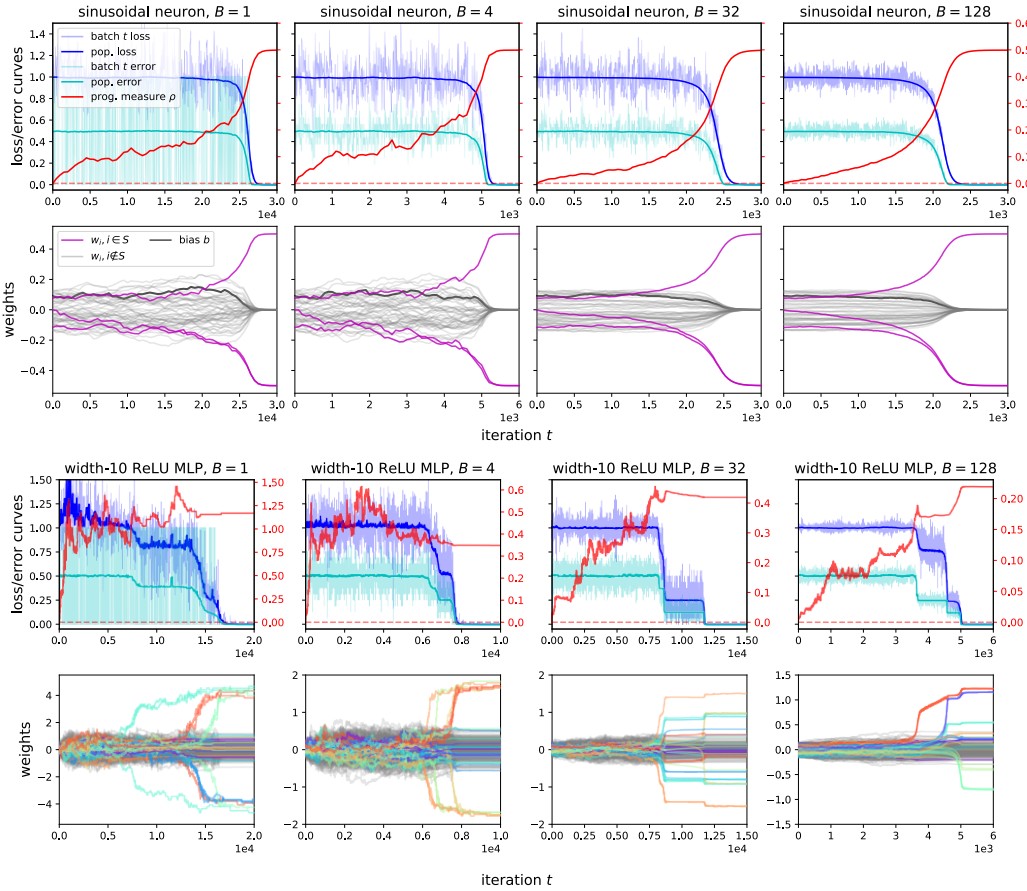

Figure 13: Supplementary plots for visualizations of the optimization trajectory $w_t$, and the hidden progress measure $\rho(w_{0:t})$. In the MLP plots, the single scalar $\rho$ is the $\infty$-norm of the entire first layer $W$, and weights are color-coded by row (i.e. neuron). With small batch sizes $B \in \{1, 4\}$, per-iteration losses are averaged over a short window (lengths 16 and 4, respectively).

the quantity $\rho([w_t]_i) : i \in S$. In the main paper, Figure 3 *(left, center)* directly visualizes the evolution of the weights $w_t$ for a single sinusoidal neuron (with a bias, but no second layer). Figure 13 supplements these plots from the main paper with additional plots of weight trajectories, at different batch sizes $B$, as well as a width-10 MLP architecture. As seen in these plots, progress only becomes visible in the loss once the relevant weights become larger than all of the irrelevant weights.

$\ell_\infty$ **path length.** Finally, we present an example of a *single* measurement of the optimization path which captures hidden progress in this setting, which can be plotted alongside loss and accuracy curves. For any iterates of a neuron's weights $w_t \in \mathbb{R}^n$, we choose the $\infty$-norm of the movement from initialization: $\rho(w_{0:t}) := ||w_t - w_0||_\infty$. We present a brief intuitive sketch of the motivation for this choice of $\rho(\cdot)$, and some additional visualizations.

From the theoretical analysis, under the approximation $\nabla\ell(w_t) \approx \nabla\ell(w_0)$ (so that feature learning is performed by estimating the initial population gradient to high precision), we can think of the $i$-th coordinate of $w_t$ as a biased random walk with constant variance $\sigma^2$; the Fourier gap condition entails that biases $\beta_i$ of these random walks are large when $i \in S$. Then, this choice of $\rho$ is an estimate for the drift term $t \cdot \max_i |\beta_i|$, which is larger than the $\sigma\sqrt{t}$ contribution of the variance for sufficiently large $t$.

This progress measure is shown alongside the loss curves in Figure 13, in red. We do not attempt to characterize the dynamics of $\rho$; we only note that they are clearly distinguishable from the maximum of $n$ unbiased random walks, even when SGD appears to make no progress in terms of loss and

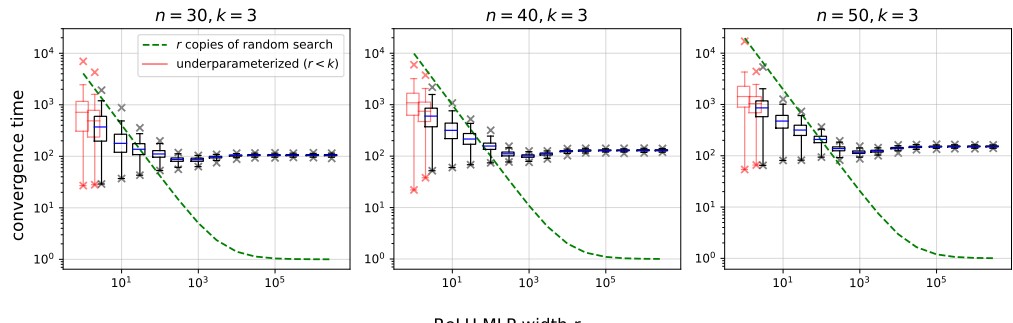

Figure 14: Number of iterations for MLPs (with standard initialization and training) to converge on sparse parity problems, in terms of width $r$. Boxes denote interquartile ranges over 1000 runs; whiskers denote $\pm 1.5 \cdot$IQR; $\times$ markers denote minimum and maximum outliers. Underparameterized models ($r < k$) are shown in red, and considered "converged" at $55\%$ accuracy. Scaling $r$ this way does not lead to $r \times$ parallel speedups, like the expected success time for $r$ copies of random search (shown in green for comparison).

accuracy. Studying hidden progress measures in deep learning more quantitatively, as well as in more general settings, presents a fruitful direction for future work.

## C.4   Convergence time vs. width

We provide supplementary plots for the experiment outlined in Figure 4 *(left)*, which probes whether extremely large widths ($r \gg n^k$) afford factor-$r$ parallel speedups of the parity learning mechanism (as one would expect from random search). On 3 parity instances $n \in \{30, 40, 50\}, k = 3$, we varied the width $r \in \{1, 2, 3, \dots, 9, 10, 30, 100, 300, \dots, 10^6, 3 \times 10^6\}$, keeping all other parameters the same ($B = 128, \eta = 0.1$).

**Results.**   We did not find evidence of such parallel speedups over 1000 runs in each setting; see Figure 14. This serves as further evidence that the mechanism by which standard training solves parity is best understood as deterministic and sequential, rather than behaving like random search over size-$k$ subsets. A benefit of width appears to be *variance reduction*: the upper tail of long convergence times is mitigated by a large number of randomly-initialized neurons.

## C.5   Learning and grokking in the finite-sample case

We provide some supplementary plots for the experiments outlined in Figure 4 *(right)*. In these settings, a fixed architecture (width-100 MLP with ReLU activations) is trained with minibatch SGD in an otherwise fixed configuration (hinge loss, learning rate $\eta = 0.1$, batches of size $B = 32$) on a finite training sample of size $m$. We also vary a *weight decay* parameter $\lambda$.

As shown in Figure 15, the weight decay parameter $\lambda$ modulates a delicate computational-statistical tradeoff: it improves generalization (expanding the range of $m$ for which training eventually finds the correct solution), but the model fails to train at large values of $\lambda$. For small $m$ and appropriately tuned $\lambda$, we observe *grokking*: the model initially overfits the training data, but finds a classifier that generalizes after a large number of iterations.

## C.6   Learning noisy parities

The other empirical results in this work focus on noiseless parity distributions $\mathcal{D}_S$, to reduce the number of sources of variance and degrees of freedom. However, the setting of *random classification noise* is important for several reasons. In this section, we briefly demonstrate that our results extend to this case. Let $\mathcal{D}_S^{(\epsilon)}$ denote the $(n, k, \epsilon)$-*noisy parity distribution*, defined by flipping the labels in the $(n, k)$-parity distribution $\mathcal{D}_S$ independently with probability $\frac{1}{2} - \epsilon$. Note that when $\epsilon = 0$, the labels

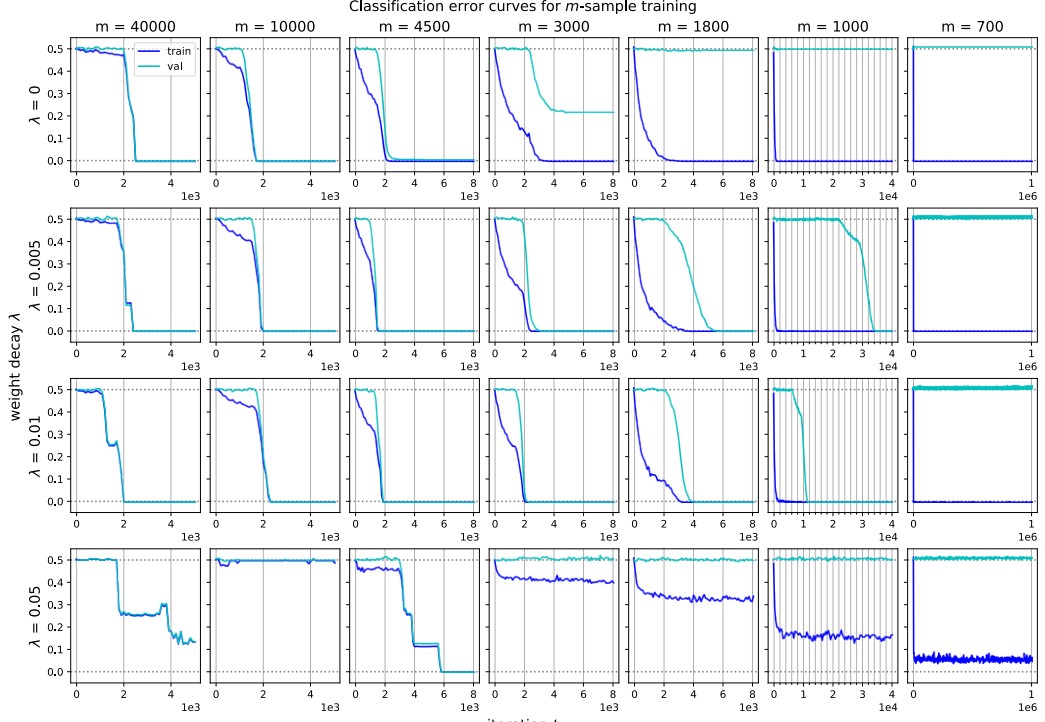

Figure 15: Supplementary plots for the finite-sample setting. The same configuration (width-100 ReLU MLP, $B = 32, \eta = 0.1$) varying the sample size $m$ (decreasing from left to right) and weight decay $\lambda$ (increasing from top to bottom). When $m$ is sufficiently large (much larger than the statistical threshold $\Theta(k \log n)$), generalization error is negligible. When $m$ is too small, the model fails to train. In between, we observe eventual convergence to the correct solution, with training curves exhibiting the *grokking* phenomenon. Weight decay governs a statistical-computational tradeoff in this setting: larger $\lambda$ improves generalization, but can cause optimization to fail (bottom row).

are completely random (thus, $S$ cannot be learned). By a standard PAC-learning argument, when $0 < \epsilon \leq \frac{1}{2}$, the statistical limit for identifying $S$ from i.i.d. samples from $\mathcal{D}_S$ scales as $\Theta\left(\frac{k \log n}{\epsilon^2}\right)$.

**Motivations.** First, learning parities from *noisy* samples is the true "emblematic computationally-hard distribution". Without noise, there is a non-SQ algorithm which avoids the exponential-in-$k$ computational barrier: Gaussian elimination can identify $S$ in $O(n^3)$ time and $\Theta(n)$ samples. Second, viewing parities as an idealized setting in which to understand training dynamics, resource scaling, and emergence in deep learning, it is important to see that this phenomenon is robust to label noise.

**Theory.** It is easy to incorporate label noise into the theoretical analysis, which works with correlations of the form $\mathbb{E}_{\mathcal{D}_S}[y\, f(x)]$; each coordinate of the population gradient of the correlation loss is a quantity of this form. In the noisy case, these quantities are replaced with

$$\mathbb{E}_{(x,y)\sim\mathcal{D}_S^{(\epsilon)}}[y\, f(x)] = \epsilon \cdot \mathbb{E}_{(x,y)\sim\mathcal{D}_S}[y\, f(x)].$$

In particular, when architecture's population gradient has a Fourier gap with parameter $\gamma$ in the noiseless case implies a Fourier gap with parameter $\epsilon \cdot \gamma$.

**Experiments.** We find that the experimental findings are robust to label noise, in the sense that models are able to obtain nontrivial (and sometimes 100%) accuracy; see Figure 16 for some training curves under various settings of $\epsilon$. This provides concrete evidence against the (already extremely dubious) hypothesis that neural networks, with standard initialization and training, learn noiseless parities by implicitly simulating an efficient algorithm such as Gaussian elimination. Note that with a

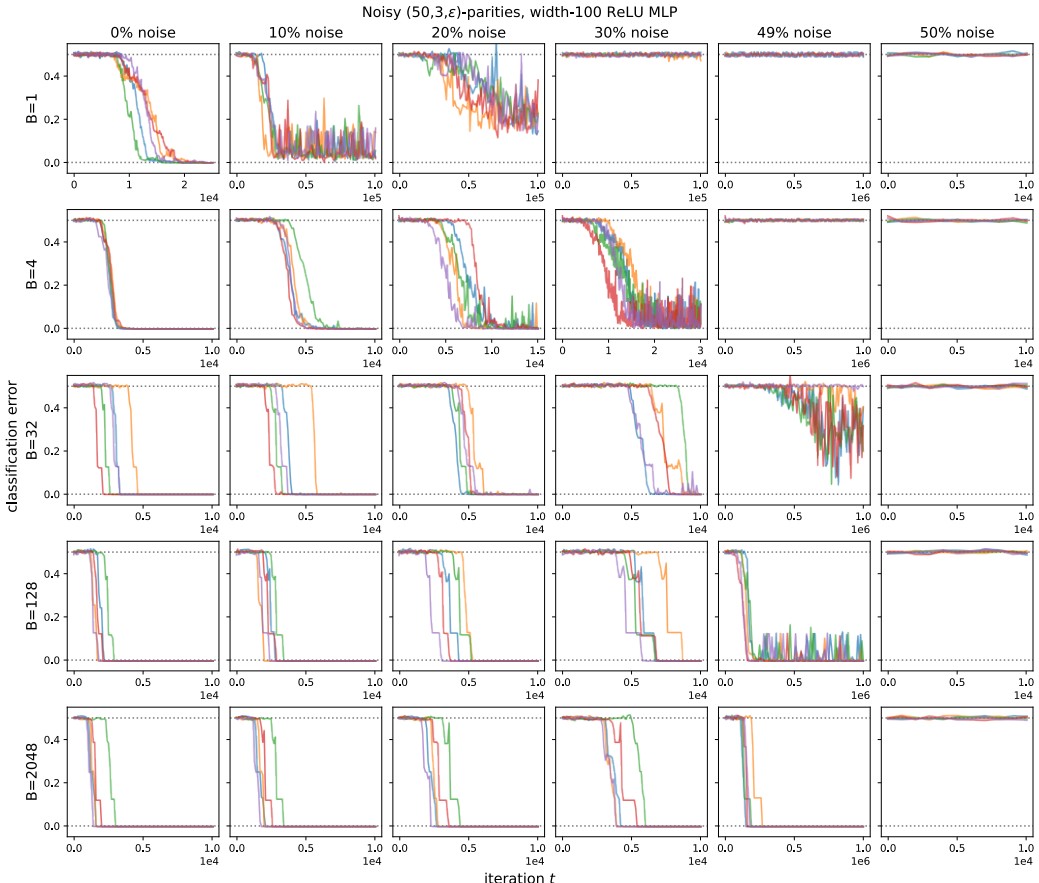

Figure 16: Training curves (over 5 random seeds) for a width-100 ReLU MLP on noisy $(50, 3, \epsilon)$-parity learning problems at various batch sizes $B$ and noise levels (flipping labels with probability $p$, so that $\epsilon = 1 - 2p$). The models learn the features and converge successfully (measured by accuracy on the noiseless distribution), even with $49\%$ of labels flipped randomly (i.e. $\epsilon = 0.01$). This is a preliminary illustration that the phenomena investigated in this paper are robust with respect to i.i.d. label noise. Note that the scale of $t$ is much larger for small batches and high noise.

constant learning rate (here, $\eta = 0.1$) and label noise, the iterates of SGD do not always converge to $100\%$ accurate solutions.

## C.7 Counterexample for layer-by-layer learning

**Notation.** Consider an $L$-layer MLP with activation $\sigma$, parameterized by weights and biases

$$\theta = (W_1, b_1, \ldots, W_{L-1}, b_{L-1}, u),$$

and defined by

$$f_{\mathsf{mlp}}(x; \theta) := (f_L \circ f_{L-1} \circ \cdots \circ f_2 \circ f_1)(x),$$

where $f_i$ denotes the function $z \mapsto \sigma(W_i z + b_i)$ for $1 \leq i \leq L - 1$, and $f_L$ denotes $z \mapsto u^\top z$. The shapes of the parameters $W_i, b_i, u$ are selected such that each function composition is well-defined. Let the *intermediate activations* at layer $i$ be denoted by

$$z_i(x; \theta) := (f_i \circ \ldots \circ f_1)(x).$$

Finally, $r_i$ (the *width at layer* $i$) refers to the dimensionality of $z_i$ as defined above.

**Construction where layer-by-layer learning is impossible.** Notice that when $\sigma$ is a degree-2 polynomial (say, $\sigma(z) = z^2$), an $L$-layer MLP can represent parities up to degree $2^{L-1}$– for example,

a 3-layer MLP (which composes quadratic activations twice) can represent a 4-sparse parity as a 2-sparse parity of 2-sparse parities. However, Equation (2) implies the following:

- An individual layer cannot represent a parity of $k > 2$ inputs.
- The population gradient (as in Equation (5) is zero (since every coordinate of the gradient is the correlation between a $k$-wise parity and a polynomial of degree 2).

Thus, this setting serves as an idealized counterexample for layer-by-layer learning: if SGD succeeds on parities with higher degree than the architecture's polynomial activations, it must do so by an end-to-end mechanism. Intuitively, earlier layers can only make progress by *knowing how their outputs will be used downstream*. Concretely, consider the population gradient of the correlation loss, with respect to a first-layer neuron's weights $w := (W_1)_{j,:}$. With layer-by-layer training, this gradient contains no information:

$$\nabla_w \mathbb{E}[-y x_i \underbrace{\sigma'(w^\top x)}_{\text{degree 1}} u_j] = 0.$$

However, in end-to-end training, the presence of downstream layers removes this barrier:

$$\nabla_w \mathbb{E}\left[ -y x_i \underbrace{\sigma'(w^\top x)}_{\text{degree 1}} \underbrace{\frac{\partial f_L \circ \ldots \circ f_2}{\partial (z_1)_j}}_{\text{degree } 2^{L-2}-1} \right],$$

giving the gradient greater representation capacity (in terms of polynomial degree). The question remains of whether end-to-end training works in this setting, which we resolve positively in small experiments.

**Results: end-to-end training works empirically.** We empirically observed successful training (to $100\%$ accuracy) in a few settings (with SGD, learning rate $\eta = 0.01$, batch size $B = 32$, and default uniform initialization as described in Appendix D.1):

- $L = 3, n \in \{10, 20, 30\}, k \in \{1, 2, 3, 4\}$. Small widths suffice: $(r_1, r_2) = (2, 1)$. Over 10 random seeds, all models converged within 20000 iterations.
- $L = 4, n \in \{10, 20, 30\}, k \in \{1, 2, 3, 4, 5, 6\}$. Widths were chosen to be slightly larger for stability: $(r_1, r_2, r_3) = (10, 10, 1)$. Over 10 random seeds, all models converged within 50000 iterations. Additionally, models trained on $(n, k) \in \{(10, 7), (20, 7), (30, 7), (10, 8)\}$ converged within 500000 iterations.

As a sanity check, the models failed to converge in experimental setups where $k > 2^{L-1}$: ($L = 2, k \geq 3$) and ($L = 3, k \geq 5$).

**Discussion.** This construction serves as a simple counterexample to the *"deep only works if shallow is good"* principle of Malach and Shalev-Shwartz (2019), demonstrating a case where a deep network can get near-perfect accuracy even when greedy layerwise training (e.g. (Belilovsky et al., 2019)) cannot beat trivial performance. It remains to characterize these positive empirical results theoretically, as well as to investigate whether there are pertinent analogues in real data distributions.

### C.8 Lack of plateaus for wide polynomial-activation MLPs

An interesting qualitative observation from the training curves in Figure 6 is that the validation accuracy curves in setting (vi) (width-1000 polynomial-activation MLPs) do not follow the same "plateau" or "staircase" pattern as the others. Figure 17 shows a few additional examples of training curves for polynomial-activation MLPs, varying the width $r$ and batch size $B$. We find that the rate of descent of the validation error increases with both of these parameters; note that this does not occur with ReLU activations (where there are sharp phase transitions between plateaus at all batch sizes).

This constitutes an exception to this paper's theme of "hidden progress" behind flat loss (or error) curves: with enough overparameterization *and* "over-sampling", the continuous progress of SGD in this setting is no longer hidden, and manifests in the training curves. This phenomenon seems to be specific to certain activation functions (i.e. $x^k$ but not ReLU); we leave it for future work to understand why and when it occurs, as well as potential practical implications.

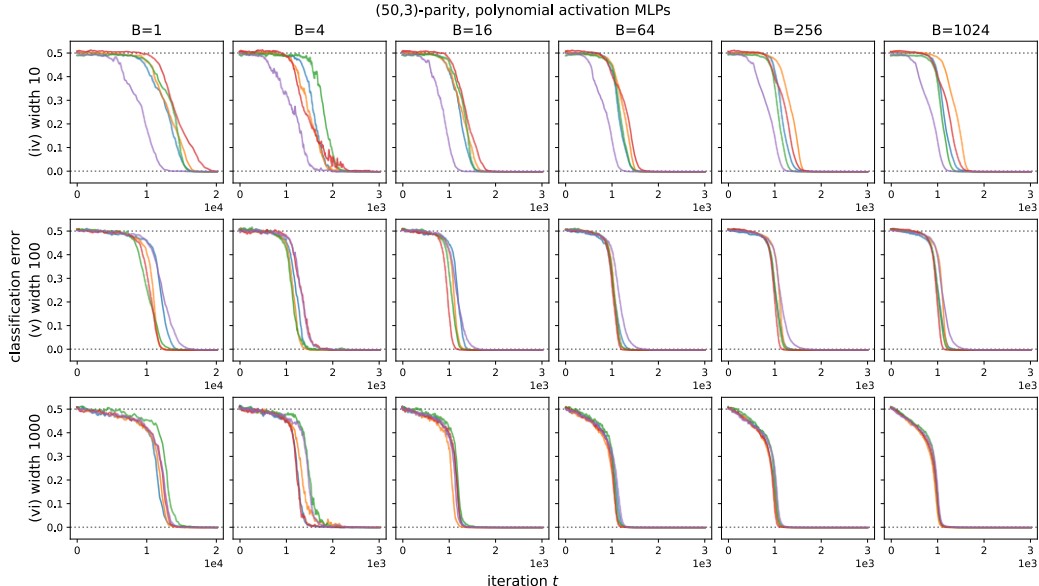

Figure 17: Additional training curves for polynomial-activation MLP architectures (iv), (v), (vi), on the (50, 3)-sparse parity problem. Unlike the other architectures, these settings exhibit continuous progress when the width $r$ and batch size $B$ are large.

# D  Details for all experiments

## D.1  Deep learning configurations

**Losses.**  Our "robust space" of empirical results use the following loss functions:

- Hinge: $\ell(y, \widehat{y}) := (1 - y\widehat{y})_+$.
- Square: $\ell(y, \widehat{y}) := (y - \widehat{y})^2$.
- Cross entropy: $\ell(y, \widehat{y}) := -\log \frac{e^{y\widehat{y}}}{1 + e^{y\widehat{y}}}$.

Additionally, the theoretical analysis considers the *correlation loss $\ell(y, \widehat{y}) := -y\widehat{y}$.*

In the configurations corresponding to all of the figures and convergence time experiments, we used the hinge loss. This was a relatively arbitrary choice (i.e. they appeared to be interchangeable upon running small experiments); an advantage of the hinge and square losses over cross entropy is that for architectures that can realize the parity function, there is a zero-loss solution with finite weights.

**Initializations.**  Our empirical results use the following i.i.d. weight initializations:

- Uniform on the interval $[-c, c]$, where the scale $c$ is chosen for all affine transformation parameters using the "Xavier initialization" convention (Glorot and Bengio, 2010). The experiments are quite tolerant to the particular choice of $c$ (as these are not deep networks); this choice, which is the default in deep learning packages, emphasizes that our positive empirical results hold under a *standard* initialization scheme.
- Gaussian with mean 0 and variance $\sigma^2$, selected using the "Kaiming initialization" convention (He et al., 2015).
- Bernoulli (i.e. random sign) initialization: the discrete distribution $\mathrm{Unif}(\{-c, c\})$, for the same choice of $c$ as for the uniform distribution.

### D.1.1  2-layer MLPs

We consider 2-layer MLPs $f(x; W, b, u) = u^\top \sigma(Wx + b)$ for two choices of activations:

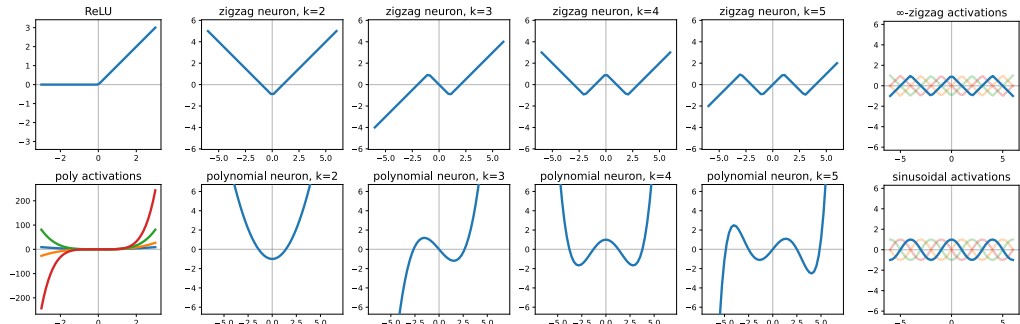

Figure 18: Visualizations of all activation functions considered in this work. As discussed in Section D.1.2, care must be taken to ensure that the architectures can realize sparse parities with the same bias $b$ (in particular, $b = 0$) across varying $k$. Multiple $k$-dependent displacements are shown for the $\infty$-zigzag and sinusoidal activations.

- ReLU: $\sigma(z) := (z)_+ = \max(0, z)$.
- Degree-$k$ polynomial: $\sigma(z) := z^k$.

In both cases, whenever $r \geq k$ (and, in the case of polynomial activations, choosing the degree to be $k$), there exists a width-$r$ MLP which can represent $k$-sparse parities: for all $(n, k)$ and $|S| = k$, there is a setting of $W, b, u$ such that $f(x; W, b, u) = \chi_S(x)$.

Note that if the output $f(x; \theta)$ is a degree-$k' < k$ polynomial in $x$ (e.g. an MLP with $\sigma(z) = z^{k'}$ activations), the architecture is incapable of representing a parity of $k$ inputs. In fact, it is incapable of representing any function that has a nonzero correlation with parity; this follows from orthogonality (Equation (2)).

### D.1.2 Single neurons

To explore the limits of concise parameterization for architectures capable of learning parities, we propose a variety of non-standard activation functions which allow a single neuron to learn sparse parities. These constructions leverage the fact that the parity is a nonlinear function of the sum of its inputs $w_S^\top x$, where $w_S := \sum_{i \in S} e_i$.

**$k$-zigzag activation.** $\sigma(\cdot)$ is the piecewise linear function which interpolates the $k + 1$ points $\{(k, +1), (k - 2, -1), (k - 4, +1), \ldots, (-k + 2, \pm 1), (-k, \mp 1)\}$ with $k$ linear regions $\{(-\infty, -k], [-k, -k + 2], \ldots, [k - 2, k], [k, +\infty)\}$. Then, $\sigma(w_S^\top x) = \chi_S(x)$.

**Oscillating polynomial activation.** $\sigma(\cdot)$ is the degree-$k$ polynomial which interpolates the same points as above.

**$\infty$-zigzag activation.** The *infinite* extension of the zigzag activation is the *triangle wave* function $\sigma(\cdot)$ which linearly interpolates the infinite set of points $\bigcup_{i \in \mathbb{Z}} \{(2i, +1), (2i + 1, -1)\}$. This can express parities of *arbitrary* degree. The $+1$ and $-1$ can be swapped (resulting in an activation which is equivalent when shifted by a bias term). However, in our experiments, we choose the sign convention depending on $k$ such that $\sigma(w_S^\top x) = +1$. This allows different convergence time curves to be more directly comparable across different $k$, since it removes the effects of the bias of the global minimizer alternating with $k$.

**Sinusoidal activation.** $\sigma(z) := \sin(z)$. The sinusoidal neuron $\sin(w^\top x + b)$ can also express parities of arbitrary degree, since it can interpolate the same set of points as the $\infty$-zigzag activation. In the experiments, we pick a shift $\beta$ and use the activation $\sigma(z) := \sin(\frac{\pi}{2} z + \beta)$, such that $\sigma(z)$ interpolates the same points as the sign convention selected for the $\infty$-zigzag activation. In the experiments in Section 3, the sinusoidal activation is additionally scaled by a factor of 2 ($z \mapsto \sigma(2z)$); this is interchangeable with scaling the learning rate and initialization, and is done to obtain more robust convergence in the particular setting of $(n, k) = (50, 3)$.

Figure 18 visualizes all families of activations considered in this paper.

### D.1.3  Parity Transformer

The Transformer experiments use a slightly simplified version of the architecture introduced in (Vaswani et al., 2017). In particular, it omits dropout, layer normalization, tied input/output embedding weights, and a positional embedding on the special [CLS] token. Including these does not change the results significantly; they are all present in the preliminary findings in (Edelman et al., 2021) (in which an "off-the-shelf" Transformer implementation successfully learns sparse parities). We specify the architecture below.

Our *Parity Transformer* has the following hyperparameters: sequence length $n$, token embedding dimension $d_{\mathrm{emb}}$, attention embedding dimension $d_{\mathrm{attn}}$, feedforward embedding dimension $d_{\mathrm{mlp}}$, and number of heads $H$. Its trainable parameters (together denoted by $\theta$) are:

- Token embeddings $E_{-1}, E_{+1}, E_{\texttt{[CLS]}} \in \mathbb{R}^{d_{\mathrm{emb}}}$ and position embeddings $P_1, \ldots, P_n \in \mathbb{R}^{d_{\mathrm{emb}}}$. Let $\theta_{\mathsf{emb}}$ denote this subset of parameters.
- Attention head matrices: $W_Q^{[h]}, W_K^{[h]}, W_V^{[h]} \in \mathbb{R}^{d_{\mathrm{emb}} \times d_{\mathrm{attn}}}$ and $W_{\mathrm{out}}^{[h]} \in \mathbb{R}^{d_{\mathrm{attn}} \times d_{\mathrm{emb}}}$, for $h = 1, \ldots, H$. Let $\theta_{\mathsf{attn}}$ denote this subset of parameters.
- MLP weights and biases: $W_1 \in \mathbb{R}^{d_{\mathrm{mlp}} \times d_{\mathrm{emb}}}, b_1 \in \mathbb{R}^{d_{\mathrm{mlp}}}, W_2 \in \mathbb{R}^{d_{\mathrm{mlp}} \times d_{\mathrm{emb}}}, b_2 \in d_{\mathrm{emb}}$. Let $\theta_{\mathsf{mlp}}$ denote this subset of parameters.
- Classification head: $u \in \mathbb{R}^{d_{\mathrm{emb}}}$.

Then,
$$f_{\mathsf{tf}}(x;\theta) := u^\top \left( f_{\mathsf{mlp}}(f_{\mathsf{attn}}(f_{\mathsf{emb}}(x;\theta_{\mathsf{emb}});\theta_{\mathsf{attn}});\theta_{\mathsf{mlp}}) \right),$$
where these submodules are defined by

- Embedding $f_{\mathsf{emb}} : \{\pm 1\}^n \to \mathbb{R}^{(n+1) \times d_{\mathrm{emb}}}$: $f_{\mathsf{emb}}(x;\theta_{\mathsf{emb}})_{i,:} := E_{x_i} + P_i$ for $i \in [n]$. We will include an the extra index [CLS], for which $f_{\mathsf{emb}}(x;\theta_{\mathsf{emb}})_{\texttt{[CLS]},:} := E_{\texttt{[CLS]}}$ (with no positional embedding). [CLS] stands for "classification", as in "use the output at this position to classify the sequence". This is a standard construction which makes the classifier permutation-invariant.
- Attention block $f_{\mathsf{attn}} : \mathbb{R}^{(n+1) \times d_{\mathrm{emb}}} \to \mathbb{R}^{d_{\mathrm{emb}}}$:

$$f_{\mathsf{attn}}(X;\theta_{\mathsf{attn}}) = X_{\texttt{[CLS]},:} + \sum_{h=1}^H \mathsf{softmax}\left( \frac{1}{\sqrt{d_{\mathrm{attn}}}} X_{\texttt{[CLS]},:} W_Q^{[h]} (X W_K^{[h]})^\top \right) W_V^{[h]} W_{\mathrm{out}}^{[h]},$$

where $\mathsf{softmax}(z) := \exp(z)/\mathbf{1}^\top \exp(z)$. Note that we have specialized this architecture to a single output, at the [CLS] position.
- MLP $f_{\mathsf{mlp}} : \mathbb{R}^{d_{\mathrm{emb}}} \to \mathbb{R}^{d_{\mathrm{emb}}}$:
$$f_{\mathsf{mlp}}(z;\theta_{\mathsf{mlp}}) := z + W_2 \sigma(W_1 x + b_1) + b_2,$$
where $\sigma(\cdot) = \mathsf{GeLU}(\cdot)$ (the Gaussian error linear unit) is the standard choice in Transformers.

**Training.** Each matrix-shaped parameter was initialized using PyTorch's default "Xavier uniform" convention. Unlike the other settings considered in this paper, we were unable to observe successful convergence beyond a few small $(n, k)$ using standard SGD. As is common practice when training Transformers, we used Adam (Kingma and Ba, 2014) with default adaptive parameters $\beta_1 = 0.9, \beta_2 = 0.999, \epsilon = 10^{-8}$ in our experiments. While there are more fine-grained accounts of why Adam outperforms vanilla SGD (Zhang et al., 2020; Agarwal et al., 2020), finding the optimal optimizer configuration and investigating ablations of this optimizer are outside the scope of this work. In this work, we only tune Adam's learning rate $\eta$.

### D.1.4  PolyNet

For positive integers $n, k$, the PolyNet architecture is parameterized by weights and biases $\theta := \{(w_i \in \mathbb{R}^n, b_i \in \mathbb{R})\}_{i=1}^k$, and is defined by

$$f_{\mathsf{PolyNet}}(x;\theta) := \prod_{i=1}^k (w_i^\top x + b_i).$$

Even with all biases $b_i$ set to 0, this architecture can realize a $k$-wise parity, by setting $\{w_i\} = \{e_j : j \in S\}$ in any permutation.

### D.1.5 Details for Figure 1 (left)

Figure 1 (left) shows training curves from 8 representative configurations, with online i.i.d. samples from the same distribution, corresponding to the $(n = 50, k = 3)$-sparse parity problem. The first row encompasses various MLP settings with standard activations:

- Setting (i): width-10 MLP with ReLU activation ($B = 32, \eta = 0.5$).
- Setting (i): width-10 MLP with ReLU activation, with large batches ($B = 1024, \eta = 0.05$).
- Setting (ii): width-100 MLP with ReLU activation, with tiny batches ($B = 1, \eta = 0.05$).
- Setting (iv): width-10 MLP with polynomial $\sigma(z) = z^3$ activation ($B = 32, \eta = 0.05$).

The second row shows other settings:

- Setting (vii): width-1 MLP with a piecewise linear $k$-zigzag activation ($B = 32, \eta = 0.2$).
- Setting (x): width-1 MLP with a sinusoidal activation (scaled and shifted for $k = 3$; see the discussion in Section D.1.2) ($B = 32, \eta = 0.05$).
- Setting (*iii): Parity Transformer, with $d_{\text{emb}} = 1024, d_{\text{attn}} = 8, H = 128$ ($B = 32, \eta = 5 \times 10^{-4}$).
- Setting (xv): degree-3 PolyNet ($B = 32, \eta = 0.07$).

### D.1.6 Details for Figure 6

The first row uses the width-10 ReLU MLP configuration $(ii)$, holding $B = 32$ and $\eta = 0.1$ while varying the task difficulty across 6 settings: $(n, k) \in \{(30, 3), (60, 3), (90, 3), (30, 4), (30, 5), (30, 6)\}$. The remaining plots are all for the $(50, 3)$ setting.

The second row uses the $k = 3$ PolyNet configuration $(xv)$, varying $(B, \eta) \in \{(1, 0.005), (4, 0.01), (16, 0.1), (64, 0.1), (256, 0.1), (1024, 0.1)\}$.

The third row uses the minimally-wide configurations (*i), (*ii), (vii), (viii), (xi), (xii) (thus presenting an example for each non-standard activation), holding batch size $B = 1$. $\eta = 0.1$ in each of the cases except (*ii), where $\eta = 0.01$.

The fourth row uses three large architectures: settings (iii), (vi), and (*iii), with $(B, \eta) \in \{(1, 0.1), (1024, 0.1), (1, 0.001), (1024, 0.01), (32, 0.0003), (1024, 0.0003).\}$ (*iii) uses the Adam optimizer instead of SGD.

### D.1.7 Details for Figure 10

Figure 10 contains scaling plots in various settings for the median convergence time $t_c$. Below, we give comprehensive details about these settings. For each of these runs, we chose $B = 32$ (settings with smaller batch sizes exhibited additional variance; with larger batch sizes, the models were slower to converge), as well as the hinge loss. We used SGD with constant learning rate $\eta$ (enumerated below), except in setting (*iii).

The top row shows MLP settings (i) through (vi). From left to right:

- Setting (i): width-10 MLP with ReLU activation ($\eta = 1$).
- Setting (ii): width-100 MLP with ReLU activation ($\eta = 1$).
- Setting (iii): width-1000 MLP with ReLU activation ($\eta = 1$).
- Setting (iv): width-10 MLP with $\sigma(z) = z^k$ activation ($\eta = 0.01$).
- Setting (v): width-100 MLP with $\sigma(z) = z^k$ activation ($\eta = 0.01$).
- Setting (vi): width-1000 MLP with $\sigma(z) = z^k$ activation ($\eta = 0.01$).

The bottom row shows miscellaneous settings. From left to right:

- Setting (vii): width-1 MLP with degree-$k$ oscillating polynomial activation interpolating the parity function ($\eta = 0.01$).

- Setting (xiv): single sinusoidal neuron with no second layer ($\eta = 0.01$).

- Setting (*iii): Parity Transformer, with $d_{\mathrm{emb}} = 1024, d_{\mathrm{attn}} = 8, H = 128$ ($B = 32, \eta = 3 \times 10^{-4}$).

- Setting (iv): degree-$k$ PolyNet ($\eta = 0.05$).

- Setting (ii): width-100 MLP with ReLU activation ($\eta = 1$), showing an expanded range of $n$ for smaller $k$.

- Setting (xv): width-100 MLP with $\sigma(z) = z^k$ activation ($\eta = 1$), showing an expanded range of $n$ for smaller $k$.

### D.2  Training curves and convergence time plots

For all example training curves in all figures (in Sections 3 and 5, as well as the appendix), population losses and accuracies are approximated using a batch of size 8192, sampled once at the beginning of training from the same distribution $\mathcal{D}_S$. All plots of single representative training runs use a fixed random seed (`torch.manual_seed(0)`); when $R$ training runs are shown, seeds $0, \ldots, R-1$ are used.

In Figures 7 and 8, validation accuracies were recorded every 10 iterations, and a run was recorded as converged if it reached $100\%$ accuracy within $10^5$ iterations; we report the 10[th] percentile over 25 random seeds, to reduce variance arising from the more initialization-sensitive settings. In Figure 9, coarse-grained scaling estimates for the (10[th] percentile) convergence time are computed as follows: for $n \in \mathcal{N} := \{10, 20, 30\}$, the smallest $\alpha$ is chosen such that $t_c \leq c \cdot (n - n_0)^\alpha$, choosing $n_0 = \min \mathcal{N} - 1 = 9$, so that $c = t_c$ at $n = 10$. These estimates are calculated to give quantitative order-of-magnitude upper bounds for the convergence time. Indeed, the power-law convergence times do not extrapolate at a constant learning rate; see Figure 2 *(right)*, the "larger $n$" plots in Figure 10, and the discussion on batch sizes and learning rates in Appendix C.2.

To reduce computational load, for the larger-scale probes of convergence times $t_c$, validation accuracies were instead checked on a sample of size 128. For the underparameterized networks (i.e. unable to represent parity, but can still get a meaningful gradient signal), this threshold was changed to 10 consecutive batches with accuracy at least $55\%$. Note that for parity learning in particular, a weak learner can be converted into a strong learner: there is an efficient algorithm (Goldreich and Levin, 1989; Kushilevitz and Mansour, 1993) which, given a classifier which achieves $1/2 + \epsilon$ accuracy on $\mathcal{D}_S$ for a constant $\epsilon > 0$, outputs $S$ with high probability.

In the median convergence time plots in Figure 1 *(right)*, Figure 2 *(right)*, and Figure 10, error bars for median convergence times in all plots are $95\%$ confidence intervals, computed from 100 bootstrap samples. Each point on the each curve corresponds to 1000 random trials. Halted curves signify more than $50\%$ of runs failing to converge within $T = 10^5$ iterations (hence, infinite medians).

### D.3  Implementation, hardware, and compute time

All training experiments were implemented using PyTorch (Paszke et al., 2019).

Although most of the networks considered in the main empirical results are relatively small, a large ($\sim 10^8$) total number of models were trained to certify the "robust space" of results and obtain precise scaling curves. These individual experiments were not large enough to benefit from GPU acceleration; on an internal cluster, the CPU compute expenditure totaled approximately 1500 CPU hours.

A subset of these experiments stood to benefit from GPU acceleration: width $r \geq 100$ MLPs; scaling behaviors for $n \geq 100$; all experiments involving Transformers. These were performed with NVIDIA Tesla P100, Tesla P40, and RTX A6000 GPUs on an internal cluster, consuming a total of approximately 200 GPU hours.