# OpenReview forum: "Hidden Progress in Deep Learning: SGD Learns Parities Near the Computational Limit"
_NeurIPS.cc/2022/Conference — NeurIPS 2022 Accept_

### Official Review · Reviewer_3q9X · 2022-07-10

**Rating:** 7
**Confidence:** 3
**Soundness:** 4 excellent
**Presentation:** 3 good
**Contribution:** 3 good

**Summary:**

This paper studies the problem of learning k-sparse parties using SGD on neural network models. In particular, standard deep learning approaches (running gradient-based methods from random initialization) exhibit discontinuous improvement in performance with increasing training time. Importantly, this kind of phase transition is quite robust and SGD can robustly find the sparse subset in a number of iterations that nearly matches optimal lower bounds without sparsity prior. This is strongly against the "random search" hypothesis. Theoretically, the authors show that the first gradient step contains important information to solve the parity problem and feature learning is crucial (as no kernel can solve the sparse parity problem). Finally, the authors argue there is a hidden progress measure which under SGD is steadily improving.

**Questions:**

See some of my comments on the weaknesses.

Other questions:
- I wonder if the authors tried different batch sizes in solving the problem. An empirical batch size scaling curve may provide some insights into the learning dynamics. Also, experiments with SGLD could also be interesting.


**Limitations:**

The authors discussed some of the limitations and broad impact at the end of the paper. I don't see any negative societal impact of this work.

**Strengths And Weaknesses:**

Strengths:
- The idea of applying deep learning methods to solve the sparse parity problem is novel and inspiring.
- The authors conducted comprehensive empirical studies and many results are quite interesting. For example, the results are robust w.r.t the architectures, data subsampling, and initialization schemes, but sensitive to particular initialization.
- The finding that SGD could solve the problem in a number of iterations that matches the lower bounds is surprising and interesting. I believe this would inspire many other follow-up works.
- The authors further proved that feature learning is important and the first gradient step can learn features that contain enough information for solving the problem, even though under a restricted setting.

Weaknesses:
- The authors made the argument that random search is the most natural hypothesis to explain the success of SGD in learning parities. I don't really buy this argument. To me, gradient-based methods have little to do with random search. It is known in the community that SGD uses only local information and is very different from random search. In my opinion, it is not surprising that SGD can do better than random search, instead it is surprising that it can match the optimal lower bound.
- The hidden progress measure looks quite arbitrary to me. The authors argue gradual weight movement is a potential measure. However, it is unclear how the weight movement connects with "real" progress. There are many problems that one can observe weight movement but no improvement in performance.

---

> ### Author Response · Authors · 2022-08-02
> **Response to Reviewer #3q9X**
>
> Thanks for the encouraging feedback and thoughtful suggestions.
>
> - **Random search as a natural hypothesis.** [W1] We certainly agree that this could be subjective. One natural reason to hypothesize random search (discussed in Appendix A.1) comes from the problem rather than the algorithm: all incorrect hypotheses (parities of incorrect subsets of indices) have error as bad as random guessing. Thus, if the learner is searching over subsets, there is _no_ local information available in the loss, and the learner cannot make “partial progress”, which would neatly (but incorrectly) explain the observed long plateau in accuracy during training followed by sudden progress. (This symmetry is broken by the nonlinearity and Fourier anti-concentration of neural nets.) Furthermore, no known algorithm is significantly faster than random search. Some other natural reasons: (1) the long tails in the Figure 2 (left) resemble those of a geometric distribution, which would be the distribution resulting from random search; (2) stochastic gradient Langevin dynamics are explicitly used for random sampling in other regimes.
> - **Intuition for progress measure.** [W2] To reiterate the discussion in Appendix C.3 on the intuition for the $\ell_\infty$ progress measure: under the Fourier gap condition, we can view the trajectory of SGD on each coordinate as a unit-variance random walk, with a slightly larger bias (i.e. drift) on the relevant coordinates. This progress measure grows as $\Theta(t)$ when at least one weight has enough drift, as opposed to $\Theta(\sqrt{t})$ (if there were no drift). We claim that a progress measure is meaningful if progress according to the measure is correlated (over the randomness of initialization and SGD) with amount of time until high accuracy is reached—e.g., the first step where accuracy is larger than (say) 80% and the first step where progress measure is larger than some value $X$ are highly correlated. This correlation would not exist if the learning process were just stumbling in the dark and thus memoryless. For disjoint-PolyNets, the connection between amount of movement of the weights and time-until-high-accuracy is implied by our theory. In the next version, we will make this explicit, and add experiments measuring this correlation for MLPs.
> - **Batch sizes.** [Q1] Note that we tried a variety of batch sizes (between 1 and 1024) in the “robust space of empirical results”. We found that the most sample-efficient (and thus sequential-compute-efficient) runs were at small batch sizes (4 to 32); we have added supplementary tables and figures in the revision. A challenge towards obtaining smooth scaling law curves is that the optimal learning rate is quite sensitive to batch size.
> - **Langevin dynamics.** [Q1] We appreciate the suggestion of explicitly incorporating random search (i.e. SGLD), and believe this is a promising direction for future work, especially in exploring training algorithms beyond SGD (on synthetic or natural tasks).

---

### Official Review · Reviewer_hMvu · 2022-07-12

**Rating:** 4
**Confidence:** 4
**Soundness:** 2 fair
**Presentation:** 2 fair
**Contribution:** 3 good

**Summary:**

UPDATED SCORE AFTER REBUTTAL.

The authors study parity functions of size k on n bits. Since there are $2^n$ subsets of the literals, sampling from them uniformly at random would take time exponential in n to find a specific parity function. In addition to this naive approach, there are standard lower bounds for learning parity functions in the SQ model (as opposed to the more general PAC model). The authors claim, using experiments, that training neural networks with SGD (without any sparsity penalty) matches this lower bound. Finally the authors suggest a "hidden progress measure" that changes smoothly despite loss and accuracy plateauing.

**Questions:**

- What is D for the NTK in Theorem 6? Is this the size of the training set?
- What does it mean for f to be continuous in line 150 on a discrete space?
- Figure 2 right: Why are these curves not straight if there is power-law scaling?
- Does the argument in lines 201-203 assume initialization samples uniformly from all subsets? Can you provide more detail.
- Please provide a citation for the Fourier spectrum of Majority.
- How can you distinguish between computational and sample resources in online learning where they are linked?


**Limitations:**

No concerns about negative societal impact.



**Strengths And Weaknesses:**

Strengths
- The problem proposed by the authors is interesting and potentially a useful model for "grokking."
- The paper considers a variety of different architectures ranging from more realistic to more idealized.

Weaknesses
- Many of the claims lack clarity or seem overstated.
- "Empirical Finding 1": I'm confused by the quantifiers in this statement. Restricting to small n and k is odd as the lower bound from the SQ model is asymptotic in n. Even the empirical evidence is not convincing. In Figure 2 the lines are clearly not straight, undercutting the claim.
- The motivation for the progress measure is weak and no explanation for the infinity-norm is given. This progress measure also does not seem to distinguish between the "stumbling in the dark" model as surely the progress measure would also increase in this case.
- Theorem 6 states there exists some distribution over $(x,y)$ that cannot be learned. This seems quite different than the experiments performed for neural networks, which I believe used the uniform distribution rather than asking whether they can learn any distribution over the literals.

---

> ### Author Response · Authors · 2022-08-02
> **Response to Reviewer #hMvu**
>
> Thanks for the review.
>
> **Non-linear scaling curves** [W1&2, Q3] The reviewer uses the observation that the scaling law exponents increase as a basis to perceive “overstated claims”. We believe that this is an erroneous interpretation, and clarify below.
> - We stress that the degradation of the scaling law curves for standard training, while a drawback for learning parities with standard training, actually reinforces our analysis; it is further evidence for a “hidden accumulated progress” mechanism. The success times of random search, a memoryless algorithm, would scale smoothly as $\sim n^k$. Instead, because SGD gradually accumulates a signal of magnitude $n^{-O(k)}$ in the population gradient over $n^{O(k)}$ iterations, many things can go wrong: numerical error; “recency bias”/“catastrophic forgetting” (i.e. insufficient sensitivity to early steps); a drifting population gradient as the weights drift (as discussed in the final paragraph of Appendix C.2). We have not attempted to disentangle these issues in this work. We see the shape of the scaling law curves as a finding of our work that is worthy of future study.
> - To reiterate our empirical claims: our claim is that a variety of neural nets, with standard initialization and training, can solve sparse parity tasks, with an apparent $c\cdot n^{\alpha k}$ scaling for _small_ $(n,k)$. For small $n, k$, we are in the regime of stochastic approximation, where batch size and SGD steps are approximately interchangeable (see response to Reviewer #5wtH). As $n, k$ get large, the scaling worsens, and this is also part of our empirical findings (hence the enlarged plots in Figure 7). We then show convergence with asymptotic $n^{O(k)}$ data and computation in the (suitably idealized) settings we are able to analyze (this is why the title states “SGD learns parities near the SQ limit”). Our message is not that standard training (as is used in practice) is optimal for parity problems– rather, that standard training approximates an idealized mechanism which _does_ achieve the $n^{O(k)}$ rate.
> - We understand the confusion, as this subtlety was presented a little too succinctly. We will address this in the final version.
>
> **Motivation for the progress measure**. [W3] See response to Reviewer #3q9X.
>
> **Linked computation and sample resources in online learning**. [Q6] The main empirical and theoretical results are in the online setting, which couples the resources of computation and dataset size. Empirically, Figure 3 and Appendix C.5 show that this is not necessary: architectures can successfully train and generalize in the multi-pass (i.e. finite-sample ERM) setting. However, this is very difficult to characterize theoretically: the gradient estimators become correlated, thus biased. We ran our scaling-law experiments in the online learning setting to elicit cleaner scaling laws, removing the irregularities induced by overfitting.
>
> Minor points:
> - **Clarification on NTK**. [W4, Q1] We apologize for the ambiguous phrasing. The distribution $\mathcal{D}$ is indeed restricted to be uniform over the inputs as in the rest of our paper, so there is no mismatch with the experiments. Also, $D$ is the embedding dimension (thus, the network width).
> - **Continuity**. [Q2] Continuous with respect to the parameters (so that gradient descent is well-defined).
> - **Lines 201-203: which initializations**? [Q4] This argument holds for coordinate-wise i.i.d. (thus, non-sparse) initializations, which are the dominant initialization schemes in practice.
> - **Fourier spectrum of Majority**. [Q5] See reference: O’Donnell. “Analysis of Boolean Functions”.

---

> > ### Comment · Reviewer_hMvu · 2022-08-08
> > **Response to authors**
> >
> > I'm grateful to the authors for their response to my questions, particularly for the additional discussion of the progress measure. Clearly there are some interesting findings in the paper, but I still have concerns about the clarity of the presentation and the strength of the empirical evidence. Addressing these issues would improve the paper significantly. I am increasing my score by 1, but I still do not feel the paper should be accepted in its current form.

---

> > > ### Author Response · Authors · 2022-08-08
> > > **Follow-up**
> > >
> > > Thanks for the follow-up.
> > >
> > > We are wondering if the uploaded revision addresses your concerns about clarity.
> > >
> > > Also, are there other remaining concerns about the “strength of empirical evidence”, after our clarifications in the response? (We have not yet incorporated these clarifications into the main paper, due to the page limit for the rebuttal revision.)
> > >
> > > Could the reviewer please elaborate on remaining concrete concerns? If there are other points that are still not clear in the new revision, please let us know.

---

### Official Review · Reviewer_5wtH · 2022-07-20

**Rating:** 6
**Confidence:** 4
**Soundness:** 2 fair
**Presentation:** 2 fair
**Contribution:** 2 fair

**Summary:**

Inspired by the discontinuous phase transition of running time, this paper studies the computation issue of k-sparse parities of n bits via deep learning method. It shows two main results: 1. With standard initialization and training, SGD can solve the k-sparse n-dimensional parities problem at optimal computation time n^(Ω(k)) steps in statistical query (SQ) model, which is much faster than the convergence time of random search 2^(Ω(n)) as previous conjecture. Additionally, it comes up with a “progress measure”, which can indicate when the sudden increasing of loss and accuracy happens. 2. It empirically and theoretically shows that SGD can do feature learning for parity problem even without sparse prior.

**Questions:**

1. I don’t know whether the input from uniform {+1,-1}^n is practical setting in k-sparse n-dimensional parities problem. If not, I think it is a too strong assumption to understand the process of training deep learning.
2. In theoretical part, the author extends the theoretical part from large batch size n^(Ο(k)) to number of steps n^(Ο(k)) , I don’t understand why we can do such extension and I think it would be helpful if there is an intuitive explanation.
3. From my understanding, does the convergence time of this parity problem only work for the special network structure “PolyNet”? If so, I think it is far from practice and its impact may be limited.
4. It was mentioned in Section 2 that  the results are based on online setting. But it is not clearly mentioned in Theorem 4. Does each iteration in Theorem 4 requires new i.i.d. samples? If so, this seems a very strong assumption to hold in practice.
5. Theorem 4 only requires ploy(k,r) iterations, which is much smaller than the number of iterations n^{O(k)} observed in experiments . While Theorem 4 and experiments have different batch sizes, this discrepancy needs more explanation in order to make Theorem 4 as a supporting result for the experiments.
6. I am curious about whether there is more naïve explanation to explain why progress measure can indicate the phase transition once the measure passes the certain threshold.

**Limitations:**

This paper does not have the limitations and potential negative societal impact.

**Strengths And Weaknesses:**

Strength:
1. Under specific setting, the author empirically demonstrates an interesting phenomenon under that SGD can solve k-sparse b-dimensional parities problem at optimal computation time in SQ algorithms, which contrasts to random search process as previously thought. Apart from empirical results, this paper provides theoretical analysis of why SGD is successful in solving the parity problem.
2. This paper has good structure. It starts from the phase transition phenomenon, and then it shows a theoretically analysis with a simple warm-up example as the introduction, which gives me an intuition.
3. The proposed progress measure seems to be a useful indicator and correlated with the performance (loss and accuracy).

Weakness
1. There is lack of explanation forsome theorems, which may make reader confused. For example, the author says that there exist a (n, k) parity distribution and satisfies some inequality constraint with an embedding in theorem 6, but I don’t know how do we get this D-dimensional embedding, is it the output of the network? And how can we use this theorem in practice?
2. There is lack of explanation for some notations. For example, what “e” denotes in line 281 and what “v(t)” is in theorem 7 (line 291).
3. The purpose of investigating the parity problem needs to further explain.  If the purpose is to understand deep networks,  why we should start from this parity problem?

---

> ### Author Response · Authors · 2022-08-02
> **Response to Reviewer 5wtH**
>
> Thanks for the review, and for raising several excellent questions. Responses to main points:
>
> **Why parity?** [Question 1; Weakness 3] The reviewer’s evaluation seems to primarily hinge upon a question of motivation: *“if the purpose is to understand deep networks, why should we start from this parity problem?”*. Below, we provide some additional context.
> * Parities (a.k.a. monomials) are the building blocks of all combinatorial functions: they are the basis elements in the Fourier analysis of Boolean functions. A construction that recurs throughout theoretical computer science is to learn combinatorial functions one parity at a time: e.g. [1, 2, 3, 4].
> * Parities under the uniform distribution are the “hardest” combinatorial functions to learn. They are maximally sensitive to the relevant inputs, and they reveal no information about the hidden features from correlations and lower-order ($\leq k-1$) moments (this gives rise to SQ lower bounds). Thus, they form a canonical family of problems with a large computational-statistical gap: the relevant features are identifiable from $O(k \log n)$ samples, but (with a small amount of noise) no known algorithm can solve the problem much faster than exhaustive search ($n^{O(k)}$ computational steps). Thus, the synthetic nature of this problem should not be viewed as “assumptions” which make the problem easier-- rather, this setting provably exemplifies the computational difficulty of feature learning.
> We hope the above will help to adjust the reviewer’s perspective on this setting– rather than a “too-strong assumption”, we view parity as a fundamental setting in which to understand how deep learning learns combinatorial functions.
>
> **Theory vs. experiments.** [Q2, Q4, Q5] The reviewer correctly notes that our theory does not explain the full range of our empirical results. A general comment is that in each of the possible directions for resolving this gap, we ran into challenging mathematical obstructions (which would be of independent interest if resolved). We hope that the reviewer can focus more on evaluating the novelty and technical challenge in the results we *are* able to prove.
> * [Q5] Theorem 4 is restricted to the large-batch setting because we were unable to find (or find prior work on) necessary conditions for general linear threshold functions to exhibit Fourier anti-concentration (even though we observe this in practice). We attempted wherever possible to bridge the gap: a more tractable analysis for a different architecture, and the discussion on conjectured Fourier gaps in Appendix C.1.
> * [Q2, Q4] The main empirical and theoretical results are in the online setting, which couples the resources of computation and dataset size. Empirically, Figure 3 and Appendix C.5 show that this is not necessary: architectures can successfully train and generalize in the multi-pass (i.e. finite-sample ERM) setting. However, this is very difficult to characterize theoretically: the gradient estimators become correlated, thus biased. We ran our scaling-law experiments in the online learning setting to elicit cleaner scaling laws, removing the irregularities induced by overfitting.
> * [Q5] This discrepancy can be heuristically resolved by the standard technique of stochastic approximation: interchanging $1$ SGD step of size $\eta$ at batch size $B$ with $B$ steps of size $\eta/B$ at batch size $1$, viewing the second-order variance term as negligible. (This is most commonly invoked in deep learning as the “linear scaling rule”.) However, turning this into an end-to-end theorem would require a better understanding of Fourier gaps for the intermediate SGD iterates.
>
> Minor points:
>
> **Clarifications on notation.** [W2] $e$ in line 281 refers to Euler’s constant. $v(t)$ in line 291 is the trajectory of the weights on the relevant indices, as written in the lines above.
>
> **Clarification on NTK.** [W1] In Theorem 6, we are simply pointing out that for a fixed kernel (such as NTK) to solve the parity task, the embedding dimension $D$ (thus, the network width) needs to be at least $\Omega(n^k)$. Thus, our empirical findings with non-overparameterized networks cannot be explained by NTK. We agree that we made this a little too concise, and will expand based on the reviewer’s feedback.
>
> **Intuition for progress measure.** [Q6] See response to Reviewer #3q9X.
>
> [1] Goldreich & Levin, “A Hard-Core Predicate for All One-Way Functions”.
>
> [2] Kushilevitz & Mansour, “Learning Decision Trees Using the Fourier Spectrum”.
>
> [3] Feldman et al., “On Agnostic Learning of Parities, Monomials and Halfspaces”.
>
> [4] Reyzin, “On Boosting Sparse Parities”.

---

> > ### Comment · Reviewer_5wtH · 2022-08-08
> > **Response to authors**
> >
> > Thanks authors for your detailed rebuttal and explanation, which addresses my many questions.
> >
> > Through your introduction, I could understand the importance and difficulties of studying parity problem in computer science community. Nonetheless, I am still confused about why we should start from parity problem to understand the scaling phenomenon in deep learning, because most inputs are not restricted to binary values ($\pm 1$) in practice and the setting of parity is far from such situation.
> >
> > Again, we recommend the authors to give some explanations for some theorem, such as Theorem 6, 8.

---

> > > ### Author Response · Authors · 2022-08-08
> > > **Follow-up**
> > >
> > > Thanks for the follow-up.
> > >
> > > We strongly believe that a theoretical setting does not need all of the superficial qualities of a real-world setup (in this case, “vocab size larger than 2”) to provide a meaningful lens into modern empirical phenomena. **Parity is the simplest setting which is statistically easy but computationally hard**, and is thus an apt setting for understanding how gradient-based methods solve inherently combinatorial feature learning tasks. In fact, computational hardness results for more complicated settings (e.g. learning an MLP on a continuous domain) are attained via reductions to the hardness of learning parities [1, 2]; this is also trivially true for larger discrete domains.
> > >
> > > We would also like to highlight again that many purely empirical works leverage binary parity as a benchmark for learning hard combinatorial functions: [3, 4] as previously mentioned, but also [5, 6].
> > >
> > > We have improved the exposition around Theorem 8 in the revision, which we hope addresses the reviewer’s concern. We have not yet incorporated clarifications for Theorem 6 (which are present in the responses regarding NTK) into the main paper, due to the page limit for the rebuttal revision. We will add them in the final version.
> > >
> > > [1] Klivans & Kothari, 2014. “Algorithms and SQ Lower Bounds for PAC Learning One-Hidden-Layer ReLU Networks”.
> > >
> > > [2] Goel et al., 2019. “Time/Accuracy Tradeoffs for Learning a ReLU with respect to Gaussian Marginals”.
> > >
> > > [3] Lu et al., 2020. “Pretrained Transformers as Universal Computation Engines”.
> > >
> > > [4] Anil et al., 2022. “Exploring Length Generalization in Large Language Models”.
> > >
> > > [5] Graves et al., 2016. “Adaptive Computation Time for Recurrent Neural Networks.”
> > >
> > > [6] Banino et al., 2021. “PonderNet: Learning to Ponder.”

---

> > > > ### Comment · Reviewer_5wtH · 2022-08-08
> > > > **Furthur response to authors**
> > > >
> > > > Thanks for the authors' reply and clarification.
> > > >
> > > > By reading the references in the recent response, I better understand the meaning of parity in practice and it solves my most concerns. Additionally, I recommend the authors to incorporate the influence of parity or some real-world examples into the appendix of your final version, otherwise readers who are unfamiliar with the background cannot directly link the parity problem to understanding the scaling in DL. It is claimed in the introduction section and I think they are not linked very well in current version.
> > > >
> > > > And a simple example,  such as "the parity of the bitstring $[0, 1, 1, 0, 1]$ is
> > > > “odd" (or 1) as opposed to “even" (or 0), because there is an odd number of 1s in the bit-string" in [4] (Anil et al., 2022. “Exploring Length Generalization in Large Language Models”.), may be more helpful for reader to have a direct understanding of the parity setting, because too many notations may make readers confused. It takes me some time to understand some notations, for example, the $[-1, 1]^n$.
> > > >
> > > > After discussing with the authors, I decide to raise my score.

---

### Author Response · Authors · 2022-08-02
**General response to all reviewers**

We thank their reviewers for their feedback and appreciate their appreciation for the phenomena we uncover. General comments:
* We have submitted a revised main paper and appendix, with numerous clarifications, small improvements, and additional references. Some highlights: (1) we have strengthened the disjoint-PolyNet analysis, which now has an end-to-end result in discrete time and any batch size $B \geq 1$; (2) we expanded the table of coarse-grained scaling estimates; (3) we made the informal theorem statements clearer.
* As we discuss in greater depth in response to Reviewer #5wtH, we believe that the parity problem, aside from its long history in many areas of computer science, mathematics, and statistics, is a natural one to study for the emerging combinatorial/reasoning/arithmetic abilities of deep learning. It is arguably the simplest example of a fundamentally combinatorial task, which (as we show) displays fundamentally different dynamics to other learning tasks. In that sense, it is a minimalistic analogue of the combinatorial/arithmetic tasks studied in the “grokking” [1] and “BIG-Bench” [2] papers. We think that the facts that (1) parity learning displays strong threshold phenomena (2) sparse parities are learnable even by architectures without sparsity bias, and (3) there is a way to measure progress even when loss is flat, are informative for further exploration of these types of problems.
* There is growing interest in using parity as a benchmark for robustly learning combinatorial functions and long-range dependencies. We have cited some of these works in the manuscript: [3, 4]. Since the submission deadline, we have become aware that parity is a fundamental benchmark for length generalization in deep models [5].
* Several reviewers asked about our notion of a “progress measure” (any function of the state of the algorithm which is predictive of the time until learning). We have clarified the reasoning (picking out the most biased of $n$ random walks) in the responses. In general, many algorithms could have a progress measure, and the choice is somewhat arbitrary: for example, for an algorithm that deterministically enumerates over subsets, the iteration count $t$ is a progress measure. For the purposes of this work, the existence of a smoothly varying progress measure refutes the “pure random search” hypothesis (a.k.a. “stumbling in the dark”)-- such an algorithm is memoryless. Further exploration of practical progress measures in deep learning is an interesting area for future work.
* Several reviewers pointed out cases where our paper does not provide full theoretical understanding of the phenomena we uncover. We stress that our work advances the community’s conceptual and theoretical understanding in various respects, but also raises various open mathematical questions that we hope will be fruitful for future study.

[1] Power et al. “Grokking: Generalization Beyond Overfitting on Small Algorithmic Datasets.”

[2] Srivastava et al. “Beyond the Imitation Game: Quantifying and Extrapolating the Capabilities of Language Models.”

[3] Hahn, “Theoretical Limitations of Self-Attention in Neural Sequence Models”.

[4] Lu et al., “Pretrained Transformers as Universal Computation Engines”.

[5] Anil et al. “Exploring Length Generalization in Large Language Models”.

---

### Comment · Area_Chair_sZPt · 2022-08-07
**Discussion period**

Thank you to all the reviewers for the great effort in reviewing the paper and the authors for the responses.

As in the discussion period, I want to ensure that reviewers have read the authors' responses and engage with the authors if needed.

If you haven't done this, could you please take a moment to read through the authors' responses, update the reviews to indicate that you have read the authors' responses, or communicate with the authors if needed? You can also share in private conversations with the reviewing team.

Please continue to share your thoughts. Thank you!

---

### Meta-Review · Area_Chair_sZPt · 2022-08-28

**Recommendation:** Accept
**Confidence:** Less certain

**Metareview:**

This paper studies learning $k$-sparse parities of $n$ bits by a neural network. The authors show empirically that SGD efficiently learns sparse parties and also provide theoretical analysis for 2-layer MLPs. In particular, the authors show that training neural networks with SGD but without any sparsity penalty can solve the problem with $n^{O(k)}$ samples and steps, matching the standard statistical query (SQ) lower bounds for learning $k$-sparse parties of $n$ bits.

The major comments pointed out by reviewers include (1) the motivation for studying the parity problem and the progress measure, (2) the mismatch between the experimental results and claim, and (3) the gap between the theoretical results and the empirical observations. To address these comments, the authors have submitted a revised version which includes a further introduction to the parity learning problem and clarifies the reasoning for choosing the progress measure. The authors also clarify the empirical claims concerning the experimental results. Though the gap between the theoretical results and the empirical observations can not be improved in the current version and needs further investigation in future work, this paper advances our theoretical understanding of the computational aspect of scaling in deep learning through the parity learning problem. So I recommend accepting, but the authors are advised to incorporate the discussions about empirical claims in the final version.

**Award:**

No

---

### Decision · Program_Chairs · 2022-09-14

Accept